# The long-term sea-level commitment from Antarctica

**Ann Kristin Klose[1,2], Violaine Coulon[3], Frank Pattyn[3], and Ricarda Winkelmann[1,2,4]**

[1]Potsdam Institute for Climate Impact Research (PIK), Member of the Leibniz Association,
PO Box 6012 03, 14412 Potsdam, Germany
[2]Institute of Physics and Astronomy, University of Potsdam, 14476 Potsdam, Germany
[3]Laboratoire de Glaciologie, Université libre de Bruxelles (ULB), Brussels, Belgium
[4]Department of Evolutionary Earth Systems Science, Max Planck Institute of Geoanthropology, 07745 Jena, Germany

**Correspondence:** Ann Kristin Klose (annkristin.klose@pik-potsdam.de) and Ricarda Winkelmann
(ricarda.winkelmann@pik-potsdam.de)

**Abstract.** The evolution of the Antarctic Ice Sheet is of vital importance given the coastal and societal implications of ice loss, with a potential to raise sea level by up to 58 m if it melts entirely. However, future ice-sheet trajectories remain highly uncertain. One of the main sources of uncertainty is related to nonlinear processes and feedbacks CE1 between the ice sheet and the Earth System on different timescales. Due to these feedbacks and ice-sheet inertia, ice loss may already be triggered in the next decades or centuries and will then unfold thereafter on timescales on the order of multiple centuries to millennia. This committed Antarctic sea-level contribution is not reflected in typical sea-level projections based on mass balance changes of the Antarctic Ice Sheet, which often cover decadal-to-centennial timescales. Here, using two ice-sheet models, we systematically assess the long-term multi-millennial sea-level commitment from Antarctica in response to warming projected over the next centuries under low- and high-emission pathways. This allows us to bring together the time horizon of stakeholder planning and the much longer response times of the Antarctic Ice Sheet.

Our results show that warming levels representative of the lower-emission pathway, SSP1-2.6, may already result in an Antarctic mass loss of up to 6 m of sea-level equivalent on multi-millennial timescales. This committed mass loss is due to a strong grounding-line retreat in the West Antarctic Amundsen Sea embayment as well as potential drainage from the Ross Ice Shelf catchment and onset of ice loss from Wilkes subglacial basin in East Antarctica. Beyond the warming levels reached by the end of this century under the higher-emission trajectory, SSP5-8.5, a collapse of the West Antarctic Ice Sheet is triggered in the entire ensemble of simulations from both ice-sheet models. Under enhanced warming, next to ice loss from the marine subglacial basins, we also find a substantial decline in ice volume grounded above sea level in East Antarctica. Over the next millennia, this gives rise to a sea-level increase of up to 40 m in our simulations, stressing the importance of including the committed Antarctic sea-level contribution in future projections.

## 1 Introduction

The future sea-level contribution from the Antarctic Ice Sheet, which stores enough ice to raise sea level by up to 58 m (Fretwell et al., 2013), is of vital importance for coastal communities ranging from small islands to the world's mega-cities, for ecosystems, and for the global economy (Clark et al., 2016).

The Antarctic Ice Sheet has experienced changing environmental conditions on various timescales, from decadal- to orbital-scale climate variability, since its presumed inception during the Eocene–Oligocene transition about 34 Myr ago (Zachos et al., 2001; DeConto and Pollard, 2003). This resulted in strong variations in its volume and extent linked to the slow multi-millennial changes in the Earth's astronomical configuration during the Early to Middle Miocene (Naish et al., 2001; Levy et al., 2016) and the Pliocene (Naish et al., 2009). While terrestrial parts of the East Antarctic Ice Sheet have persisted for millions of years (Sugden et al., 1995;

Shakun et al., 2018), ice-sheet variability involved an occasional collapse of the West Antarctic Ice Sheet (Naish et al., 2009) and inward migration of ice-sheet margins in marine-based sectors of East Antarctica (that is, where the ice sheet is grounded below sea level) during Pliocene warm periods (Cook et al., 2013; Patterson et al., 2014; Aitken et al., 2016). During Pleistocene interglacials, Antarctic ice loss from the East Antarctic Wilkes subglacial basin (Wilson et al., 2018; Blackburn et al., 2020) and across the Weddell Sea embayment (Turney et al., 2020) may have contributed to sea-level high stands of 6 to 9 m higher than present (including a contribution from thermal expansion and mass loss from the Greenland Ice Sheet; Dutton et al., 2015).

The future trajectory of the Antarctic Ice Sheet under progressing warming, however, is highly uncertain. This is due to uncertainties in the understanding and representation of ice-sheet processes and ice–climate interactions (Fox-Kemper et al., 2021) as well as the potentially high magnitudes and rates of recent and projected warming. The present rate of warming is unprecedented in at least 2000 years, with an increase of 1.1 °C in the global mean surface temperature between 1850–1900 and 2011–2020 (Gulev et al., 2021). The amount of warming projected for the end of this century under the Shared Socioeconomic Pathways (e.g. for the higher-emission scenario, SSP5-8.5, with an increase in global annual mean surface air temperature of 3.6 to 6.5 °C relative to 1850–1900; Lee et al., 2021) is comparable to the transition from the Last Glacial Maximum to the beginning of the Holocene approximately 11 700 years before present but is expected to develop on much shorter timescales.

At present, accelerated mass loss of the Antarctic Ice Sheet is concentrated in West Antarctica and the East Antarctic Wilkes Land (Rignot et al., 2019; Otosaka et al., 2023; Li et al., 2016; Miles et al., 2021), likely driven by ocean-induced melting due to the intrusion of warm water into the ice-shelf cavities (Paolo et al., 2015). In future projections so far, for instance in those provided by the recent Ice Sheet Model Intercomparison Project for CMIP6 (IS-MIP6; Seroussi et al., 2020; Payne et al., 2021), the transient sea-level response to the projected warming ranges from a slight mass gain to a mass loss of the Antarctic Ice Sheet by the end of this century under multiple emission scenarios, with the largest spread in sea-level change given by the higher-emission pathways RCP8.5 (Representative Concentration Pathway) and SSP5-8.5. The bulk of sea-level rise, however, is expected to unfold beyond the end of this century (Clark et al., 2016; Fox-Kemper et al., 2021) due to the inertia of the continental-scale ice sheet in combination with the potential crossing of critical thresholds with ongoing warming (Lenton et al., 2023). This long-term committed sea-level response, which has already been triggered or may be triggered during the next decades or centuries (but unfolds thereafter over multiple centuries to millennia), might be substantially higher than the transient realized sea-level change, but it is not represented in typical sea-level projections (Seroussi et al., 2020; Edwards et al., 2021). Here, we assess this expected long-term committed contribution to sea-level change from the Antarctic Ice Sheet by stabilizing the climatic boundary conditions projected over the next centuries at different points in time and by letting the ice sheet evolve over several millennia. We furthermore quantify the difference or offset between the transient realized sea-level contribution from Antarctica at a particular point in time and the respective long-term committed sea-level contribution.

The slow ice-sheet response to perturbations in its climatic boundary conditions owing to high inertia results in a time lag between the forcing and the resulting mass change. As the ice-sheet response unfolds on centennial-to-multi-millennial timescales, sea level may keep rising for millennia to come even if warming is stabilized (Golledge et al., 2015; Winkelmann et al., 2015). This is due in particular to the softening-induced increase in the creep component of the ice flow and to internal feedbacks (Clarke et al., 1977; Golledge et al., 2015).

In addition, the Antarctic Ice Sheet is subject to several amplifying (and dampening) feedbacks, determining its long-term stability (Fyke et al., 2018; Garbe et al., 2020). For example, accelerated ice loss may be related to the amplifying surface-melt–elevation feedback (Oerlemans, 1981; Levermann and Winkelmann, 2016). With the lowering of the ice-sheet surface due to melting, it is exposed to higher air temperatures. Surface melting is, in turn, enhanced, promoting persistent ice loss upon crossing a critical temperature threshold. Furthermore, the marine parts of the ice sheet, e.g. in West Antarctica or in the East Antarctic Aurora and Wilkes subglacial basins, have been found to be susceptible to self-sustained, potentially irreversible grounding-line retreat (Feldmann et al., 2014; Mengel and Levermann, 2014; Garbe et al., 2020; Rosier et al., 2021). The rapid grounding-line retreat is often associated with amplifying feedbacks, in which the increased ice flow across the grounding line caused by an initial retreat fosters further retreat. In a theoretical flowline setup, it was shown that due to the ice flux being a nonlinear function of the ice thickness, ice sheets grounded below sea level on a retrograde inland sloping bed are unstable (Marine Ice Sheet Instability; Weertman, 1974; Schoof, 2007). More complex stability conditions arise in three dimensions when accounting for additional processes such as buttressing (Gudmundsson et al., 2012; Haseloff and Sergienko, 2018; Pegler, 2018), calving, and submarine melting (Haseloff and Sergienko, 2022) or the presence of feedbacks between the ice sheet and its environment (Sergienko, 2022). Ice loss may be dampened, on the other hand, by negative feedbacks such as introduced by, for example, the isostatic rebound of the solid Earth underlying the ice sheet, which could potentially stabilize West Antarctic grounding lines (Barletta et al., 2018; Coulon et al., 2021).

Depending on the interplay in these feedbacks, persistent mass loss may be triggered once critical forcings or tipping points (Lenton et al., 2008; Armstrong McKay et al., 2022),

for instance in temperature, are crossed. The Antarctic Ice Sheet was therefore classified as a tipping element of the climate system (Lenton et al., 2008; Armstrong McKay et al., 2022; Lenton et al., 2023). Due to the inertia of ice sheets and the related delay in their transient response following a realistic warming trajectory under, for example, a higher-emission pathway, the ice-sheet volume trajectory likely deviates from the ice-sheet equilibrium response to warming (Garbe et al., 2020; Rosier et al., 2021). While the consequences of self-sustained ice loss potentially triggered in the next decades or centuries may play out and become visible over millennial timescales (Reese et al., 2023; characterized as a slow onset of tipping; Ritchie et al., 2021), tipping may also be sped up by forcing beyond the critical threshold. For the Greenland Ice Sheet, it has been shown that the timescales of ice-sheet decline strongly depend on how far its critical temperature threshold is exceeded (Robinson et al., 2012).

Previous assessments of the long-term contribution to sea-level rise from the Antarctic Ice Sheet have been primarily restricted to a single ice-sheet model and have thus rarely explored model uncertainties, including uncertainties in ice-sheet processes, their parameterizations in ice-sheet models and distinct initialization approaches, as well as uncertainties in the future climate (climate forcing uncertainty; Golledge et al., 2015; Clark et al., 2016). They suggest that the grounding lines in the Amundsen Sea embayment might at present already be undergoing a self-sustained retreat until a new stable geometric configuration is reached or that this retreat might be imminent under sustained present-day climate (Favier et al., 2014; Joughin et al., 2014; Arthern and Williams, 2017; Seroussi et al., 2017; Golledge et al., 2019, 2021; Reese et al., 2023). The potential for pronounced grounding-line recession in marine-based portions of West Antarctica and the East Antarctic Wilkes subglacial basin until the end of the millennium has been illustrated for higher-emission scenarios (Golledge et al., 2015; Winkelmann et al., 2015; Clark et al., 2016; Bulthuis et al., 2019; Chambers et al., 2022; Coulon et al., 2024), giving rise to an Antarctic mass loss of multiple metres of sea-level equivalent. On multi-millennial timescales, the loss of the portion of the ice sheet grounded above sea level in East Antarctica may be locked in for strong atmospheric warming, which would eventually commit the Antarctic Ice Sheet to contributing several tens of metres to sea-level rise (Winkelmann et al., 2015; Clark et al., 2016).

Here we systematically study the long-term multi-millennial evolution of the Antarctic Ice Sheet in response to a wide range of possible future climate trajectories and thereby quantify its sea-level commitment for stabilized climates at different points in time over the course of the next centuries, taking into account uncertainties in future Antarctic climate and ice-sheet processes, by means of two different ice-sheet models: the Parallel Ice Sheet Model (PISM; Bueler and Brown, 2009; Winkelmann et al., 2011) and Kori-ULB previously called f.ETISh; Pattyn, 2017). The remain-

der of this paper is structured as follows: in the following section, Sect. 2, we describe the methods for projecting Antarctica's sea-level contribution on multi-millennial timescales. Results are presented in Sect. 3 and discussed in Sect. 4 with a focus on different sources of uncertainty arising from the divergence of future climate trajectories (climate forcing uncertainty) as well as from ice-sheet processes, their parameterizations in ice-sheet models and related parameter choices next to distinct initialization approaches (model uncertainties).

## 2 Methods

### 2.1 Ice-sheet models

#### 2.1.1 PISM

The Parallel Ice Sheet Model (PISM; Bueler and Brown, 2009; Winkelmann et al., 2011) is an open-source, thermo-mechanically coupled ice-sheet–stream–shelf model. In hybrid mode, the shallow ice approximation (SIA) and shallow shelf approximation (SSA) are solved and superimposed, giving rise to different dynamic regimes from the slow-flowing ice in the ice-sheet interior to the faster-flowing streams and ice shelves. We use a modified version of PISM release v1.0 (Garbe et al., 2020). In particular, centred differences of the ice thickness across the grounding line are calculated to derive the surface gradient, which have been shown to improve the representation of the driving stress at the grounding line (Reese et al., 2023). We use a rectangular grid with a horizontal resolution of 16 km and a vertical grid structure with the highest resolution at the base of the ice sheet and shelves.

Basal shear stress $\tau_b$ and shallow shelf approximation basal sliding velocities $u_b$ are related in a general power law of the form

$$\tau_b = -\tau_c \frac{u_b}{u_{th}^q |u_b|^{1-q}}, \tag{1}$$

with the threshold velocity $u_{th} = 100 \, \text{m yr}^{-1}$ and the sliding exponent $q$. The yield stress $\tau_c$ is determined by the Mohr–Coulomb failure criterion (Cuffey and Paterson, 2010) as

$$\tau_c = \tan(\phi) N_{till}, \tag{2}$$

including the till friction angle $\phi$ and the effective pressure $N_{till}$. The till friction angle is parameterized as a piecewise linear function of bed elevation (Martin et al., 2011) in our simulations, with a lower value of 24° for topography below $-700 \, \text{m}$ and an upper value of 30° for topography above 500 m (following Reese et al., 2023). The effective pressure $N_{till}$ in PISM is a function of the overburden pressure $P_0$ and the fraction of the effective water thickness in the till layer $s = W_{till}/W_{max}$:

$$N_{till} = \min \left\{ P_0, N_0 \left( \frac{\delta P_0}{N_0} \right)^s 10^{(e_0/C_c)(1-s)} \right\}, \tag{3}$$

with the till effective overburden fraction $\delta$ and the constants $N_0$, $e_0$ and $C_c$ (where values are chosen following Bueler and van Pelt, 2015). The amount of water from basal melt in the till layer $W_{till}$, with a maximum of $W_{max} = 2$ m, evolves according to the non-conserving "null" hydrology model (as described in Bueler and van Pelt, 2015), with a decay rate $C$ of water in the till. The grounding-line position is simulated at a sub-grid scale, evolving freely without imposing additional flux conditions. Basal resistance is linearly interpolated on a sub-grid scale around the grounding line (Feldmann et al., 2014), while sub-shelf melt in partially floating cells is not applied in the simulations presented here. We apply eigencalving, which linearly relates the calving rate to the spreading rate tensor with a proportionality factor $K = 1 \times 10^{17}$ m s (Levermann et al., 2012). Additionally, thin ice below 50 m is removed at the calving front (thickness calving), and a maximum extent for ice shelves is defined (Albrecht et al., 2020; Garbe et al., 2020). Our simulations include the effect of the viscous and elastic response of the bedrock to changes in ice load following Lingle and Clark (1985) and Bueler et al. (2007), with an upper-mantle viscosity $\eta = 1 \times 10^{21}$ Pa s and density $\rho = 3300$ kg m$^{-3}$ as well as a flexural rigidity of the lithosphere of $5 \times 10^{24}$ N m.

### 2.1.2 Kori-ULB

The Kori-ULB ice flow model, which is the follow-up of the f.ETISh model (Pattyn, 2017), is a vertically integrated, thermomechanically coupled hybrid ice-sheet–ice-shelf model that incorporates relevant features for studying the evolution of the Antarctic Ice Sheet such as the surface melt–elevation feedback, basal sliding, sub-shelf melting, calving and bedrock deformation. The ice flow is governed by a combination of the shallow ice approximation (SIA) and shallow shelf approximation (SSA) for grounded ice and by the shallow shelf approximation for floating ice shelves (Bueler and Brown, 2009; Winkelmann et al., 2011). Simulations of the multi-millennial Antarctic sea-level contribution presented here were performed using Kori-ULB version 0.91 (Coulon et al., 2024) at a horizontal resolution of 16 km.

Basal sliding is parameterized using a Weertman sliding law, i.e.

$$\boldsymbol{\tau}_b = A_b^{-1/m} |\boldsymbol{v}_b|^{1/m-1} \boldsymbol{v}_b, \tag{4}$$

where $\boldsymbol{\tau}_b$ and $\boldsymbol{v}_b$ are the basal shear stress and the basal velocity, respectively, and with a basal sliding exponent $m = 3$. The values of the basal sliding coefficients $A_b$ are inferred following the nudging method of Pollard and DeConto (2012b) and Bernales et al. (2017, see Sect. 2.2.2).

At the grounding line, a flux condition (related to the ice thickness at the grounding line; Schoof, 2007) is imposed as in Pollard and DeConto (2012a) and Pollard and DeConto (2020) to account for grounding-line migration. The implementation can reproduce the steady-state behaviour of the grounding line and its migration (Schoof, 2007) at a coarse resolution (Pattyn et al., 2013). Using this flux condition, the marine ice-sheet behaviour in Antarctica was simulated using large-scale ice-sheet models (Pollard and DeConto, 2012a; DeConto and Pollard, 2016; Pattyn, 2017; Sun et al., 2020) with similar results under buttressed conditions as in high-resolution models (Pollard and DeConto, 2020). Calving at the ice front depends on the parameterized combined penetration depths of surface and basal crevasses relative to the total ice thickness. Similar to Pollard et al. (2015) and DeConto and Pollard (2016), the parameterization of the crevasse penetration depths involves the divergence of the ice velocity, the accumulated strain and the ice thickness. Bedrock adjustment in response to changes in ice and ocean load is taken into account by means of the commonly used Elastic Lithosphere-Relaxed Asthenosphere (ELRA) model, where the solid-Earth System is represented by a relaxed viscous asthenosphere below a thin elastic lithosphere plate (Le Meur and Huybrechts, 1996; Coulon et al., 2021). A spatially uniform relaxation time of 3000 years and a flexural rigidity of the lithosphere of $10^{25}$ N m were chosen for the simulations presented here.

## 2.2 Experimental design

### 2.2.1 Assessing Antarctic sea-level commitment

We aim to determine the long-term multi-millennial sea-level contribution from the Antarctic Ice Sheet (and thereby its sea-level commitment) using the ice-sheet models PISM and Kori-ULB (compare to Sect. 2.1). After initializing the ice-sheet models to obtain initial Antarctic Ice Sheet states and running historical simulations (described in more detail in the following sections, Sect. 2.2.2 and 2.2.3, and illustrated in Fig. 1), we assess the response of the Antarctic Ice Sheet to changes in the atmospheric and oceanic boundary conditions derived from state-of-the-art climate model projections available from the sixth phase of the Coupled Model Intercomparison Project (CMIP6) under the Shared Socioeconomic Pathways SSP1-2.6 and SSP5-8.5 (thus covering a wide range from lower- to higher-emission scenarios). As we are interested in the long-term ice-sheet response, we focus on a set of CMIP6 general circulation models (GCMs) that provide forcing through the year 2300. Using these, we obtain the transient realized sea-level contribution from the Antarctic Ice Sheet at given points in time (for instance, in the year 2100; filled dots in Fig. 1). We then quantify the long-term committed sea-level contribution by stabilizing forcing conditions of the climate trajectories at regular intervals in time ("branching off" in the years 2050, 2100, 2150, 2200, 2250 and 2300; compare to Table S1 in the Supplement) and by letting the ice sheet evolve over several millennia (through the year 7000) under constant climatic boundary conditions characteristic of the respective branch-off year (Fig. 1, open dots). Climate conditions of the distinct branch-off years are

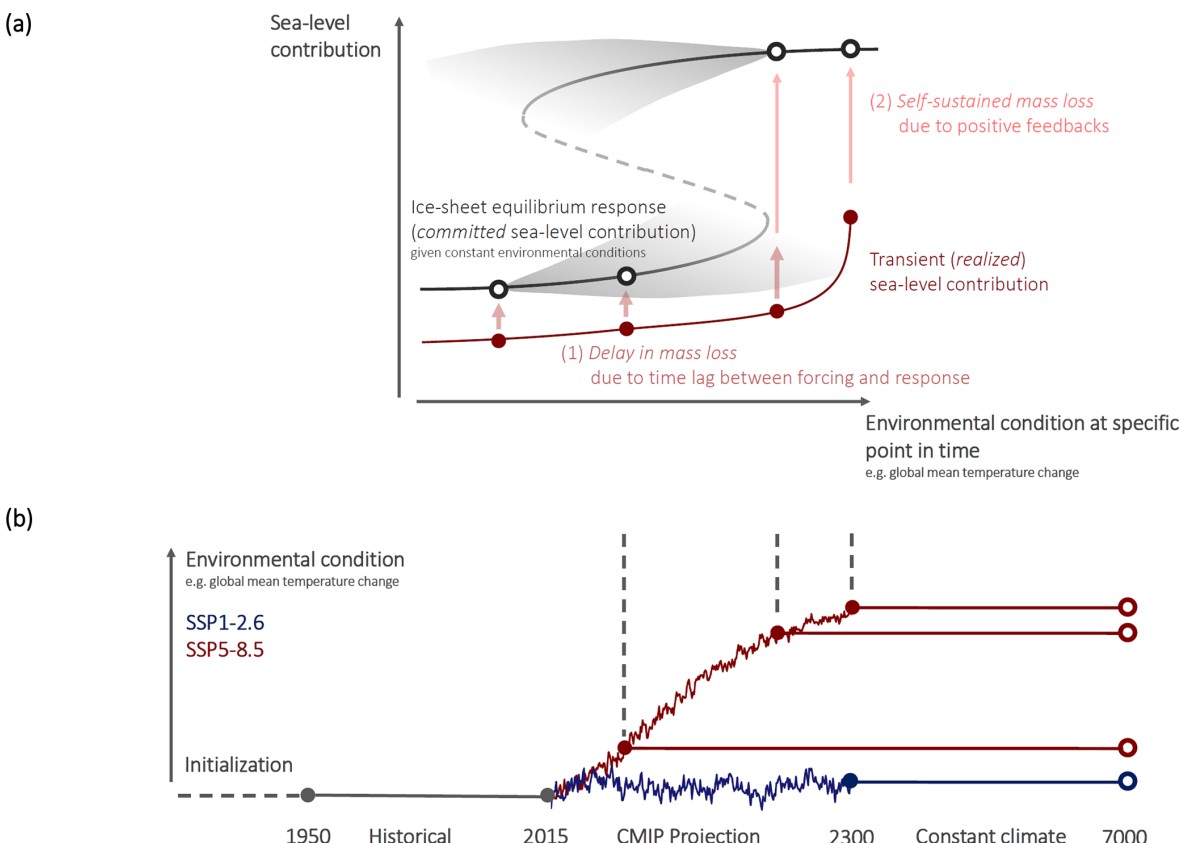

**Figure 1.** Schematic of the experimental design. **(a)** Idealized and simplified stability diagram of the Antarctic Ice Sheet as a possible tipping element, which illustrates some underlying factors potentially contributing to the substantial difference or offset between the transient realized and long-term committed ice-sheet responses to progressing warming (in terms of sea-level contribution). For example, the crossing of critical thresholds with ongoing warming may result in accelerated mass loss. This is associated with the stepwise change (jump) towards a higher sea-level contribution indicated as (2). **(b)** Schematic summary of the experimental design used for assessing the long-term committed contribution from the Antarctic Ice Sheet to sea-level rise using the ice-sheet models Kori-ULB and PISM. Starting with the initialization of the ice-sheet models to build ice-sheet states associated with the year 1950, historical simulations are run until the year 2015 (present day). Using potential future climate trajectories based on CMIP6 GCMs under emission pathways SSP1-2.6 (blue) and SSP5-8.5 (red), the transient realized Antarctic sea-level contribution is projected until the year 2300. Additional simulations branching off at regular intervals in time determine the Antarctic sea-level commitment under stabilized climatic boundary conditions sustained over several millennia.

determined as the mean over the previous 10 years before the branch-off.

### 2.2.2 Initialization

Both ice-sheet models are initialized using constant histori-
5 cal climatic boundary conditions representing the year 1950
(Fig. 1). The historical climatic boundary conditions are con-
structed using the historical changes in the atmosphere and
ocean with respect to the reference period from 1995 to
2014 from the Norwegian Earth System Model (NorESM1-
10 M; Bentsen et al., 2013) in CMIP5. The atmospheric and
oceanic anomalies from NorESM1-M are averaged over the
period of 1945–1955 and subsequently added to present-day
atmospheric temperatures and precipitation derived from re-
gional climate models (RCMs) as well as observed present-
15 day ocean temperatures and salinities. Present-day atmo-

spheric climatologies are derived from the RCMs Modèle
Atmosphérique Régional (MARv3.11; Kittel et al., 2021)
and Regional Atmospheric Climate MOdel (RACMO2.3p2;
van Wessem et al., 2018) to take into account uncertain-
ties in the representation of present-day Antarctic surface 20
climate (compare to Mottram et al., 2021). While a recent
intercomparison concluded that Antarctic climate is repre-
sented reasonably well compared to observations in state-
of-the-art RCMs, disagreement between the RCMs with re-
spect to surface mass balance components (such as precipita- 25
tion and atmospheric temperatures as applied to the ice-sheet
models here) exists for some areas (Mottram et al., 2021).
Both present-day atmospheric climatologies are involved in
the initialization of each ice-sheet model, resulting in four
(initial) ice-sheet model configurations (Table S2 in the Sup- 30

plement). For ocean temperatures and salinities, present-day observations based on Schmidtko et al. (2014) are used.

To build initial ice-sheet states with PISM, a spin-up approach is applied for each of the historical atmospheric climatologies (around the year 1950; see above) individually. Uncertainties in ice-sheet model parameters are taken into account by running an ensemble of spin-up simulations and choosing the initial ice-sheet state which fits well with observations of present-day ice thickness, ice velocities and grounding-line position. More specifically, starting from Bedmap2 ice thickness and topography (Fretwell et al., 2013), PISM is run for 600 000 years with constant geometry to obtain a thermodynamic equilibrium. Applying the constant historical atmospheric and oceanic boundary conditions associated with the year 1950, an ensemble of simulations with varying ice-sheet model parameters (guided by recent PISM ensembles; Albrecht et al., 2020; Reese et al., 2023) is run for several thousand years. Here, we include the SIA enhancement factor ($E_{\mathrm{SIA}} \in 1.5, 2$; varied around the reference value of Albrecht et al., 2020) and parameters related to basal sliding, namely the sliding exponent ($q \in 0.25, 0.5, 0.75$; within the range investigated by Albrecht et al., 2020; Lowry et al., 2021), the till effective overburden fraction ($\delta \in 1.5, 2.0, 2.5$; Bueler and van Pelt, 2015) and the decay rate of till water content ($C \in 7, 10\,\mathrm{mm\,yr^{-1}}$; equivalent to the range explored by Albrecht et al., 2020). After 5000 years of simulation, the ensemble members are assessed using a scoring method following Albrecht et al. (2020) and Reese et al. (2020). The scoring method is based on the mean-square-error mismatch of grounded and floating ice area, ice thickness, grounding-line location, and surface velocity compared to present-day observations (Rignot et al., 2011; Fretwell et al., 2013). Each indicator is evaluated for the entire Antarctic domain as well as for the Amundsen, Ronne–Filchner and Ross regions individually. The five best-scoring ensemble members in terms of the continental- and basin-scale indicators are continued until reaching 50 000 years of simulation, and simulations of the multi-millennial Antarctic sea-level contribution are based on the ice-sheet configuration performing well in the scoring after 50 000 years. Respective ice-sheet model parameters are given in Table S2.

For Kori-ULB, initial ice-sheet states and basal sliding coefficients are obtained in an inverse simulation following Pollard and DeConto (2012b) for each of the historical atmospheric climatologies associated with the year 1950 (as described above). In this inverse simulation, the differences from the observed present-day ice thickness (Bedmachine; Morlighem et al., 2020) are minimized by iteratively adjusting the basal sliding coefficients under grounded ice and the sub-shelf melt rates under floating ice (Bernales et al., 2017). The optimized field of basal sliding coefficients in Kori-ULB is characterized by high basal sliding coefficients at the ice-sheet margins turning into regions of low slipperiness (low basal sliding coefficients) towards the interior of West Antarctica. It thus differs from the basal friction experienced by the Antarctic Ice Sheet in simulations with PISM presented here, where overall slippery bed conditions in the interior of marine subglacial basins are found, given the parameterized, bed-elevation-dependent material properties of the subglacial till (in particular, the till friction angle; Sect. 2.1). These inter-model differences in basal friction linked to the applied initialization approaches are expected to influence the simulated ice-sheet response to changes in Antarctic climate. The resulting sub-shelf melt rates may be interpreted as balanced sub-shelf melt rates and are independent of the oceanic boundary conditions (forcing), while the ice-sheet states are in steady-state with the constant historical atmospheric climatologies. Applying constant historical atmospheric and oceanic boundary conditions associated with the year 1950, a short relaxation is run for 10 years after the ice-sheet model initialization and before the historical simulation. This limits the initial shock that may result from the transition from the balanced sub-shelf melt rates derived during the inverse simulation to the imposed sub-shelf melt parameterization scheme. The two initial ice-sheet states resulting from the inverse simulation are therefore in quasi-equilibrium for historical climatic boundary conditions.

Given that we include two common ways of initializing ice-sheet models (compare to, e.g. Seroussi et al., 2019, 2020), we sample uncertainties associated with the choice of the initialization approach. While an inverse simulation allows us to reproduce the observed present-day ice-sheet geometry well, the resulting parameter fields (such as the basal sliding coefficients in Kori-ULB) may compensate for errors or uncertainties in other ice-sheet processes (Aschwanden et al., 2013; Berends et al., 2023b). In addition, it is assumed that the parameter fields obtained in the inverse simulation to match the observed present-day ice thickness do not change in the future CE2. In contrast, in the simulated ice-sheet state resulting from a spin-up, ice-sheet variables may be modelled in a consistent way, but its geometry might differ from the observed ice sheet. It is the result of the covered ice-sheet physics in the ice-sheet model for a set of uncertain parameters without any nudging.

The simulated grounding-line position and ice thickness of the initial ice-sheet states are compared to present-day observations in Fig. S1 in the Supplement. As a result of the inverse simulation, the grounding-line position and ice thickness compare well to present-day observations in the initial ice-sheet states for Kori-ULB (Fig. S1a and c). With the spin-up approach applied in PISM, the initial ice-sheet states are characterized by larger ice-thickness differences compared to present-day observations (Fig. S1b and d). Overall, ice in West Antarctica and in some coastal regions in East Antarctica (e.g. in Queen Maud Land, upstream of Amery Ice Shelf and in Wilkes Land) is thicker than observed at present (comparable to Reese et al., 2023), while the ice thickness in the interior of East Antarctica is underestimated. In addition, the grounding line in the Siple Coast area (and in the catchment

draining Ronne–Filchner Ice Shelf for the MAR atmospheric climatology) is located upstream of the observed grounding line in the present day (Fig. S1b and d), as previously seen in ice-sheet model initializations using a spin-up approach (e.g. Reese et al., 2023; Sutter et al., 2023). These differences should be taken into account when interpreting the simulated long-term evolution of the Antarctic Ice Sheet.

### 2.2.3 Forcing and boundary conditions over the historical period and until 2300

Starting from the initial ice-sheet states and climate conditions associated with the year 1950 described above, we run historical simulations for the time period from 1950 to 2015 (Fig. 1). Changes in atmospheric and oceanic conditions over the historical period are derived from NorESM1-M, as recommended by ISMIP6 (Barthel et al., 2020; Nowicki et al., 2020). The atmospheric and oceanic forcing conditions through the year 2300, which are applied to the ice-sheet models to study the future evolution of the Antarctic Ice Sheet, rely on a subset of state-of-the-art GCM projections available within CMIP6 (MRI-ESM2-0, UKESM1-0-LL, CESM2-WACCM and IPSL-CM6A-LR). We apply climate forcing under the Shared Socioeconomic Pathways SSP1-2.6 and SSP5-8.5. The selection of CMIP6 GCMs was guided by the limited availability of extended climate projections through the year 2300 within ScenarioMIP (O'Neill et al., 2016), while the climate sensitivity of the available GCMs (Meehl et al., 2020) and their performance in comparison to observations (e.g. Beadling et al., 2020; Bracegirdle et al., 2020; Purich and England, 2021) were considered secondary criteria.

Following Nowicki et al. (2020), spatially varying atmospheric (near-surface air temperature) and oceanic (salinity and temperature) anomalies with respect to the 1995–2014 mean climatology are derived from NorESM1-M over the historical period as well as from projections of the selected CMIP6 GCMs through the year 2300. Note that for precipitation, ratios (instead of anomalies) with respect to the 1995–2014 mean precipitation are determined to avoid "negative" absolute precipitation (e.g. Goosse et al., 2010, Eq. 30). The respective anomalies are added to the constant present-day climatologies for the atmosphere (MAR and RACMO) and for the ocean (Schmidtko et al., 2014). Thus, the resulting forcing matches present-day climate conditions in the 1995–2014 reference period (as in Reese et al., 2023). For the ocean properties, yearly averaged forcing is applied to the ice-sheet models. Missing values for the oceanic forcing on the continental shelf (arising due to the coarse resolution of CMIP6 GCMs) and in currently ice-covered regions are filled following Kreuzer et al. (2021), i.e. by averaging over all existing values in neighbouring cells. In addition, the ocean properties derived from CMIP6 GCMs are linearly interpolated to the basin-averaged continental shelf depth (Kreuzer et al., 2021) to determine sub-shelf melt (see below). Monthly forcing is

used at the interface of the ice sheet and the atmosphere (as in Golledge et al., 2019).

Surface melt and runoff are determined from monthly atmospheric temperature and precipitation (i.e. accounting for the seasonal cycle) using a positive-degree-day (PDD) scheme (Reeh, 1991). The number of positive degree days is calculated following the approach presented in Calov and Greve (2005), with a default value for the standard deviation of $\sigma = 5\,°C$ and $\sigma = 4\,°C$ for PISM and Kori-ULB, respectively. Snow accumulation rates are derived from precipitation via an atmospheric temperature threshold with a linear transition between snow and rain. In Kori-ULB, natural variability is considered when determining snow accumulation rates (similar to the calculation of the number of positive degree days) using a standard deviation of $\sigma = 3.5\,°C$. Melt coefficients of 3 mm ofwaterequivalent per PDD for snow and 8 mm ofwaterequivalent per PDD for ice are used in both ice-sheet models following a comparison with MAR estimates through the year 2100 (Kittel et al., 2021; Coulon et al., 2024). A constant fraction of surface melt refreezes in PISM (Reeh, 1991), while Kori-ULB applies a simple thermodynamic parameterization of the refreezing process (Huybrechts and De Wolde, 1999; Coulon et al., 2024). Computing surface melt and runoff in a PDD approach follows De-Conto et al. (2021) and Garbe et al. (2020), who are also studying the future Antarctic Ice Sheet response to a changing climate and the ice-sheet stability on multi-millennial timescales. Applying the surface mass balance determined by RCMs, which are in turn forced by GCM climate projections, would be an alternative approach to project the Antarctic sea-level contribution. However, surface mass balance estimates from RCMs are not yet available beyond the end of this century and may be biased by the use of a static ice-sheet geometry neglecting, for example, the surface melt–elevation feedback (Kittel et al., 2021).

To account for the surface melt–elevation feedback, the near-surface air temperature is corrected for changes in the ice-sheet surface elevation. More specifically, the air temperatures $T_{forcing}$ provided to the ice-sheet models are shifted linearly with a change in surface elevation $\Delta h$ as

$$T = T_{forcing} + \Gamma \Delta h, \tag{5}$$

following the atmospheric lapse rate $\Gamma$ of $8\,°C\,km^{-1}$.

Sub-shelf melt rates are computed using the Potsdam Ice-shelf Cavity mOdel (PICO; Reese et al., 2018). PICO calculates sub-shelf melt rates from far-field salinities and temperatures and parameterizes the vertical overturning circulation in ice-shelf cavities. The values of the PICO overturning strength parameter $C$ and the turbulent heat exchange coefficient $\gamma_T^*$ are an individual choice for each ice-sheet model to match sub-shelf melt sensitivities and/or observed melt rates. $C = 3 \times 10^6\,m^6\,s^{-1}\,kg^{-1}$ and $\gamma_T^* = 7 \times 10^{-5}\,m\,s^{-1}$ (with correction of the ocean properties from Schmidtko et al., 2014 to match the observed present-day melt rates from Adusumilli et al., 2020) are

used for PISM simulations, as they have been found to fit melt sensitivities well (Reese et al., 2023). For Kori-ULB simulations, the overturning strength and the turbulent heat exchange coefficient are $C = 1 \times 10^6\,\mathrm{m^6\,s^{-1}\,kg^{-1}}$ and

5 $\gamma_\mathrm{T}^* = 4 \times 10^{-5}\,\mathrm{m\,s^{-1}}$, respectively.

## 3   Results

We here present the transient response of the Antarctic Ice Sheet over the historical period (Sect. 3.1) and to a range of possible future climate trajectories through 2300 (Sect. 3.2),

along with the associated committed ice-sheet evolution on multi-millennial timescales (Sect. 3.3). The dependency of the committed Antarctic sea-level contribution on the climate conditions in Antarctica sustained over several thousands of years after their stabilization at different points in time during

the next centuries is assessed in Sect. 3.4.

### 3.1   Historical ice-sheet evolution

The pattern of observed present-day rates of ice-thickness change (e.g. Smith et al., 2020) is captured overall by both ice-sheet models in response to the historical NorESM1-M

climate trajectory from 1950 to 2015 (Fig. 2a–c), with thinning in the Amundsen and Bellingshausen seas regions and the Antarctic Peninsula and thickening in the ice-sheet interior. The magnitude of ice-sheet thinning in the Amundsen Sea embayment is, however, underestimated compared

to present-day observations in the historical simulations with PISM presented here (Fig. 2a and c). In addition, we find ice loss for the Ross, Ronne-Filchner and Amery ice shelves in PISM, in contrast to present-day observations (Fig. 2a and c).

The evolution of the continent-wide integrated surface

mass balance is relatively similar for both ice-sheet models but occurs on a higher (although still within RCM uncertainties) level in PISM than in Kori-ULB (Fig. 2d). While sub-shelf melt increases in PISM from about $300\,\mathrm{Gt\,yr^{-1}}$ in 1950 to approximately $1100\,\mathrm{Gt\,yr^{-1}}$ in 2015 at the lower

end of present-day observations (Fig. 2e, solid lines), at about $1800\,\mathrm{Gt\,yr^{-1}}$, the basal mass balance is on the order of the observational record in Kori-ULB over the entire historical period, slightly exceeding its upper end in 2015 Fig. 2e, dashed lines). The continent-wide aggregated

sub-shelf melt rates observed in the present day are thus reproduced by both sets of PICO parameters (compare to Sect. 2.2.3), but they result in different sensitivities of sub-shelf melt rates to ocean temperature changes over the historical period (Fig. 2e; Reese et al., 2023).

Mass loss in the Amundsen Sea embayment dominates the overall observed ice-sheet mass changes in Antarctica to date (Otosaka et al., 2023). Given the lower magnitude of ice-sheet thinning of the Pine Island and Thwaites glaciers in PISM and stronger sub-shelf melt in Kori-ULB, we find

diverging ice-sheet trajectories between both ice-sheet mod-

els in terms of the Antarctic sea-level contribution over the historical period from 1950 to 2015: Kori-ULB shows an integrated mass loss with a sea-level contribution of about $+4\,\mathrm{mm}$ in 2015 (Fig. 2f, dashed lines), while overall the ice sheet gains mass equivalent to a sea-level change ranging be- 55 tween $-4$ and $-6\,\mathrm{mm}$ in PISM (Fig. 2f, solid lines, within the spread of a recent ensemble of historical ice-sheet trajectories; Reese et al., 2023).

In the future evolution of the Antarctic Ice Sheet determined by PISM (Sect. 3.2–3.4), changes in the regions of 60 the Ross and Ronne–Filchner ice shelves could thus be overestimated, while the lower thinning rates over the historical period in the Amundsen Sea embayment could suggest a reduced sensitivity of the Thwaites and Pine Island glaciers to changes in Antarctic climate in these simulations. 65

### 3.2   Future transient ice-sheet response until 2300

By the end of this century, the projected Antarctic sea-level contribution varies between $-5.0$ to $+8.0\,\mathrm{cm}$ and $-6.0$ to $+6.0\,\mathrm{cm}$ compared to the present day in response to SSP1-2.6 and SSP5-8.5 climate trajectories, respectively (Fig. 3a and c, 70 Table 1). Projected ice-sheet changes by 2100 are comparable across emission pathways (Fig. 3; in line with Edwards et al., 2021; Lowry et al., 2021; Coulon et al., 2024), given a very similar evolution of Antarctic climate at least during the first half of the 21st century (Fig. 4). The sea-level con- 75 tribution due to mass balance changes of the Antarctic Ice Sheet projected by Kori-ULB and PISM by the end of this century is also within the range of recent estimates by IS-MIP6, ranging from $-9$ to $+30\,\mathrm{cm}$ under higher-emission pathways and from $-1.4$ to $+15.5\,\mathrm{cm}$ under lower-emission 80 pathways (Seroussi et al., 2020; Payne et al., 2021). The simulated trends of ice-sheet changes over the historical period are continued in these projections over this century in both ice-sheet models. That is, Kori-ULB projects a positive sea-level contribution from Antarctica compared to the present 85 day (Fig. 3a and c, dashed lines). PISM projects a sea-level drop by 2100 (Fig. 3a and c, solid lines).

Following the lower-emission pathway, SSP1-2.6, through 2300 results in an Antarctic mass change ranging from $-0.2$ to $+0.5\,\mathrm{m}$ of sea-level equivalent compared to the 90 present day (Fig. 3a, Table 1). In this range of Antarctic mass changes, Kori-ULB projects a steadily increasing Antarctic contribution to sea-level rise (Fig. 3a, dashed lines). While some PISM ice-sheet trajectories show the onset of mass loss after an initial mass gain (e.g. for CESM2-WACCM climate, 95 indicated in blue), the Antarctic sea-level contribution projected by PISM in 2300 compared to the present day remains negative (Fig. 3a, solid lines).

The overall sign of ice-sheet mass changes contributing to a change in sea level depends on the balance between 100 the dynamic response to sub-shelf melting and ice-shelf thinning and the surface mass balance, driven by changes in Antarctic climate (Fig. 4). We find that with projected

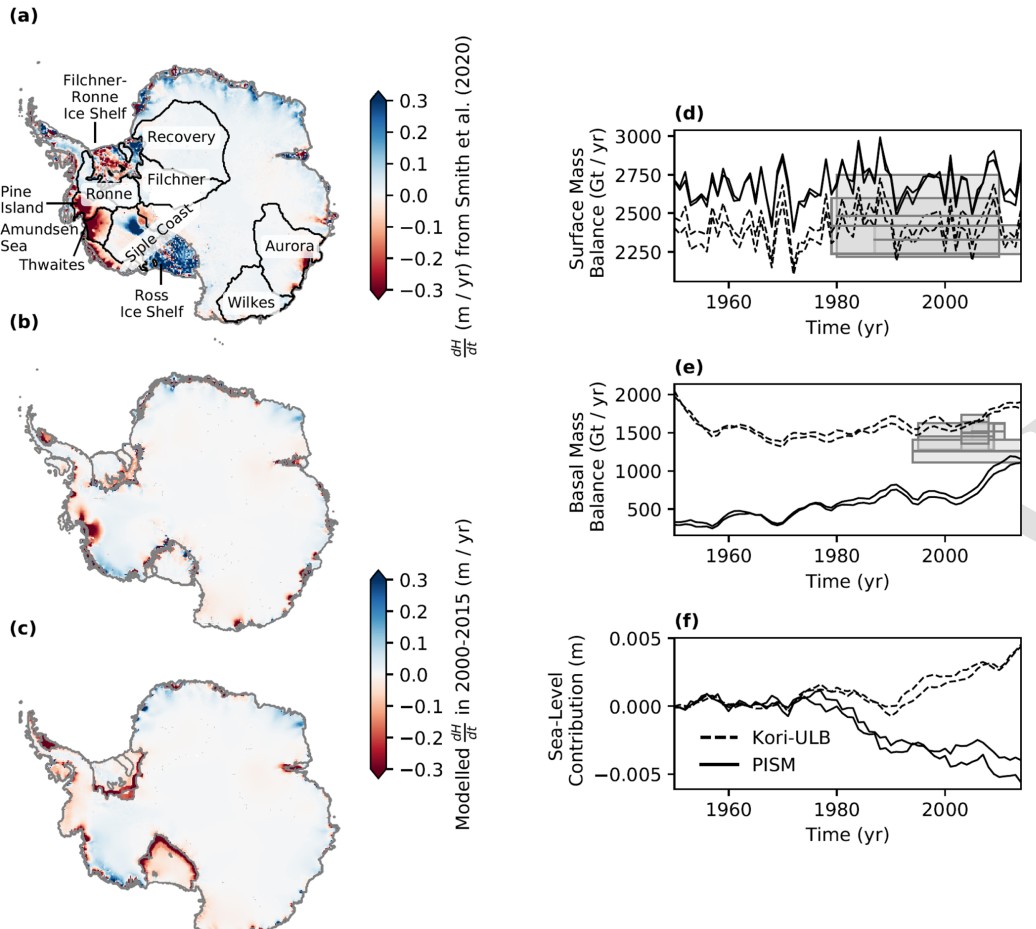

**Figure 2.** Trajectories of the Antarctic Ice Sheet over the historical period from 1950 to 2015. **(a–c)** Rates of observed ice-thickness change, based on Smith et al. (2020, **a**) and as determined by the ice-sheet models Kori-ULB **(b)** and PISM **(c)** in response to historical changes in Antarctic climate derived from NorESM1-M. The modelled ice-thickness change is averaged across the ice-sheet model configurations associated with atmospheric climatologies based on MAR and RACMO (Table S2). **(d–f)** Evolution of the Antarctic Ice Sheet as determined by the ice-sheet models Kori-ULB (dashed lines) and PISM (solid lines) using atmospheric climatologies based on MAR and RACMO in terms of the surface mass balance **(d)**, sub-shelf melt **(e)** and sea-level contribution **(f)** based on volume above floatation. Observations of the ice-sheet mass balance components (as in Coulon et al., 2024) are given by grey boxes (indicating the time period and uncertainties in the respective observations), where the solid line shows the mean.

Antarctic-averaged atmospheric warming of up to 3.6 °C in 2300 (Fig. 4a, Table S1), the integrated surface mass balance remains positive and at its present-day level until 2300 for both ice-sheet models under this lower-emission pathway, with strong GCM-dependent variability (Fig. 4b). The evolution of sub-shelf melt through 2300 is overall consistent across Kori-ULB and PISM and follows the characteristics of the projected changes in circum-Antarctic ocean temperatures by each GCM (Fig. 4d and e): abrupt circum-Antarctic ocean warming through 2050 in UKESM-1-0-LL and IPSL-CM6A-LR (of about 0.5 and 0.3 °C, respectively) results in a strong initial increase in sub-shelf melt (Fig. 4d and e, grey and pink). In contrast, a steady rise in CESM2-WACCM circum-Antarctic ocean temperatures of 0.7 °C by 2300 is accompanied by a continuous increase in sub-shelf melt (Fig. 4d and e, blue). Overall, the magnitude of sub-shelf melt is higher for projections by Kori-ULB (dashed lines) compared to PISM (solid lines), following the respective levels reached at the end of the historical period (Figs. 4e and 2e). The response in dynamic discharge contributing to a sea-level increase is thus stronger in Kori-ULB than in PISM in the Amundsen Sea embayment and the East Antarctic Totten Glacier (Fig. 3b), explaining the diverging sea-level contribution under SSP1-2.6 until 2300.

With significant changes in Antarctic climate from 2100 onwards under the higher-emission pathway, SSP5-8.5 (Fig. 4a and d), Antarctica's sea-level contribution increases to +0.7–+3.1 m by 2300 (Fig. 3c, Table 1). The transient contribution of the Antarctic Ice Sheet to sea-level rise through 2300 is in line with the results of Golledge et al.

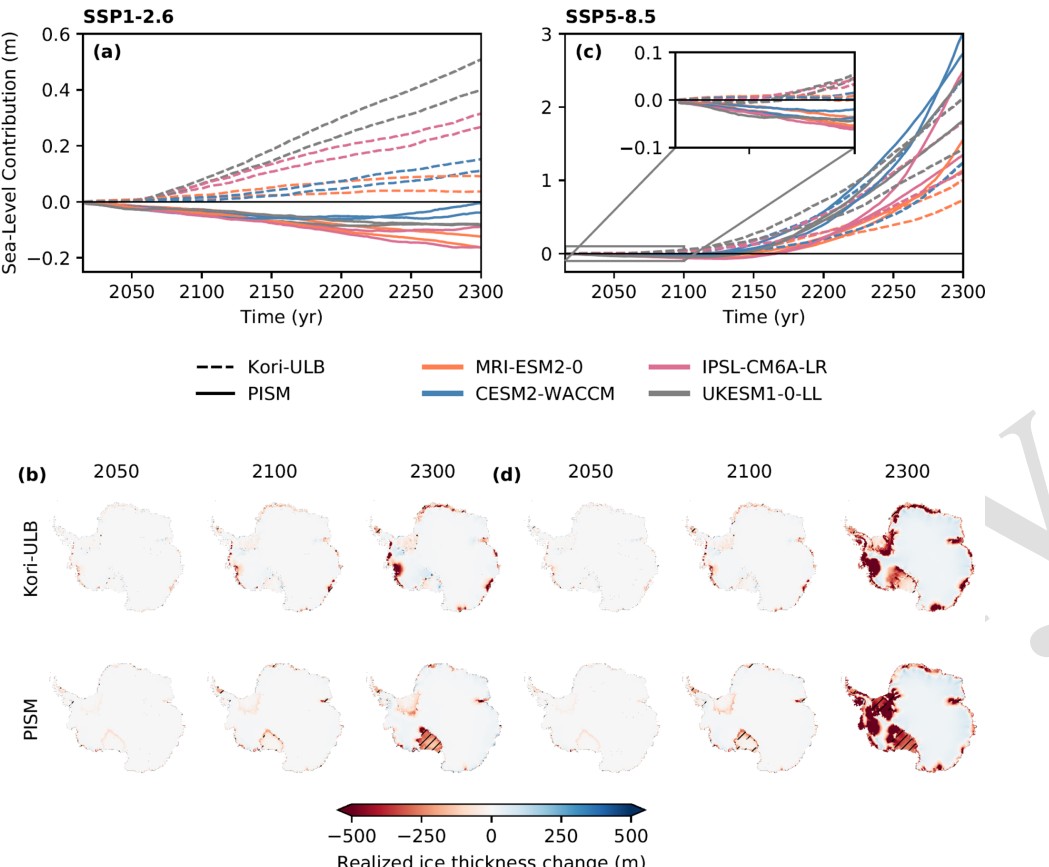

**Figure 3.** Projected Antarctic ice loss on multi-centennial timescales in response to changing climate conditions under emission pathways SSP1-2.6 **(a, b)** and SSP5-8.5 **(c, d)**. **(a, c)** Transient sea-level contribution (in metres of sea-level equivalent) from the Antarctic Ice Sheet until 2300 in response to changing climate conditions projected by four CMIP6 GCMs (given by the colour), as determined by the ice-sheet models Kori-ULB (dashed lines) and PISM (solid lines). **(b, d)** Mean ice-thickness change in the years 2050, 2100 and 2300, as determined by the ice-sheet models Kori-ULB (upper row) and PISM (lower row). The ice-thickness change is averaged across the applied CMIP6 GCMs and the respective ice-sheet model configurations (Table S2). A potential loss of ice shelves is indicated by hatching.

(2015, showing an ice loss of +1.6 to +2.96 m of sea-level equivalent under RCP8.5), for instance, and is consistent with the Antarctic contribution to sea-level change reported in the latest Intergovernmental Panel on Climate Change (IPCC) assessment of −0.3 to +3.2 m of sea-level equivalent (Fox-Kemper et al., 2021). It is caused by pronounced mass loss of the West Antarctica Thwaites and Pine Island glaciers as well as thinning upstream of Ronne–Filchner and Ross ice shelves projected by both ice-sheet models (Fig. 3d). The grounded ice-sheet response is accompanied by thinning of the major Antarctic ice shelves, including the Ross and Ronne–Filchner ice shelves, which are (in PISM only) eventually lost sequentially by 2150 and 2300, respectively (Fig. 3d).

At the same time, the integrated surface mass balance starts to decrease and may turn negative during the next centuries with strong atmospheric warming (Fig. 4c). This pattern is evident in projections by both Kori-ULB and PISM. The decline in the ice-sheet surface mass balance is most pronounced for CESM2-WACCM (Fig. 4a and c, blue), projecting an Antarctic temperature increase of more than 15 °C beyond 2200 and thereby covering atmospheric warming ranges that are not found in any of the other GCMs (Fig. 4a, blue, Table S1). Following the progressing ocean warming under SSP5-8.5 (reaching an ocean temperature change of up to 3 °C for UKESM1-0-LL in 2300; Fig. 4d), sub-shelf melt continues to increase beyond 2100 (Fig. 4f). It eventually levels off after 2150 with the loss of West Antarctic ice shelves (Fig. 4f, also compare to Coulon et al., 2024). An additional increase in sub-shelf melt beyond 2150 occurs in some PISM simulations (Fig. 4f, solid lines), with a substantial contribution from smaller ice shelves in the Ross and Amundsen Sea embayment formed during the grounding-line retreat in combination with the chosen PICO parameters (characterized by a relatively higher melt sensitivity; Reese et al., 2023), overtaking magnitudes of aggregated sub-shelf melt determined in Kori-ULB (Fig. 4f, dashed lines).

**Table 1.** Mean, minimum and maximum values of the combined realized and committed Antarctic ice loss (in metres of sea-level equivalent) in response to changing climate conditions under emission pathways SSP1-2.6 and SSP5-8.5, as determined by the ice-sheet models Kori-ULB and PISM. Ice loss is given for different points in time where climatic boundary conditions are stabilized. For a given point in time, upper rows represent SSP1-2.6 and lower rows correspond to SSP5-8.5.

| | | Realized | Committed in the year 3000 | Committed in the year 5000 | Committed in the year 7000 |
|---|---|---|---|---|---|
| 2050 | SSP1-2.6 | −0.01 | 0.96 | 2.08 | 2.19 |
| | | (−0.03, 0.01) | (−0.11, 2.63) | (0.62, 4.09) | (0.41, 3.97) |
| | SSP5-8.5 | −0.01 | 1.14 | 2.05 | 2.15 |
| | | (−0.03, 0.01) | (−0.04, 3.61) | (0.69, 3.94) | (0.51, 4.21) |
| 2100 | SSP1-2.6 | 0.00 | 1.35 | 2.64 | 2.90 |
| | | (−0.05, 0.08) | (0.21, 3.70) | (0.88, 5.17) | (0.59, 6.32) |
| | SSP5-8.5 | −0.01 | 2.56 | 4.32 | 4.81 |
| | | (−0.06, 0.06) | (0.98, 5.17) | (1.07, 8.01) | (1.18, 8.81) |
| 2150 | SSP1-2.6 | 0.01 | 1.31 | 2.48 | 2.66 |
| | | (−0.08, 0.19) | (0.10, 3.36) | (0.67, 5.71) | (0.31, 6.54) |
| | SSP5-8.5 | 0.09 | 4.51 | 7.30 | 8.79 |
| | | (−0.04, 0.28) | (1.74, 7.61) | (1.84, 13.67) | (1.74, 17.66) |
| 2200 | SSP1-2.6 | 0.03 | 1.10 | 2.37 | 2.54 |
| | | (−0.11, 0.30) | (−0.11, 3.00) | (0.38, 5.52) | (−0.03, 6.22) |
| | SSP5-8.5 | 0.38 | 5.81 | 10.50 | 12.55 |
| | | (0.16, 0.73) | (2.38, 9.88) | (3.54, 19.93) | (4.36, 26.54) |
| 2250 | SSP1-2.6 | 0.05 | 1.24 | 2.67 | 2.77 |
| | | (−0.15, 0.40) | (0.03, 2.94) | (0.45, 5.07) | (0.08, 6.01) |
| | SSP5-8.5 | 0.93 | 7.15 | 14.14 | 17.63 |
| | | (0.40, 1.48) | (2.42, 12.44) | (3.21, 27.14) | (3.92, 36.76) |
| 2300 | SSP1-2.6 | 0.07 | 1.22 | 2.58 | 2.69 |
| | | (−0.16, 0.51) | (−0.13, 2.94) | (0.26, 5.56) | (−0.21, 5.88) |
| | SSP5-8.5 | 1.79 | 7.97 | 16.10 | 19.61 |
| | | (0.74, 3.07) | (3.27, 13.86) | (5.88, 31.67) | (7.52, 40.77) |

In summary, while the ice-sheet trajectories under the lower-emission scenario, SSP1-2.6, are still influenced by the simulated historical trends and differences in ice-sheet modelling choices (Seroussi et al., 2023), we find that climate drivers dominate the projected multi-centennial ice-sheet changes under the higher-emission pathway, SSP5-8.5. In particular, in line with Coulon et al. (2024), our projections indicate that the atmosphere becomes an amplifying driver of Antarctic mass loss beyond the end of this century, irrespective of the ice-sheet model.

## 3.3 Long-term committed ice-sheet evolution over the next millennia

Atmospheric and oceanic warming projected for the upcoming centuries may trigger changes in the dynamics and geometry of the Antarctic Ice Sheet that are not realized on the same timescales (as the forcing) but unfold thereafter over the course of the following millennia due to ice-sheet inertia and nonlinear feedbacks. After determining the transient realized sea-level contribution over the next centuries (Sect. 3.2), we investigate this long-term committed evolution of the Antarctic Ice Sheet by stabilizing climatic boundary conditions at different points in time and by letting the ice sheet evolve over several millennia (compare to Sect. 2.2.1; Fig. 1).

The bulk of our simulations shows that sea level may keep rising for centuries to millennia to come, even if warming is stabilized (Fig. 5a–d; consistent with, e.g. Winkelmann et al., 2015; Van Breedam et al., 2020). The delayed response of the Antarctic Ice Sheet on millennial timescales gives rise to a substantial difference between the transient realized (described in the previous Sect. 3.2) and the long-term committed Antarctic sea-level contributions, being more than 100 (10) times the sea-level change projected by 2100 (2300) (Figs. 3 and 5, Table 1).

We find a sharp increase in the Antarctic sea-level contribution over the next millennium, irrespective of the emission pathway (Fig. 5a–d).

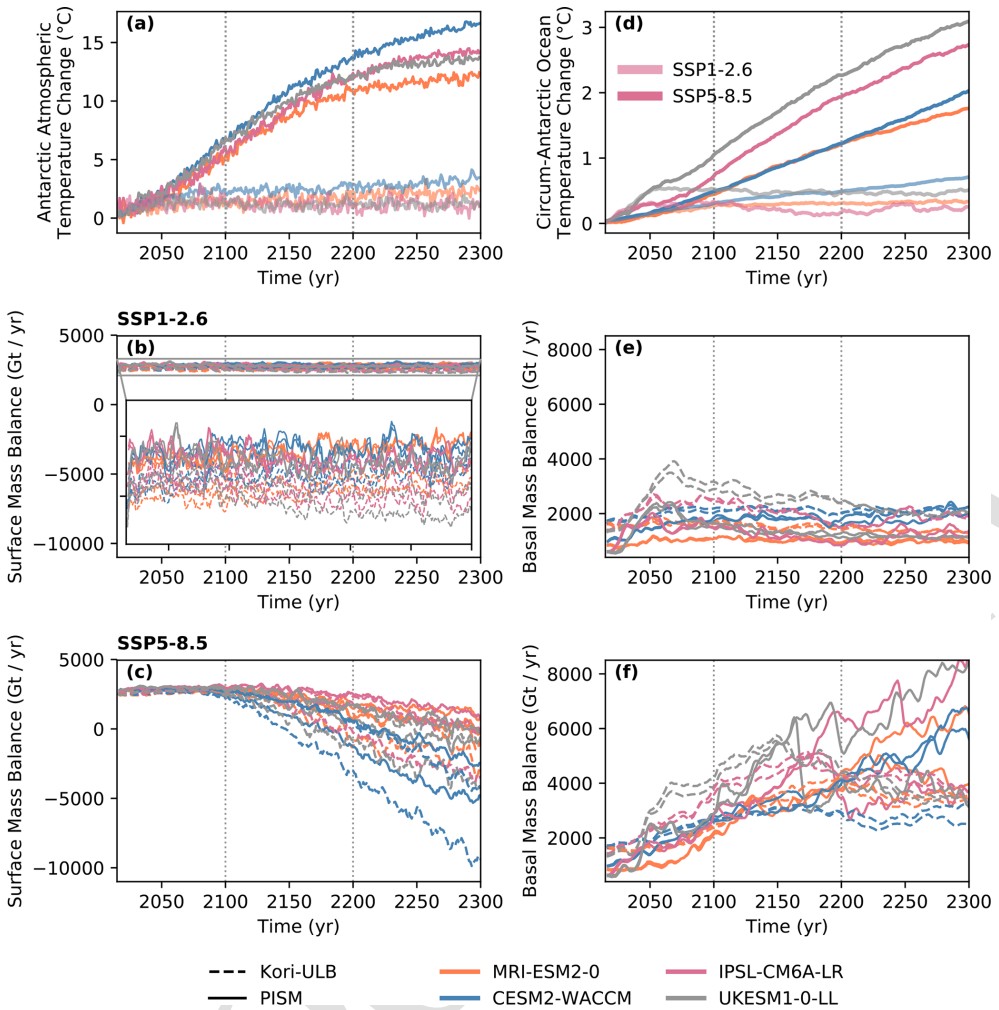

**Figure 4.** Future Antarctic climate and projected ice-sheet mass balance components on multi-centennial timescales under emission pathways SSP1-2.6 (**b** and **e**; transparent colours in **a** and **d**) and SSP5-8.5 (**c** and **f**; opaque colours in **a** and **d**) for four CMIP6 GCMs (given by the colour) in terms of **(a)** Antarctic-averaged atmospheric temperature change, **(b, c)** surface mass balance, **(d)** circum-Antarctic ocean temperature change and **(e, f)** sub-shelf melt as determined by the ice-sheet models Kori-ULB (dashed lines) and PISM (solid lines).

### 3.3.1 Antarctic sea-level commitment under the lower-emission pathway, SSP1-2.6

When following the lower-emission pathway, SSP1-2.6, the ice-sheet response levels off after a peak in the rate of Antarctic ice loss within this millennium or at the latest by the beginning of the following millennium (Fig. 5a and c). Some of the ice-sheet trajectories eventually show a decline in the Antarctic sea-level contribution on multi-millennial timescales (Fig. 5a and c; e.g. for sustained MRI-ESM2-0 climate indicated in orange), with a thickening trend upstream of Ross Ice Shelf (in Kori-ULB only; see below) and in the ice-sheet interior towards the year 7000 that outweighs the initial mass loss. Abrupt changes in the Antarctic sea-level contribution may also be delayed for MRI-ESM2-0 climate (Fig. 5a and c, orange) with a lag of up to multiple millennia from the onset of the perturbation in climatic boundary con-

ditions in PISM simulations. This delay is related to the later onset of substantial grounding-line retreat in the Amundsen Sea embayment in these simulations, with comparably smaller projected oceanic changes in MRI-ESM2-0 (compared to other climate trajectories under the lower-emission pathway; Fig. 4a and d).

Irrespective of the timing of abrupt ice loss, the multi-millennial ice-sheet trajectories are eventually characterized by qualitatively different stages of ice-sheet decline, with a very similar magnitude of the committed Antarctic sea-level contribution determined by the applied GCM forcing for each ice-sheet model (Fig. 5a and c). That is, in our simulations under the SSP1-2.6 pathway, we find that the long-term Antarctic sea-level commitment in the year 7000 does not strongly depend on the point in time after which climatic boundary conditions are stabilized. (Figs. 5e and 6a). When sustaining the warming level potentially reached by

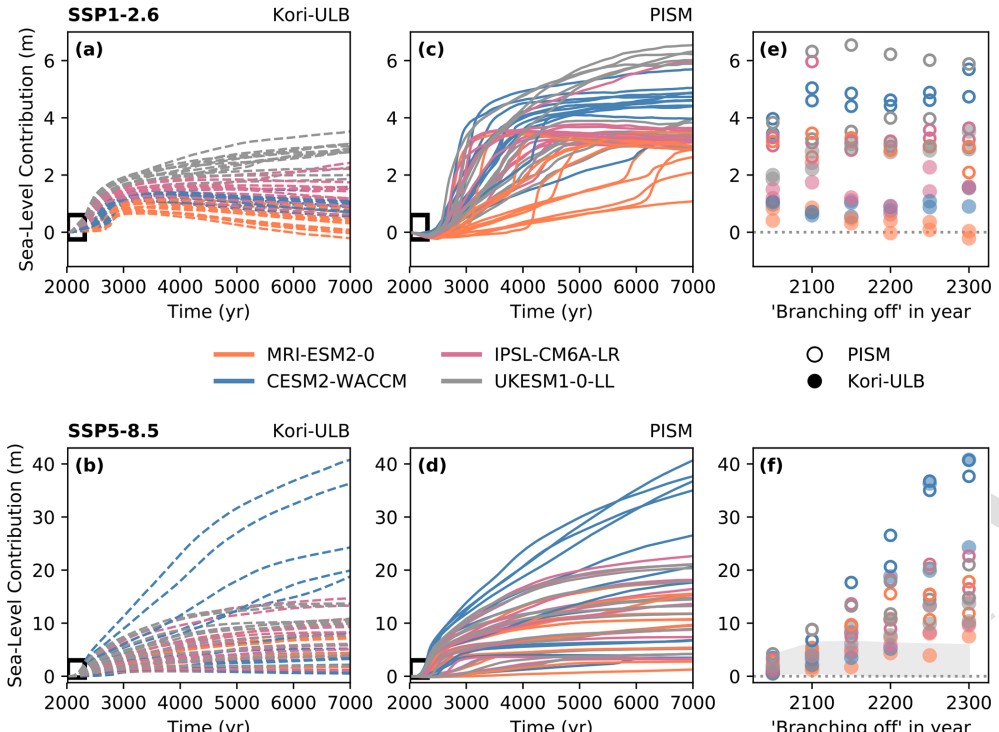

**Figure 5.** Projected Antarctic ice loss on multi-millennial timescales in response to changing climate conditions under emission pathways SSP1-2.6 (**a, c, e**) and SSP5-8.5 (**b, d, f**) projected by four CMIP6 GCMs (given by colour). (**a–d**) Antarctic sea-level contribution (in metres of sea-level equivalent) until the year 7000 to warming potentially reached at different points in time throughout the next centuries, as determined by the ice-sheet models Kori-ULB (dashed lines in **a** and **b**) and PISM (solid lines in **c** and **d**). (**e, f**) Committed Antarctic sea-level contribution in the year 7000 when stabilizing climatic boundary conditions at different points in time (that is, branching off in the years 2050, 2100, 2150, 2200, 2250 and 2300; compare to Sect. 2.2.1 and Fig. 1). Filled and open markers correspond to the long-term sea-level change determined by the ice-sheet models Kori-ULB and PISM, respectively. For comparison of the committed sea-level change under both emission pathways, the range of Antarctic ice loss by the year 7000 under SSP1-2.6 (**e**) is reported as the light-grey shading in (**f**).

2050, sea level may increase by +0.4 to +4.0 m in the long term (Fig. 5e, Table 1). For climate conditions representative of the end of this century and thereafter, Antarctic mass changes range between −0.2 and +6.5 m of sea-level equivalent, which unfold over the next millennia (Fig. 5e, Table 1).

This strong modulation of the magnitude of the committed Antarctic sea-level contribution by the applied GCM forcing for each ice-sheet model (Figs. 5e and S2 in the Supplement) is linked to substantial differences in the trajectories of atmospheric-to-oceanic warming between the applied GCMs under this lower-emission pathway (Fig. 4a and d). Their impact on the ice-sheet response plays out and becomes evident on longer timescales (on the order of millennia).

Across both ice-sheet models and all GCM forcings, we find a long-term recession of the grounding lines in the Amundsen Sea embayment under this lower-emission pathway, with a connection from Pine Island Glacier to Ronne Ice Shelf (Fig. 6a).

Pronounced grounding-line retreat in the East Antarctic Wilkes subglacial basin in both Kori-ULB and PISM, potentially locked in by 2050, adds up to +1.5 m to the long-term

sea-level change (Figs. 5e, grey and pink, and 6a). Long-term ice loss from this region is promoted by the abrupt and stronger ocean warming projected by IPSL-CM6A-LR and (especially) UKESM1-0-LL in the first half of this century compared to the other GCMs (Figs. 4d and S2). The grounding line in the Wilkes subglacial basin CE3 experiences only very limited or no retreat under the other GCM climate trajectories in Kori-ULB and PISM simulations (Fig. S2), respectively, despite stronger atmospheric warming levels when following CESM2-WACCM climate under the lower-emission pathway. This is linked to the less abrupt, more gradual change in ocean temperatures compared to those in UKESM1-0-LL and IPSL-CM6A-LR in the next 2 centuries (Fig. 4a and d).

The magnitude of Antarctic sea-level commitment under SSP1-2.6 warming is further modulated by the long-term consequences of a potential collapse of the Ross and Ronne–Filchner ice shelves: in Kori-ULB, both large ice shelves are preserved through the year 7000, and we find a grounding-line advance and upstream thickening in the Siple Coast region (Fig. 6a). This long-term ice-sheet response in the

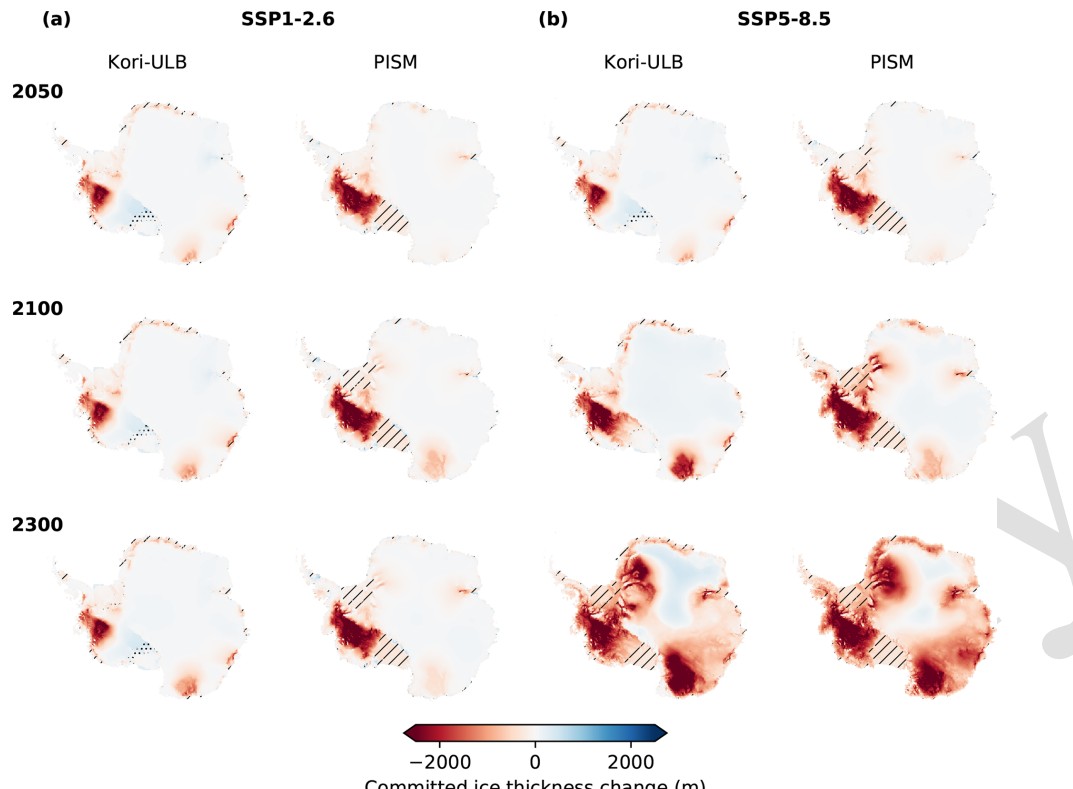

**Figure 6.** Committed ice-sheet responses to changing climate conditions under emission pathways SSP1-2.6 **(a)** and SSP5-8.5 **(b)**. Shown is the mean ice-thickness change in the year 7000 when stabilizing climatic boundary conditions in the years 2050, 2100 and 2300 (from top to bottom), as determined by the ice-sheet models Kori-ULB (left columns) and PISM (right columns). For each ice-sheet model, the ice-thickness change is averaged across the applied CMIP6 GCMs and the respective ice-sheet model configurations (Table S2). A potential loss of ice shelves is indicated by hatching. Dots mark areas of potential ice-sheet advance.

Siple Coast may, in part, result from a drift of the initialization procedure, given lower sub-shelf melt rates obtained with PICO in this area compared to those that were obtained from the initialization approach to keep the ice sheet steady (Sect. 2.2.2). A thickening signal upstream of Ross Ice Shelf has also been observed over the past decades (with the stagnation of Kamb Ice Stream; Smith et al., 2020). The simulated thickening upstream of Ross Ice Shelf contributes to the decay in the long-term Antarctic sea-level contribution over time after the year 3000 in some Kori-ULB simulations, which is most pronounced for sustained MRI-ESM2-0 climate (Fig. 5a, orange). The preservation of these buttressing ice shelves limits the long-term mass loss from Antarctica under SSP1-2.6 to less than +3.5 m of sea-level equivalent in the Kori-ULB simulations (with the upper bound reached under sustained UKESM1-0-LL and IPSL-CM6A-LR climate due to a combined grounding-line retreat in Wilkes subglacial basin and the Amundsen Sea embayment; Fig. 5a and e, filled grey and pink markers). In PISM, a substantial portion of the marine ice-sheet in West Antarctica is lost with the collapse of Ross Ice Shelf and the subsequent retreat of the Siple Coast grounding line by the year 7000 under most

climate trajectories considered (Figs. 6a and S2). The loss of Ross Ice Shelf and the stronger sensitivity of the Siple Coast grounding line under SSP1-2.6 climate in the PISM simulations presented here may be related to the initialized upstream grounding-line location compared to present-day observations (compare Sect. 2.2.2 and Fig. S1; as previously seen in ice-sheet model initializations using a spin-up approach, e.g. Reese et al., 2023; Sutter et al., 2023) and related to the simulated thinning in Ross Ice Shelf over the historical period (compare Sect. 3.1 and Fig. 2c; also when determining the historical ice-sheet evolution at a higher horizontal resolution using PISM, e.g. Reese et al., 2020, 2023). In addition, the higher basal melt sensitivity (compare Sects. 2.2.3 and 3.1) also translates the projected ocean warming into pronounced ice-shelf thinning (Figs. 4f and 6a). Furthermore, once grounding-line retreat is triggered, a collapse of the West Antarctic Ice Sheet may be more likely in PISM than in Kori-ULB, where low slipperiness towards the interior of West Antarctica (given low basal sliding coefficients retrieved in the inverse simulations; Sect. 2.2.2) slows down ice-sheet retreat. The combined ice loss from West Antarctica and the East Antarctic Wilkes subglacial basin in PISM gives

rise to the upper end of the Antarctic sea-level commitment of up to +6.5 m found under the lower-emission pathway in our simulations (Fig. 5c and e, open grey markers).

Ronne–Filchner Ice Shelf remains intact in most of our simulations under SSP1-2.6 except for CESM2-WACCM with strong atmospheric and oceanic changes in the Weddell Sea region in combination with higher sub-shelf melt sensitivities in PISM (Figs. 6a and S2). Its loss triggers ice-sheet retreat in the adjacent Recovery subglacial basin as a consequence of reduced buttressing, raising sea level by up to +5 m in the long term (Figs. 5c and e, open blue markers; 6a; and S2).

Overall, both ice-sheet models agree on a substantial committed retreat in the Amundsen Sea embayment when following the lower-emission pathway, SSP1-2.6, over the coming centuries. With the loss of Ross Ice Shelf, a collapse of the West Antarctic Ice Sheet may even unfold on multimillennial timescales, as shown in the simulations with PISM presented here. Committed mass loss in East Antarctica seems less likely based on our set of simulations and strongly depends on the projected Antarctic climate trajectory under the SSP1-2.6 scenario.

### 3.3.2 Antarctic sea-level commitment under the higher-emission pathway, SSP5-8.5

Under the higher-emission scenario, SSP5-8.5, the potential magnitude and the range of the long-term Antarctic sea-level commitment substantially increase with the point in time at which atmospheric and oceanic warming is stabilized (Fig. 5f). The growing spread of the Antarctic sea-level commitment for a given branch-off point in time is, to a large extent, caused by the divergence of the climate trajectories projected by the four GCMs under this higher-emission pathway (Fig. 4a and d).

Under the sustained warming levels potentially reached during the next decades (that is, by 2050) under these highemission climate trajectories, committed ice-sheet changes are comparable to committed changes under the loweremission pathway, SSP1-2.6 (Sect. 3.3.1). This results from a very similar projected evolution of the Antarctic climate over the first half of this century irrespective of the emission pathway (see above). That is, long-term mass loss is likewise projected to mainly arise from marine regions in West Antarctica (Fig. 6), leading to +0.5 to +4.2 m of sea-level rise on multi-millennial timescales (Fig. 5f, Table 1).

With significant changes in Antarctic climate projected for the end of this century and the atmosphere shifting towards an amplifying driver of mass loss in the transient ice-sheet response (Fig. 4a–d, Sect. 3.2), we also find the committed Antarctic sea-level contribution to be substantially larger, with a doubling of the long-term mass loss ranging between +1.2 and +8.8 m of sea-level equivalent compared to ice-sheet changes triggered within the next decades (Fig. 5f, Table 1). The mass loss by the year 3000 amounts to +1.0 to

+5.2 m of sea-level equivalent (Table 1), consistent with Chambers et al. (2022). Marine parts of the West Antarctic Ice Sheet (now also including the Ross Ice Shelf catchment in simulations with Kori-ULB) show a significant retreat in our entire ensemble of simulations across both ice-sheet models, potentially accompanied by an inland retreat of the grounding line in the East Antarctic Wilkes subglacial basin in the long term (Fig. 6b).

For warming levels that are reached under SSP5-8.5 for any of the branch-off points in time after 2100, the pattern of the committed ice-sheet response is overall consistent between Kori-ULB and PISM. For these branch-off points in time, the difference in future Antarctic climates projected by the four GCMs is significant (Fig. 4a and d, Table S1). Mass loss of the Antarctic Ice Sheet may continue well beyond the end of this millennium with high rates of the Antarctic contribution to sea-level rise until the end of our simulations in the year 7000 (Fig. 5b and d). Depending on the GCM climate trajectory, the long-term ice loss is limited to West Antarctica and the East Antarctic marine Ronne–Filchner, Recovery, Wilkes and Aurora subglacial basins (Figs. 6b and S3 in the Supplement) or also includes parts of the ice sheet grounded above sea level in East Antarctica with very pronounced atmospheric warming in CESM2-WACCM (Figs. 4a, blue; 6b; and S3). For sustained warming levels representative of 2300, the long-term contribution of the Antarctic Ice Sheet to sea-level rise may then be as high as +40.8 m (given sustained CESM2-WACCM climate with ice loss of +7.5 to +40.8 m, +5.9 to +31.7 m and +3.3 to +13.9 m of sea-level equivalent by the years 7000, 5000 and 3000, respectively; compare to Table 1).

Overall, multi-millennial Antarctic ice loss is strongly enhanced with the stabilization of climatic boundary conditions later in time under the higher-emission pathway, SSP5-8.5, across both ice-sheet models, ranging from a long-term collapse of the West Antarctic Ice Sheet in response to warming projected by 2100 to ice loss from major marine subglacial basins in East Antarctica with progressing atmospheric temperature changes after the end of this century. The Antarctic sea-level commitment could even increase up to +40 m with the decline of terrestrial parts of the East Antarctic Ice Sheet for warming reached after 2200, but this is associated with substantial GCM uncertainty.

Comparing the long-term ice loss under both emission pathways, we find that the committed Antarctic sea-level contribution for a given branch-off point in time diverges beyond the end of this century (Fig. 5f; comparing projected multi-millennial ice loss under SSP5-8.5 to that under SSP1-2.6, indicated by the light-grey box). That is, as for the projected transient sea-level change, the emission pathways also become increasingly relevant for the long-term mass loss from Antarctica when stabilizing climatic boundary conditions after 2100 (Sect. 3.2 and Coulon et al., 2024), and thus every decade of additional warming raises the Antarctic sea-level commitment substantially in our simulations. At

the same time, even the lower-emission scenario may pose a considerable risk of Antarctic ice loss raising sea level by multiple metres over the next millennia, depending on the GCM climate trajectory and ice-sheet modelling choices (Sect. 3.3.1).

It should be noted that our simulations end in the year 7000. As a consequence, in some cases the ice sheet has not yet reached a new equilibrium with the sustained climatic boundary conditions (Fig. 5b and d). The complete loss of the East Antarctic Ice Sheet can thus not be ruled out on even longer timescales.

## 3.4 Potential threshold behaviour in response to changing climatic boundary conditions

Figure 7 summarizes the committed sea-level contribution from the Antarctic Ice Sheet (in the year 7000) for a given Antarctic-averaged atmospheric warming level, thereby overcoming the dependency of ice loss on the diverging climate trajectories (Sect. 3.3). It also allows us to explore potential thresholds in the relationship between climatic boundary conditions potentially reached throughout the next centuries and the long-term committed ice-sheet response.

In our simulations, we can identify distinct clusters of qualitatively different ice-sheet behaviour with increasing warming on the continental scale (Fig. 7) and locate critical thresholds in climatic boundary conditions inducing persistent ice loss on a basin scale (Fig. 8).

For Antarctic atmospheric warming levels of up to 4 °C reached by 2300 under SSP1-2.6 and by 2050 under SSP5-8.5, we find a committed collapse of the Amundsen Sea basin and in some cases even a partial collapse of the West Antarctic Ice Sheet, resulting in an Antarctic mass loss of up to +6.5 m of sea-level equivalent over multi-millennial timescales (Fig. 7a and b as a zoom, I and II).

The committed strong grounding-line retreat in the Amundsen Sea embayment for this warming range (consistently shown across both ice-sheet models; Figs. 7c and d, I and II, and 8a and b) is in line with previous work (Golledge et al., 2017; Garbe et al., 2020; Coulon et al., 2024).

Uncertainties in the Antarctic sea-level commitment for Antarctic-averaged atmospheric warming below 4 °C (Fig. 7b) are related to (i) varying ice-sheet sensitivities in the Ross (Figs. 7c and d, I and II, and 8c) and Ronne–Filchner catchments (Figs. 7c and d, II, and 8e and f), depending on ice-sheet modelling choices (compare to Sect. 3.3.1), and (ii) the onset of ice loss in the East Antarctic Wilkes subglacial basin, depending on the ocean warming in the applied GCMs (Fig. 7a, triangles; and c and d, I; compare to Sect. 3.3.1). We find the committed grounding-line retreat in Wilkes subglacial basin being triggered by ocean warming exceeding a basin-averaged ocean temperature change of +0.5 to +1 °C across both ice-sheet models (Figs. 7c, d, and I; 8g; and S4g in the Supplement). The loss of ice in this par-

ticular (Wilkes) subglacial basin is initiated at slightly lower ocean warming than in Golledge et al. (2015) and Garbe et al. (2020) but is located within the range of idealized experiments by Mengel and Levermann (2014). A long-term collapse of Ronne–Filchner Ice Shelf and ice loss from ice streams draining the Eastern Weddell Sea sector is found in PISM simulations when atmospheric warming in Antarctica exceeds 2.5 °C (Figs. 7b and d, II, and 8e and f). This is in line with the long-term ice-sheet response found in Golledge et al. (2015) following emission pathway RCP4.5 with Antarctic atmospheric warming of about 2.2 °C. Given the lower sensitivity of Ronne–Filchner Ice Shelf (Sect. 3.3.1), such a retreat in the subglacial basins connected to this major ice shelf occurs only at higher warming levels for Kori-ULB (see below, Fig. 8e and f).

Antarctic atmospheric warming levels of +5 to +7 °C are projected to occur by the end of this century when following the higher-emission pathway, SSP5-8.5, resulting in a long-term ice loss ranging between +1.2 and +8.8 m of sea-level equivalent (Fig. 7a, III). This spread in Antarctic sea-level commitment contains the recent estimate by Van Breedam et al. (2020) of +6.6 m of sea-level equivalent under warming of approximately +7 °C over the next 10 000 years. In this warming range, ice loss from the Ross Ice Shelf catchment in West Antarctica also contributes substantially to the long-term sea-level change in Kori-ULB (Figs. 7c, III, and 8c). That is, we find a committed complete collapse of the West Antarctic Ice Sheet in both ice-sheet models for Antarctic-averaged atmospheric warming above +4 °C (Fig. 7c and d, III). The inter-model spread in Antarctic mass loss is explained by stronger thinning of ice draining into the eastern Weddell Sea region in PISM (Figs. 7c and d, III, and 8e and f) in combination with more pronounced thickening in the inner parts of East Antarctica in Kori-ULB within this warming range (Fig. S3; when sustaining warming projected for 2100).

Exceeding atmospheric warming of +8 °C in Antarctica gives rise to a further increase in the fraction of ice that is lost on multi-millennial timescales: up to 40 % at +14 °C (Fig. 7, IV). Enhanced long-term mass loss aligned with progressing Antarctic atmospheric warming is found in particular for the Ronne–Filchner, Recovery and Wilkes subglacial basins (Figs. 7c and d, IV, and 8d–g). In addition, some of our simulations suggest the onset of ice drainage from the East Antarctic Aurora subglacial basin at an Antarctic-averaged atmospheric temperature increase of +10 °C (in accordance with Golledge et al., 2015), associated with a substantially increasing long-term contribution to sea-level rise from this basin (for PISM, Figs. 7d, IV, and 8h) in this warming range. The ice stored in the Aurora subglacial basin `CE4` is lost completely in both ice-sheet models at even higher atmospheric warming levels (see below and Fig. 8h). While the general dependency of long-term ice loss from the Aurora region on the atmospheric forcing in our simulations (Fig. 8h compared to Fig. S4h) is in agreement with Golledge et al. (2017),

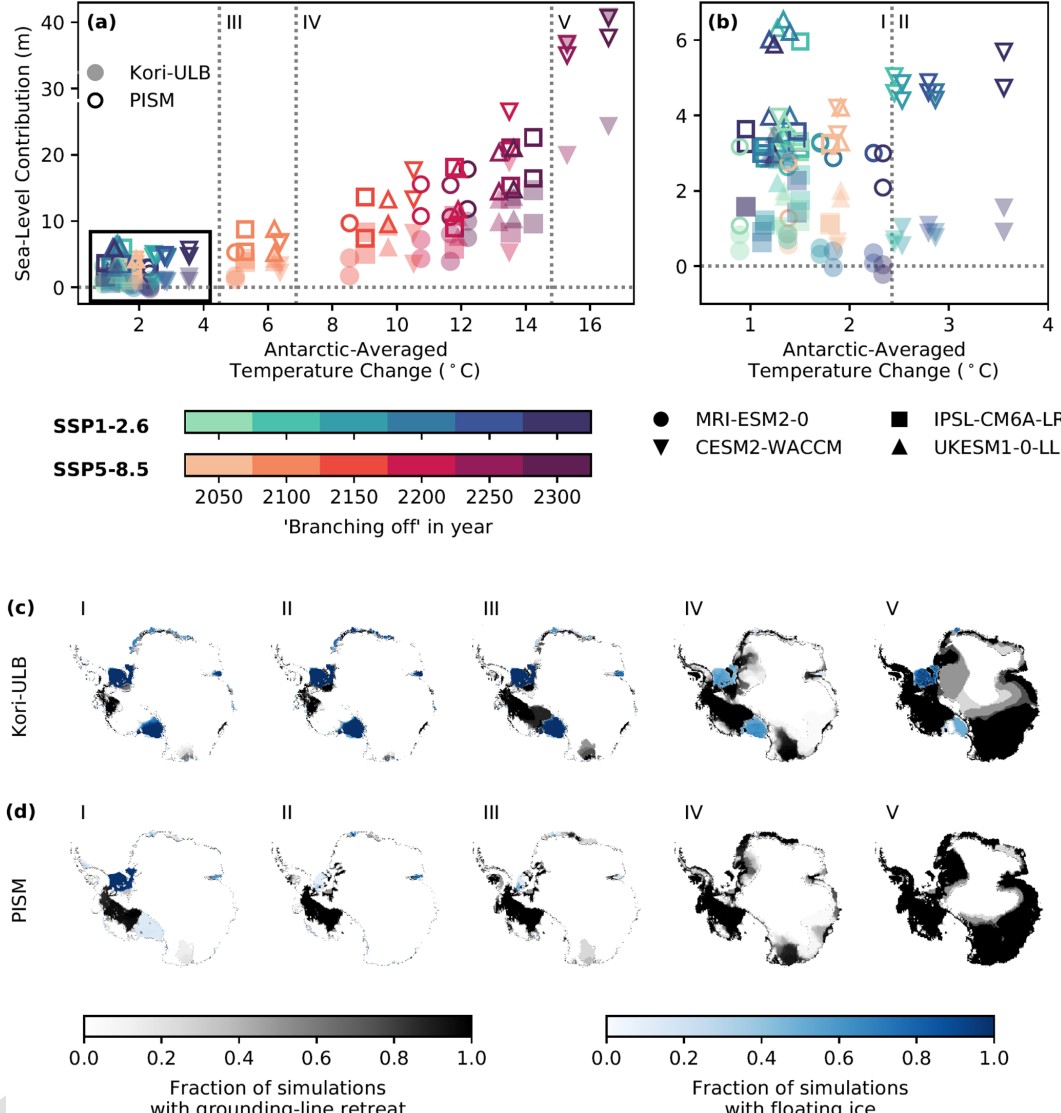

**Figure 7.** Committed sea-level contribution from the Antarctic Ice Sheet. **(a, b)** Long-term ice loss from the Antarctic Ice Sheet (in metres of sea-level equivalent) for the year 7000 in response to Antarctic-averaged atmospheric temperature change (compared to 1995–2014) projected by four CMIP6 GCMs (given by marker shape), as determined by the ice-sheet models Kori-ULB (filled markers) and PISM (open markers). The change in climatic boundary conditions projected at different points in time during the next centuries under emission pathways SSP1-2.6 and SSP5.8.5 (given by the colour) is sustained for several millennia. Panel **(b)** is a zoom of **(a)** for a low-to-intermediate Antarctic-averaged atmospheric temperature change. **(c, d)** Fractions of simulations that show grounding-line retreat in the year 7000 in Kori-ULB **(c)** and PISM **(d)**. The fraction is determined for all simulations that are assigned to the distinct clusters I–V.

stronger warming of the atmosphere is required here to trigger the retreat.

For sustained Antarctic atmospheric warming above +15 °C (projected by CESM2-WACCM after 2200), we find that large parts of the Antarctic marine subglacial basins are lost and ice grounded above sea level in East Antarctica starts to decline substantially on multi-millennial timescales in our entire ensemble of simulations (Fig. 7, V). Over the next millennia, this gives rise to Antarctic ice loss equivalent to a sea-level change of up to +40.8 m. We cannot

rule out that Antarctica could become ice-free under these high-warming trajectories, consistent with Winkelmann et al. (2015) and Garbe et al. (2020), given the continued increase in the Antarctic sea-level contribution at the end of our simulations (Fig. 5b and d).

Overall, the sea-level commitment increases nonlinearly with increasing atmospheric warming in Antarctica (Fig. 7a), consistent with the nonlinear response to warming in quasi-equilibrium (that is, when temperatures change much more slowly than typical rates of changes of an ice sheet; Garbe

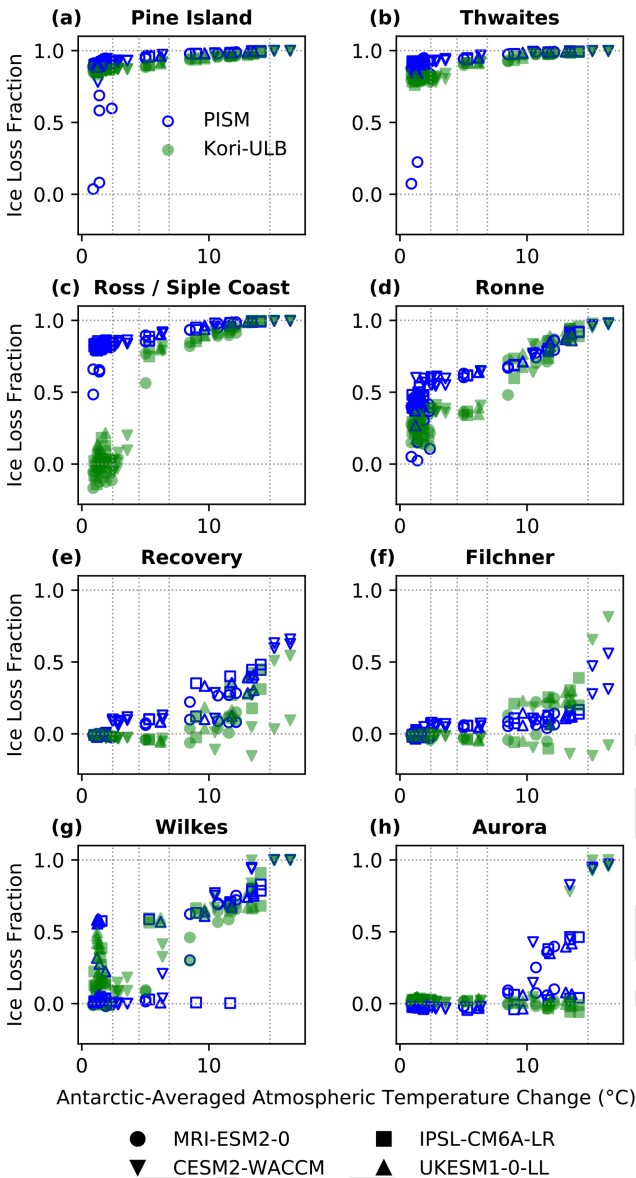

**Figure 8.** Ice loss from Antarctic drainage basins. Long-term ice loss from different Antarctic drainage basins (as a fraction of respective sea-level rise potential) for the year 7000 in response to Antarctic-averaged atmospheric temperature change (compared to 1995–2014) projected by four CMIP6 GCMs (given by marker shape). Filled green and open blue markers correspond to the long-term ice loss determined by the ice-sheet models Kori-ULB and PISM, respectively.

et al., 2020). Our committed ice-sheet states complement this quasi-equilibrium response of the Antarctic Ice Sheet (obtained by Garbe et al., 2020), as they record the long-term response to faster warming as projected under the different SSP scenarios.

Under equivalent warming, the long-term changes in the dynamics and geometry of the Antarctic Ice Sheet are largely consistent for each ice-sheet model configuration (com-

pare to Table S2). We find a spread in long-term Antarctic mass loss at a given warming level due to model uncertainties (e.g. arising in the ice-sheet model initialization and physics), which is pronounced for low-to-intermediate warming levels in Antarctica covered by the lower-emission pathway, SSP1-2.6 (Fig. 7b). In other words, the ice-sheet sensitivity to warming, in particular in some Antarctic marine basins, varies with the ice-sheet model configuration (compare to Table S2). In this warming range, varying trajectories of atmospheric-to-oceanic warming across GCMs may also play out and may modulate Antarctica's sea-level contribution on longer timescales (Sect. 3.3.1). Beyond the low-to-intermediate warming levels covered by the lower-emission pathway, SSP1-2.6, the pattern of long-term mass loss and the resulting sea-level contribution from Antarctica are robust overall, with a step-wise long-term decline of the Antarctic Ice Sheet across two ice-sheet models (Fig. 7a, c and d): with increasing warming, our simulations suggest a committed partial collapse of the West Antarctic Ice Sheet (I and II) associated with a substantial retreat in the Amundsen Sea embayment up to its complete collapse (III), followed by enhanced long-term mass loss from the East Antarctic marine Wilkes, Recovery and Aurora subglacial basins (IV) and an eventual decline of terrestrial parts of the ice sheet (V).

## 4 Discussion

In this paper, we determine the multi-millennial sea-level contribution from the Antarctic Ice Sheet under a range of possible climate trajectories for both low- and high-emission pathways. In particular, we quantify the long-term Antarctic sea-level commitment when stabilizing climatic boundary conditions at different points in time. That is, the atmospheric and oceanic changes potentially established during the upcoming decades and centuries in Antarctica are sustained for several thousands of years, and we explore their long-term impacts on the Antarctic Ice Sheet. Simulations are carried out systematically for stabilized Antarctic climates at different points in time over the course of the next centuries and consistently with the stand-alone ice-sheet models Kori-ULB and PISM, thereby accounting for some model uncertainty.

Our simulations illustrate a substantial difference between the transient realized and long-term committed sea-level contributions from the Antarctic Ice Sheet. While the projected Antarctic mass change by the end of this century is limited (spanning a range from −0.1 to +0.1 m of sea-level equivalent), the Antarctic Ice Sheet may be committed to a strong grounding-line retreat in the Amundsen Sea embayment up to a potential collapse of the West Antarctic Ice Sheet under sustained climatic boundary conditions at levels projected to be reached during this century even under the lower-emission pathway, SSP1-2.6, depending on ice-sheet modelling choices. Mass loss from the marine Wilkes subglacial basin in East Antarctica may unfold on multi-

millennial timescales with the strong ocean warming projected by some GCMs as early as the second half of this century under the lower-emission scenario. With the stabilization of climatic boundary conditions beyond the end of this century under the higher-emission pathway, SSP5-8.5, the long-term Antarctic sea-level contribution under these high-warming trajectories diverges from the sea-level commitment under SSP1-2.6. This is due to a successive ice-sheet retreat in major East Antarctic marine subglacial basins, additionally triggered by progressing warming. Next to ice loss from these marine parts, a substantial decline of the non-marine East Antarctic Ice Sheet may eventually result in long-term mass loss of up to +40.8 m of sea-level equivalent, subject to substantial uncertainties, especially due to the GCM climate forcing.

Determining the committed evolution of the Antarctic Ice Sheet triggered by the warming projected for the next decades and centuries extends previous studies of the long-term ice-sheet response under sustained present-day climate. For example, Golledge et al. (2019), Reese et al. (2023) and Coulon et al. (2024) suggest a committed (potentially irreversible) grounding-line retreat in the Amundsen Sea embayment under present-day climate in response to atmospheric and oceanic changes over the past decades. We also add to the assessment of the long-term Antarctic sea-level commitment to future warming by the end of this century (Lowry et al., 2021; Chambers et al., 2022) and by 2300 (Golledge et al., 2015; Bulthuis et al., 2019; Coulon et al., 2024) by exploring the multi-millennial consequences of stabilizing climatic boundary conditions in a consistent way for two ice-sheet models at different points in time over the course of the next centuries. From a dynamical systems perspective, the long-term stability of the Antarctic Ice Sheet under present and potential future rates of warming (much faster than typical rates of change of an ice sheet) is studied. This complements the quasi-equilibrium ice-sheet response to warming presented by Garbe et al. (2020) and, thereby, bridges the gap between the quasi-equilibrium response and the transient realized sea-level contribution from Antarctica (e.g. by the end of this century as in Seroussi et al., 2020).

The Antarctic sea-level commitment is subject to growing uncertainties related to the substantial spread of warming projected by the selected GCMs, in particular beyond the end of this century under SSP5-8.5 (climate forcing uncertainty). Ice-sheet sensitivities and critical temperature thresholds giving rise to self-sustained ice loss further vary in our simulations as a result of model uncertainties. These uncertainties may be induced by differences in the ice-sheet model structure and the parameterization of certain ice-sheet processes (e.g. Seroussi et al., 2020), parameter choices (e.g. Nias et al., 2016; Bulthuis et al., 2019; Coulon et al., 2024), and initialization approaches (e.g. Aschwanden et al., 2013; Aðalgeirsdóttir et al., 2014; Seroussi et al., 2019; Berends et al., 2023b). In the following, the roles of these different sources of uncertainty in the multi-millennial Antarctic sea-level response presented in this work are discussed.

## 4.1 Uncertainties in Antarctic sea-level commitment due to climate forcing

Following recent efforts in projecting the Antarctic sea-level contribution on different timescales ranging from the next decades until 2100 to several millennia (Seroussi et al., 2020; Payne et al., 2021; Golledge et al., 2015), we base the imposed changes in Antarctic climate on state-of-the-art GCMs from CMIP6. Biases in the chosen GCMs and a poor representation of conditions at the ice-sheet margins due to their coarse resolutions (Beadling et al., 2020; Bracegirdle et al., 2020; Purich and England, 2021) may influence the simulated mass changes of the Antarctic Ice Sheet. In our simulations, the climate forcing from the four employed GCMs results in a wide spread of the realized and especially the committed sea-level contributions from Antarctica, making it one of the most important sources of uncertainty.

For the low-to-intermediate warming levels covered by the lower-emission pathway, SSP1-2.6, the uncertainty introduced by the climate forcing at a given point in time is comparable to the uncertainty caused by ice-sheet modelling choices (compare Fig. 5e and Sect. 4.2). Here, the Antarctic sea-level commitment is modulated by varying trajectories of atmospheric-to-oceanic warming across the GCMs for each ice-sheet model (Fig. 4a and d). We find that the growing spread of the committed Antarctic sea-level contribution under the higher-emission scenario, SSP5-8.5, when stabilizing the ice-sheet boundary conditions later in time (Fig. 5f) can, to a large extent, be attributed to the divergence of the climate trajectories projected by the four GCMs beyond the end of this century (Fig. 4a and d).

The GCMs providing available future changes in Antarctic climate through 2300, on which our analysis is based, are characterized by different warming rates and a large range of climate sensitivities with a high upper bound (Meehl et al., 2020), which may not be in accordance with paleoclimatic evidence (Zhu et al., 2021). This introduces a substantial climate forcing uncertainty at a specific point in time: while for MRI-ESM2-0 a comparably low equilibrium climate sensitivity was determined (3.2 °C, below the multi-model mean of 3.7 °C; Meehl et al., 2020), the equilibrium climate sensitivities of UKESM1-0-LL (5.3 °C), CESM2-WACCM (4.8 °C) and IPSL-CM6A-LR (4.6 °C) are at the upper end of the range of climate sensitivities reported for CMIP6. CESM2-WACCM also shows significantly stronger Antarctic-averaged atmospheric warming under SSP5-8.5 beyond 2200 than projected by the other GCMs (Table S1).

This translates into a growing uncertainty in the projected long-term Antarctic ice loss under SSP5-8.5, with a substantially higher committed sea-level contribution from Antarctica for the same year under sustained CESM2-WACCM climate compared to the other applied CMIP6 GCMs. For ex-

ample, the committed Antarctic sea-level contribution determined by Kori-ULB and PISM under CESM2-WACCM climate ranges between $+24.3$ and $+40.8$ m (when assuming stabilization of climatic boundary conditions representative of 2300). In contrast, we find a long-term Antarctic mass loss of up to $+22.6$ m of sea-level equivalent for the other CMIP6 GCMs. These higher magnitudes of imposed warming in some of the selected GCMs employed here may also explain the substantially higher upper range of the Antarctic mass loss committed under stabilized climatic boundary conditions reached by 2300 in our simulations (with committed ice loss of $+3.3$ to $+13.9$, $+5.9$ to $+31.7$ and $+7.5$ to $40.8$ m of sea-level equivalent by the years 3000, 5000 and 7000; Table 1) compared to some previous estimates (under RCP8.5; Golledge et al., 2015; Bulthuis et al., 2019). Consequently, the derived future climate trajectories and respective projected multi-millennial Antarctic ice loss should only be related to emissions with care and can rather be seen as a potential range of long-term futures for Antarctica.

As noted above, the majority of projections of future Antarctic climate to date cover this century only (O'Neill et al., 2016; Tebaldi et al., 2021). The pronounced projected changes in Antarctic climate and the substantial GCM uncertainty, especially beyond the end of the century, which were reflected in our assessment of the Antarctic sea-level commitment, highlight the need for a thorough assessment of potential multi-centennial Antarctic climate trajectories in future research as a basis for improving our understanding of the associated long-term Antarctic Ice Sheet response (on timescales on the order of centuries to millennia).

To determine the long-term Antarctic sea-level commitment, we here assume constant climatic boundary conditions on multi-millennial timescales. This allows us to assess the committed sea-level contribution from the Antarctic Ice Sheet and its stability under an idealized combination of atmospheric and oceanic changes with respect to the present day. This approach has been invoked previously (e.g. Golledge et al., 2015) but comes with certain assumptions. For instance, a continued ocean response to changing $CO_2$ conditions and atmospheric warming (Li et al., 2013) may result in an altered ratio of atmospheric and oceanic changes beyond the point in time where the stabilization of climatic boundary conditions is assumed here. Observed interannual and decadal variability (Jenkins et al., 2018; Paolo et al., 2018) is neglected in the imposed constant climatic boundary conditions for simplicity, while it has been shown to potentially result in a lower long-term ice-sheet volume (compared to a stable climate; Mikkelsen et al., 2018) that may even include ice-sheet retreat (Robel et al., 2019; Christian et al., 2020).

In addition, climate trajectories distinct from climate stabilization scenarios, e.g. temperature overshoot pathways (Tokarska et al., 2019), may impact the ice-sheet response in the near and far future. The response of the Antarctic Ice Sheet to a reversal of climatic boundary conditions after exceeding warming of, for example, 1.5 °C, including the potential for "safe" overshoots (Ritchie et al., 2021), is not well constrained. Based on the long-term Antarctic sea-level contribution presented here, the (ir)reversibility of this committed Antarctic mass loss for a reversal of climatic boundary conditions and relevant timescales can be assessed in a next step.

## 4.2 Uncertainties in Antarctic sea-level commitment arising from model uncertainties

Under strong atmospheric warming (with an Antarctic atmospheric temperature change above 8 °C in our simulations) where the ice-sheet decline is amplified by atmospheric changes rather than being mostly driven by ocean warming (compare to Coulon et al., 2024), the pattern of long-term mass loss and the resulting sea-level contribution from Antarctica on multi-millennial timescales are robust overall across both ice-sheet models, irrespective of their initialization approaches and structural differences. This includes enhanced ice loss from major East Antarctic marine subglacial basins with progressing warming, following a committed collapse of the West Antarctic Ice Sheet (for warming projected for the end of this century under SSP5-8.5; Sects. 3.3.2 and 3.4).

Model uncertainty is most pronounced for ice loss from Antarctic marine subglacial basins within the low-to-intermediate warming levels covered by the lower-emission pathway, SSP1-2.6. Depending on ice-sheet modelling choices, we find varying timings, basin-scale temperature thresholds and rates of grounding-line retreat in West Antarctica (Sect. 3.3.1 and 3.4 and Fig. 8).

On shorter, multi-centennial timescales, Kori-ULB projections are characterized by a higher sensitivity and an earlier onset of ice loss from the Amundsen Sea embayment under the lower-emission pathway compared to the simulations with PISM presented here (Fig. 3a and b). This stronger dynamical response of the Thwaites and Pine Island glaciers in Kori-ULB results in a higher Antarctic sea-level contribution by 2300, continuing the simulated trends in this region over the historical period (Fig. 2). Both ice-sheet models agree on a long-term retreat of grounding lines of the Pine Island and Thwaites glaciers under SSP1-2.6, consistent with previous findings that the grounding lines at present might already be undergoing a self-sustained retreat or that this retreat might be imminent due to changes in Antarctic climate over the past decades (Favier et al., 2014; Golledge et al., 2019; Reese et al., 2023; Coulon et al., 2024).

The multi-metre spread of the Antarctic sea-level contribution on longer, multi-millennial timescales for Antarctic-averaged atmospheric warming of up to 4 °C (covered by the lower–emission pathway, SSP1-2.6) is, to a large extent, associated with varying ice-sheet sensitivities in the catchments draining the Ross and Ronne–Filchner ice shelves, ac-

companied by different responses of these major buttressing Antarctic ice shelves (Fig. 7; Sect. 3.3.1 and 3.4).

Here, the uncertainty in the onset of ice-sheet retreat can be linked to certain geometrical features of the initial ice-sheet states as an outcome of the applied initialization approaches (Sect. 2.2.2) as well as different but plausible ice-sheet modelling choices, e.g. for determining sub-shelf melt (Sect. 2.2.3). For example, the Ross and Ronne–Filchner ice shelves restraining the ice flowing from the grounded ice sheet are sustained longer overall in Kori-ULB, potentially related to a combination of the calving schemes employed in the ice-sheet models (Levermann et al., 2012; Pollard et al., 2015; DeConto and Pollard, 2016) and to different PICO sub-shelf melt sensitivities to changes in ocean temperatures (Sect. 2.2.3; Reese et al., 2018, 2023). A simulated upstream location of the Siple Coast grounding line (Fig. S1; Reese et al., 2023; Sutter et al., 2023) and thinning of Ross Ice Shelf over the historical period (Fig. 2; Reese et al., 2020, 2023) following a spin-up approach in PISM (Sect. 2.2.2) as well as a potential drift of the Siple Coast grounding line in Kori-ULB (Sect. 3.3.1) given lower sub-shelf melt rates obtained with PICO in this area compared to those that were obtained from the initialization approach to keep the ice sheet steady (Sect. 2.2.2) may also contribute to these varying ice-sheet sensitivities in the Siple Coast region.

The rates of grounding-line retreat are then dictated by basal friction (e.g. Cornford et al., 2020) that deviates spatially between both ice-sheet models: once triggered, faster large-scale West Antarctic grounding-line retreat unfolds in the PISM simulations presented here for low-to-intermediate warming, promoted by overall slippery bed conditions in the interior of marine subglacial basins given the parameterized, bed-elevation-dependent material properties of the subglacial till (in particular, the till friction angle; Sect. 2.1). Grounding lines face less slippery bed conditions when retreating towards the interior of the West Antarctic Ice Sheet in Kori-ULB, based on the optimized lower sliding coefficients from the inverse simulation (Sect. 2.2.2). Therefore, stronger forcing (that is, warming levels reached by the end of this century under SSP5-8.5) is required to overcome this low slipperiness towards the interior of West Antarctica and to induce a complete collapse of the West Antarctic Ice Sheet.

Overall, the uncertainty in Antarctic sea-level commitment to warming projected by 2300 under the lower-emission pathway, SSP1-2.6, associated with ice-sheet modelling choices (ranging from −0.13 to 2.94 m; Table 1) is comparable in the year 3000 to the spread in Antarctic ice-sheet trajectories related to parametric uncertainties in ice–climate interactions (−0.73 to 2.90 m; Coulon et al., 2024).

The range of possible long-term ice-sheet trajectories under SSP1-2.6 suggests the Ross Ice Shelf catchment as an important focus region for future assessments of the multi-millennial Antarctic sea-level contribution, given also a possible intrusion of modified Circumpolar Deep Water onto the eastern Ross Sea continental shelf followed by strong sub-shelf melting (Siahaan et al., 2022), as previously simulated for the Filchner Trough (e.g. Hellmer et al., 2012), and its potential to lead to a complete collapse of the West Antarctic Ice Sheet (e.g. Martin et al., 2019).

In a next step, the long-term ice-sheet response including larger parts of the parameter space covered in the initial-state ensemble and beyond should also be explored to quantify how parametric uncertainties translate into Antarctic sea-level commitment and to extend Coulon et al. (2024) to multi-millennial timescales. While we take into account distinct Antarctic ice-sheet representations as a result of an initial-state ensemble covering relevant model parameters and including two ice-sheet models, parametric uncertainties cannot be fully explored here due to computational constraints in favour of sampling a wide range of possible future climates.

For example, to determine sub-shelf melt, PICO (Reese et al., 2018, 2023) is chosen out of a diverse set of available sub-shelf melt parameterizations (recently compared in, e.g. Burgard et al., 2022; Berends et al., 2023a). PICO has been shown to reproduce observed sub-shelf melt rates related to the vertical overturning circulation in ice-shelf cavities averaged over Antarctic ice shelves and to resemble the typical pattern of strongest melt near the grounding line (Reese et al., 2018, 2023). However, smoother spatial fields of sub-shelf melt in PICO compared to observations (Reese et al., 2018) and quantifying melt as linearly related to temperature (Reese et al., 2018; Burgard et al., 2022) may underestimate the long-term ice-sheet response to changes in Antarctic climate. While the chosen combinations of the overturning parameter and the effective turbulent heat exchange coefficient resemble sub-shelf melt sensitivities and/or observed melt rates (Sects. 2.2.3 and 3.1), substantial parametric uncertainties related to sub-shelf melt exist (e.g. Seroussi et al., 2023; Coulon et al., 2024) and cannot be further explored here.

Finally, our simulations are performed on a comparably coarse horizontal resolution of 16 km, allowing for a large number of long-term simulations as presented here, which was needed to cover a wide range of uncertainties. The migration of the grounding line in PISM is captured reasonably well, even on such a coarse resolution, with a sub-grid interpolation scheme (Feldmann et al., 2014) that enables the reproduction CE5 glacial cycles of the Antarctic Ice Sheet (Albrecht et al., 2020). Garbe et al. (2020) showed (using PISM) that the overall hysteresis behaviour of the Antarctic Ice Sheet is robust across model resolutions. In Kori-ULB, resolving grounding-line dynamics at a coarse resolution is addressed by imposing a flux condition (Pollard and De-Conto, 2012a, 2020), which results in good agreement with high-resolution ice-sheet models.

### 4.3 Limitations related to processes and feedback mechanisms

Several amplifying and dampening feedbacks between the Antarctic Ice Sheet and the Earth System (Fyke et al., 2018) are missing in stand-alone ice-sheet model projections such as the ones presented here but may be relevant to the long-term mass changes and stability of the Antarctic Ice Sheet. Including the missing feedbacks in future fully coupled assessments of the Antarctic sea-level commitment could change the timing and rates of mass loss determined in our simulations, by either accelerating or dampening Antarctic ice loss. However, such fully coupled Earth System models including ice sheets, which are capable of simulating the multi-millennial ice-sheet response as needed for this study, are not yet available.

We here include some of the relevant feedbacks using parameterizations. Atmospheric temperatures imposed on the Antarctic Ice Sheet are modified in our simulations using the atmospheric lapse rate to account for the impact of changing ice-surface elevation. This also feeds into the surface melt determined by the positive-degree-day approach, depicting the surface melt–elevation feedback (Levermann and Winkelmann, 2016).

The potentially strong decline in ice volume under the warming projected for higher-emission pathways may, however, additionally result in changes in the atmospheric circulation and respective precipitation patterns (e.g. compare to Merz et al., 2014, for Greenland), which are not covered in our simulations. By applying the positive-degree-day approach to determine the future ice-sheet surface melt, we do not account for the amplifying melt–albedo feedback (Jakobs et al., 2019, 2021). While polar-oriented regional climate models cannot currently provide boundary conditions (or dynamically interact with ice sheets) on the multi-millennial timescales considered here, intermediate-complexity approaches such as the recently introduced (simple) diurnal Energy Balance Model (Zeitz et al., 2021; Garbe et al., 2023) may allow us to include the potentially accelerating effect of changes in albedo on projected ice loss through enhanced surface melting (Garbe et al., 2023) in the future.

Surface melting of Antarctic ice shelves facilitates hydrofracturing and may trigger ice-shelf collapse (Pollard et al., 2015; Trusel et al., 2015; Lai et al., 2020; van Wessem et al., 2023) and potentially the Marine Ice Cliff Instability (MICI; Bassis and Walker, 2012; Pollard et al., 2015). While temperature thresholds for melt pond formation as a precursor for such ice-shelf loss may be exceeded by the end of the century (van Wessem et al., 2023), the availability of parameterizations to include these processes in ice-sheet models is still limited (Pollard et al., 2015; DeConto and Pollard, 2016; Seroussi et al., 2020). Considering hydrofracturing (following Pollard et al., 2015; DeConto and Pollard, 2016; Seroussi et al., 2020) may speed up grounding-line retreat in Antarc-tic marine subglacial basins due to earlier ice-shelf breakup (Seroussi et al., 2020; Coulon et al., 2024).

In addition, freshwater fluxes from mass balance changes of the Antarctic Ice Sheet into the surrounding ocean have been suggested to result in atmospheric cooling in the Southern Hemisphere, competing with the potential enhancement of ice loss by the end of the century in amplifying feedbacks due to subsurface ocean warming (Golledge et al., 2019; DeConto et al., 2021). This amplifying feedback could have played a role in abrupt ice discharge events during the last deglaciation (Weber et al., 2014). It remains to be explored how such ice–ocean feedbacks could play out on multi-millennial timescales in Antarctica's future.

Finally, while bedrock adjustment to changes in ice load is included in our simulations, opposing Earth structures between West and East Antarctica are not considered: by assuming uniform solid-Earth properties, ocean-driven ice loss from marine subglacial basins in East Antarctica may be underestimated on millennial timescales (Coulon et al., 2021), such that our estimates of the committed East Antarctic mass loss may be seen as conservative. On the other hand, taking into account characteristic rheological properties of the solid Earth in West Antarctica could promote rapid bedrock uplift, thereby delaying ice-sheet changes (Coulon et al., 2021).

## 5 Conclusion

While various sources of uncertainty in quantifying the long-term Antarctic sea-level commitment remain to be explored, our analysis shows across two ice-sheet models and a multitude of varying climate, model and parametric uncertainties that the multi-millennial impacts of warming projected over the next decades and centuries on the Antarctic Ice Sheet are profound when compared to typical sea-level projections, as for instance in the IPCC assessments. The Antarctic sea-level commitment to warming projected over the next centuries increases nonlinearly. The multi-millennial ice-sheet response grows step-wise from a pronounced grounding-line retreat of the Thwaites and Pine Island glaciers under the lower-emission pathway, SSP1-2.6, to a long-term complete collapse of the West Antarctic Ice Sheet triggered in 2100 at the latest for the higher-emission scenario, SSP5-8.5, followed by mass loss from major marine subglacial basins in East Antarctica. It is possible that projected warming after 2200 under SSP5-8.5 gives rise to additional pronounced ice loss of up to +40m of sea-level equivalent from terrestrial ice-sheet parts. Our findings thus stress the importance of complementing typical decadal-to-centennial projections of the future evolution of the Antarctic Ice Sheet by the respective committed Antarctic sea-level contribution for long-term decision making.

*Code and data availability.* The source code for PISM is publicly available on GitHub via https://www.pism.io (last access: 14 August 2024) and Zenodo (https://doi.org/10.5281/zenodo.11910451, Klose et al., 2024). The code for the Kori-ULB ice-sheet model is publicly available on GitHub via https://github.com/FrankPat/Kori-dev (last access: 14 August 2024) and Zenodo (https://doi.org/10.5281/zenodo.8398771, Coulon et al., 2023). The data needed to produce the figures and tables, and the scripts have been archived within the Zenodo repository at https://doi.org/10.5281/zenodo.11910451 (Klose et al., 2024). All datasets used in this study are freely accessible through their original references. The CMIP6 forcing data used in this study are accessible through the CMIP6 search interface (https://esgf-node.llnl.gov/search/cmip6/, ESFG, 2024).

*Supplement.* The supplement related to this article is available online at: https://doi.org/10.5194/tc-18-1-2024-supplement.

*Author contributions.* RW conceived the study. AKK and VC processed the forcing data, initialized the ice-sheet models and ran the model simulations. AKK performed the data analysis, produced the figures and wrote the original manuscript with regular inputs from VC. All authors contributed to the final version of the paper.

*Competing interests.* The contact author has declared that none of the authors has any competing interests.

ther geographical representation in this paper. While Copernicus Publications makes every effort to include appropriate place names, the final responsibility lies with the authors.

*Acknowledgements.* This publication was supported by PROTECT. This project received funding from the European Union Horizon 2020 research and innovation programme under grant agreement no. 869304, PROTECT contribution number 97. The authors gratefully acknowledge the European Regional Development Fund (ERDF), the German Federal Ministry of Education and Research (BMBF), and Land Brandenburg for supporting this project by providing resources on the high-performance computer system at the Potsdam Institute for Climate Impact Research. Development of PISM is supported by NASA grants 20-CRYO2020-0052 and 80NSSC22K0274 and NSF grant OAC-2118285. Computational resources for Kori-ULB simulations have been provided by the Consortium des Équipements de Calcul Intensif (CÉCI), funded by the Fonds de la Recherche Scientifique de Belgique (F.R.S.-FNRS) under grant no. 2.5020.11 and by the Walloon Region. Ann Kristin Klose and Ricarda Winkelmann were supported by the European Union Horizon 2020 research and innovation programme under grant agreement no. 820575 (TiPACCs). We acknowledge the World Climate Research Programme's Working Group on Coupled Modelling, which is responsible for CMIP, and we thank the climate modelling groups (whose models are listed in Table S1 of this paper) for producing and making their model output available. This work has been performed in the context of the FutureLab on Earth Resilience in the Anthropocene at the Potsdam Institute for Climate Impact Research.

*Financial support.* This research has been supported by the EU Horizon 2020 programme (grant nos. 869304 and 820575).

The publication of this article was funded by the Open Access Fund of the Leibniz Association.

*Review statement.* This paper was edited by Florence Colleoni and reviewed by one anonymous referee.

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

**Remarks from the language copy-editor**

CE1    Thank you for the clarification regarding feedbacks and the prepositions in/of. All these changes have been made throughout the paper.

CE2    Your proposed change was not grammatically correct. Would this fit your meaning?

CE3    Here the article is necessary because you are referring to the basin.

CE4    Please see my previous comment about "basin".

CE5    Enable + a following verb does require an object, but the noun construction does not. Please see the change here.