# Peer review of "The long-term sea-level commitment from Antarctica"

_The Cryosphere, 2023_

## Referee Comment (RC1)

Klose et al. investigates the millennial-scale commitment of the Antarctic Ice Sheet to global sea level from 21$^{st}$ century emissions scenarios using an ensemble of two continental-scale ice sheet models. Their results demonstrate that a multi-meter sea-level commitment of a low emissions scenario (SSP1-2.6) cannot be ruled out over millennial timescales, highlighting the difference between what they define as the transient "realised" sea level contribution from the long-term "committed" sea level contribution. Under high emissions (SSP5-8.5), the sea level contribution is as high as 40 m over 7000 years, with significant loss of the EAIS.

The study is a comprehensive and well-written and has potential to be a useful contribution to *The Cryosphere*. One of the main points of novelty of this study is that the millennial-scale projections are performed with two different ice sheet models in a consistent manner. However, the approaches used to initialise these ice sheet models are very different, resulting in quite substantial bias in one of the models. It does not seem clear to me that the authors have considered to what extent the inter-model uncertainties they describe are due to these large differences in the ice sheet initial state. I suggest that the authors address this aspect in particular to improve the work. This review is divided into general comments and specific comments.

General comments

Two different approaches are used to initialise the ice sheet models. For PISM, an ensemble of spin-up simulations were run and the model output were scored based on the fit of modelled floating and grounded ice area, ice thickness, and surface velocity to Bedmap2 and Rignot et al. (2011) velocities. Fig S1 indicates that the initial state of PISM for the projection is hundreds of meters too thick for most of WAIS, and hundreds of meters too thin for most of EAIS. For Kori-ULB, a nudging procedure is implemented to minimise model drift from Bedmachine-Antarctica. The authors should elaborate on why these two different methods are employed as well as the potential impacts on their results.

Differences in the input datasets, such as bed topography, could alone account for some of the ice sheet model differences (e.g. Wernecke et al., 2022), but this is generally not discussed other than with regard to present-day climatologies.

The results section jumps straight into the projection experiments, but I think it is worth commenting on the historical simulations. Notably, the two models show differences in the direction of SLE change over the course of the historical run, with one of the models showing a basal melt rate of nearly double the other (Fig S2). This is important context for the transient ice-sheet response.

As a specific example of the impact of the initial state, for SSP1-2.6, the two different ice sheet models display different short-term and long-term behaviour. In general, Kori-ULB simulations show a higher centennial-scale sea level contribution than the PISM simulations, but this reverses for many of the simulations by the year 3000 (judging from Fig 2c). To what extent is this slower but eventually larger response of PISM due to its initial bias in ice thickness? One of the key findings highlighted in the abstract is this large sea level contribution under SSP1-2.6 of up to 6 m, but would a PISM model with a thinner initial WAIS produce the same result? This is worth exploring with a few sensitivity experiments.

Specific comments

Could you provide a table of experiments run, both in terms of forcings and ice sheet model parameter values?

Line 180: capitalise "Initialisation"

Line 184-185: NorESM1-M is CMIP5? "ocean and atmosphere anomalies" refers to anomalies of this particular GCM?

Fig 1a: Should the top part of the curve have a steeper slope? (i.e. accelerated mass loss)

Line 192: By atmospheric climatologies, do you mean from the RCMs or the CMIP forcing?

Line 204: So scoring for Bedmap2 for PISM, but Bedmachine for Kori-ULB?

Line 214: "balanced"

Line 230: "GCMs"

Line 240-243: Do you average at or over a particular depth?

Line 270: Is the reason for the difference in parameter values that they produce better fit to observations for those particular ice sheet models?

Line 278: Is the negative SLE change from a model that has positive SLE change over the historical period? It is worth specifying if the models that show modern ice mass loss have a negative or positive SL contribution.

Line 280: I would think the difference in initial state is a larger contributor to the differences between the two models than the dynamic response to the forcings.

Line 284-286: PISM is too thick in the ASE to start with.

Line 297-300: The models don't include hydrofracture parameterization, correct? Does surface temperature of ice shelves reach threshold for melt pond formation? e.g. van Wessem et al. (2023)

Line 303: "(ISMIP6)"

Line 315: But you have some simulations that show the opposite

Line 339-342: This sentence is confusing.

Figure 2: I suggest changing the colors because SSP1-2.6 is generally dark blue, and SSP5-8.5 is generally dark red, but here the colors refer to different GCMs.

Fig 3: Clarify the time of commitment. Are these means of both models?

Line 391: "changes"

Line 399: By regional warming, do you mean of air temperature?

Line 406: Is this realistic that the Ross Ice Shelf doesn't collapse under sustained SSP5 forcing? It could be an artifact of the basal melt parameterization, or the fact that there is no hydrofracture parameterization implemented.

Line 409: For how much warming? And is this due to the initial condition (i.e. PISM has more of WAIS to lose…)?

Line 413: What do you mean by "uncertainty"?

Line 477: The projection experiments are consistent, but the initialisations are not.

Line 567-593: Initial state discussion focuses on the choice of present atmospheric forcing, but what about the initialisation procedure, which seems to result in quite different ice thicknesses?

I suggest that you reduce the number of supplemental figures by consolidating Fig S4 to Fig S19.

References

Wernecke, A., Edwards, T. L., Holden, P. B., Edwards, N. R., & Cornford, S. L. (2022). Quantifying the impact of bedrock topography uncertainty in Pine Island Glacier projections for this century. *Geophysical Research Letters*, *49*(6), e2021GL096589.

van Wessem, J. M., van den Broeke, M. R., Wouters, B., & Lhermitte, S. (2023). Variable temperature thresholds of melt pond formation on Antarctic ice shelves. *Nature Climate Change*, 13(2), 161-166.

---

## Editor Comment (EC1)

[revised manuscript text omitted]

---

## Author Comment (AC1)

**Response to the comments of the reviewers for the manuscript**
**'The long-term sea-level commitment from Antarctica'**

**by A. K. Klose, V. Coulon, F. Pattyn, and R. Winkelmann**

We would like to thank the reviewers for carefully reading our manuscript and for their efforts in creating their review comments. We considered their suggestions thoroughly and adapted the manuscript accordingly.

In the revised version of the manuscript, we have particularly addressed the following points as raised by all reviewers:

- We have included a discussion of the applied **ice-sheet initialization approaches** as well as an **assessment of the resulting initial ice-sheet states** and the **historical ice-sheet trajectories** determined by Kori-ULB and PISM in comparison to observations.
- We have reformulated the **description of our results** to improve clarity and readability. Our results are presented in a more detailed context of the future Antarctic climates projected by the applied four CMIP6 GCMs. Ice-sheet model agreement in the short- and long-term Antarctic sea-level contribution is highlighted and uncertainties in the Antarctic sea-level commitment due to both diverging climate trajectories and model uncertainties (including uncertainties in ice-sheet processes, their parameterisation in ice-sheet models and distinct initialization approaches) are assessed.
- We have reorganized and added **figures** to allow a better understanding of the simulated ice-sheet response on different timescales, depending on the applied GCM forcing and ice-sheet models.

Note that PISM experiments have been rerun to ensure consistency of the till friction angle parameterization across possible restarts of the experiments (compare Section 2.1 in the original manuscript), with a negligible difference to the simulations presented in the original manuscript.

We provide a point-by-point response to all comments below. The reviewers' comments are given in bold font (with related sections from the original manuscript given in grey), the authors' reply in normal font and changes to the text in italic font. Line numbers mentioned in our responses refer to the manuscript version showing how these proposed changes would be implemented. It is attached at the end of this document.

We are grateful for the opportunity to further improve our manuscript and are looking forward to your feedback.

Sincerely yours,

Ann Kristin Klose, Violaine Coulon, Frank Pattyn, and Ricarda Winkelmann

*Reviewer Comment 1*

**Klose et al. investigates the millennial-scale commitment of the Antarctic Ice Sheet to global sea level from 21st century emissions scenarios using an ensemble of two continental-scale ice sheet models. Their results demonstrate that a multi-meter sea level commitment of a low emissions scenario (SSP1-2.6) cannot be ruled out over millennial timescales, highlighting the difference between what they define as the transient "realised" sea level contribution from the long-term "committed" sea level contribution. Under high emissions (SSP5-8.5), the sea level contribution is as high as 40 m over 7000 years, with significant loss of the EAIS.**

**The study is a comprehensive and well-written and has potential to be a useful contribution to *The Cryosphere*. One of the main points of novelty of this study is that the millennial-scale projections are performed with two different ice sheet models in a consistent manner. However, the approaches used to initialise these ice sheet models are very different, resulting in quite substantial bias in one of the models. It does not seem clear to me that the authors have considered to what extent the inter-model uncertainties they describe are due to these large differences in the ice sheet initial state. I suggest that the authors address this aspect in particular to improve the work. This review is divided into general comments and specific comments.**

We are grateful for the overall positive evaluation of our work. We thank the reviewer for carefully reading our manuscript and providing us with helpful comments to improve our manuscript, in particular, with regard to the discussion of the potential role of different initialization approaches for model differences in the Antarctic sea-level contribution on shorter (muli-centennial) and longer (multi-millennial) timescales.

**General comments**

**Two different approaches are used to initialise the ice sheet models. For PISM, an ensemble of spin-up simulations were run and the model output were scored based on the fit of modelled floating and grounded ice area, ice thickness, and surface velocity to Bedmap2 and Rignot et al. (2011) velocities. Fig S1 indicates that the initial state of PISM for the projection is hundreds of meters too thick for most of WAIS, and hundreds of meters too thin for most of EAIS. For Kori-ULB, a nudging procedure is implemented to minimise model drift from Bedmachine-Antarctica. The authors should elaborate on why these two different methods are employed as well as the potential impacts on their results.**

**Differences in the input datasets, such as bed topography, could alone account for some of the ice sheet model differences (e.g. Wernecke et al., 2022), but this is generally not discussed other than with regard to present-day climatologies.**

**The results section jumps straight into the projection experiments, but I think it is worth commenting on the historical simulations. Notably, the two models show differences in the direction of SLE change over the course of the historical run, with one of the models showing a basal melt rate of nearly double the other (Fig S2). This is important context for the transient ice-sheet response.**

**As a specific example of the impact of the initial state, for SSP1-2.6, the two different ice sheet models display different short-term and long-term behaviour. In general, Kori-**

**ULB simulations show a higher centennial-scale sea level contribution than the PISM simulations, but this reverses for many of the simulations by the year 3000 (judging from Fig 2c). To what extent is this slower but eventually larger response of PISM due to its initial bias in ice thickness? One of the key findings highlighted in the abstract is this large sea level contribution under SSP1-2.6 of up to 6 m, but would a PISM model with a thinner initial WAIS produce the same result? This is worth exploring with a few sensitivity experiments.**

We thank the reviewer for the detailed general comment that is very helpful for improving the presentation and discussion of our results, in particular with regard to the applied initialization approaches and potentially related model differences in the response of the Antarctic Ice Sheet. Our response to this general comment is structured along the following main points raised by the reviewer:

(1) Applied initialization approaches and resulting initial ice-sheet states compared to observations
(2) Historical ice-sheet trajectories simulated by Kori-ULB and PISM compared to observations
(3) Differences in ice-sheet model behaviour on shorter (multi-centennial) and longer (multi-millennial) timescales, in relation to the initial ice-sheet states

**(1) **Applied initialization approaches and resulting initial ice-sheet states**

Two different initialization approaches are included in our work, based on the state-of-the-art ice-sheet models Kori-ULB and PISM. We do not consider the application of these different initialization approaches as inconsistency in our experimental setup, but rather as an advantage. Given that we include two common ways of initializing ice-sheet models (compare e.g., Seroussi et al., 2019, 2020), we sample uncertainties associated with the choice of the initialization approach.

In our view, this is important, as no single initialization approach available and applied to date can lead to an initial ice-sheet state that fully captures the characteristics of the (present-day) Antarctic Ice Sheet. While an inverse simulation allows to generate ice-sheet states that match (present-day) observations to a large degree (e.g., in terms of the ice-sheet geometry), the resulting parameter fields (e.g., basal sliding coefficients) may compensate for errors or uncertainties in other ice-sheet processes (Berends et al., 2023; Aschwanden et al., 2013). In addition, this approach assumes that the fields obtained in the inverse simulation to match (present-day) observations do not change in the future. In contrast, in the ice-sheet state resulting from a spin-up, the ice-sheet variables may be modelled in a consistent way, but its geometry might differ from the observed ice sheet. It is the result of the covered ice-sheet physics in the model for a set of uncertain parameters, without any nudging.

At present, each ice-sheet model tends to apply a preferred initialization approach for e.g. projections of the Antarctic sea-level contribution. While some assessments on the influence of the initialization approach exist (e.g., in the framework of initMIP, Seroussi et al., 2019), a direct comparison of different initialization approaches within a single ice-sheet model and their impacts on Antarctic sea-level projections is rare and should be part of future research. As such, there is no clear evidence that either of the initialization approaches is to be preferred.

In line with the editor comments, we have included a discussion of these initialization approaches and their advantages as well as a comparison of the initial ice-sheet states in our study to observations in Section 2.2.2 of the revised manuscript (lines 237-255):

*Given that we include two common ways of initializing ice-sheet models (compare e.g., Seroussi et al., 2019, Seroussi et al., 2020), we sample uncertainties associated with the choice of the initialization approach. While an inverse simulation allows to reproduce the present–day ice sheet geometry well, the resulting parameter fields (such as basal sliding coefficients in Kori-ULB) may compensate for errors or uncertainties in other ice–sheet processes (Berends et al., 2023b; Aschwanden et al., 2013). In addition, it is assumed that the field obtained in the inverse simulation to match present–day observations does not change in the future. In contrast, in the simulated ice–sheet state resulting from a spin-up the ice–sheet variables may be modelled in a consistent way, but its geometry might differ from the observed ice sheet. It is the result of the covered ice–sheet physics in the ice–sheet model for a set of uncertain parameters, without any nudging.*

*The simulated grounding–line position and ice thickness of the initial ice–sheet states are compared to present–day observations in Figure S1. As a result of the inverse simulation, the grounding–line position and ice thickness compare well to present–day observations in the initial ice–sheet states for Kori-ULB (Fig. S1a and c). With the spin–up approach applied in PISM, the initial ice–sheet states are characterized by larger ice thickness differences compared to present–day observations (Fig. S1b and d). Overall, ice in West Antarctica and in some coastal regions in East Antarctica (e.g,. in Dronning Maud Land, upstream of Amery Ice Shelf and in Wilkes Land) is thicker than observed at present (comparable to Reese et al., 2023), while the ice thickness in the interior of East Antarctica is underestimated. In addition, the grounding line in the Siple Coast area (and in the catchment draining Ronne–Filchner Ice Shelf for the MAR climatology) is located upstream of the observed grounding line in the present–day (Fig. S1 b and d), as previously seen in a model initialisation in a spin–up approach, e.g., Reese et al. (2023) and Sutter et al. (2023). These differences should be taken into account when interpreting the simulated long–term evolution of the Antarctic Ice Sheet.*

As an outcome related to these initialization approaches, basal friction deviates spatially between both ice-sheet models, in particular in the interior of West Antarctic marine basins. This can also be expected to influence the ice-sheet response and its timescales. In Section 2.2.2 of the revised manuscript, a paragraph describing the optimized field of basal sliding coefficients in Kori-ULB in comparison to the parameterized material properties of subglacial till in PISM is added (lines 225-230) as follows:

*The optimized field of basal sliding coefficients in Kori-ULB is characterized by high basal sliding coefficients at the ice–sheet margins, turning into regions of low slipperiness (low basal sliding coefficients) towards the interior of West Antarctica. It thus differs from the basal friction experienced by the Antarctic Ice Sheet in experiments with PISM, where overall slippery bed conditions in the interior of marine subglacial basins are found, given the parameterized, bed–elevation dependent material properties of the subglacial till (in particular, the till friction angle; Sect. 2.1). These inter–model differences in basal friction linked to the applied initialization approaches are expected to influence the ice–sheet response.*

(2) **Historical ice-sheet trajectories simulated by Kori-ULB and PISM compared to observations**

Over the historical period and in response to the historical NorESM1-M climate trajectory, we find ice-sheet thinning in the Amundsen and Bellingshausen Sea regions and ice-sheet thickening in the interior of East Antarctica, overall matching the observed pattern of mass change in these regions (e.g., Smith et al., 2020) with both ice-sheet models.

In the PISM simulations presented in our study, the magnitude of mass loss from Thwaites and Pine Island glaciers in the Amundsen Sea sector is, however, underestimated compared to these observations. This ice loss in West Antarctica (and the Antarctic Peninsula) is dominating the overall observed ice-sheet mass changes to date (Otosaka et al., 2023). The lower sensitivity in the Amundsen Sea sector over the historical period in the PISM simulations shown here may thus explain the overall negative sea-level contribution to 2015. It may also contribute to the delayed response in this region on centennial timescales compared to ice-sheet changes projected by Kori-ULB, and should be taken into account when interpreting the projected Antarctic Ice Sheet evolution.

In the presented PISM simulations, we also find ice loss for Ross, Ronne-Filchner and Amery ice shelves, in contrast to observed ice thickening in these regions in present-day (Smith et al., 2020). This sensitivity in the Siple Coast region already during the historical period may also contribute to a larger long-term response in the PISM simulations presented here compared to the ice-sheet changes projected by Kori-ULB for low to intermediate warming levels.

Note that the hindcasting period of 65 years is relatively short (with an overlap of only about 35 years with available observations) compared to the typical response timescales of ice sheets of up to thousands of years, also given a lack of observational records. In addition, the hindcasts presented here are based on the historical climate trajectory from a single GCM (NorESM1-M; Bentsen et al., 2013) with potential biases in Antarctic climate, that could also cause some discrepancies in the simulated ice-sheet evolution compared to observations.
To date, many projections of future Antarctic Ice Sheet trajectories, also those presented in the recent IPCC assessment (Fox-Kemper et al., 2021), do not include any historical period. A multi-model community effort may be required to improve model hindcasts over the observational period for the Antarctic Ice Sheet (Aschwanden et al., 2021).

In line with the general editor comment, we have (1) added an assessment of the simulated ice-sheet trajectories in response to the NorESM1-M climate trajectory over the historical period as outlined above as Section 3.1, (2) included a related figure as Figure 2, and (3) linked the simulated historical ice-sheet response to the projected ice-sheet response to 2300 (Sect. 3.2) in the revised manuscript.

The main related paragraph in the revised manuscript (lines 314-338) reads as:

> *The pattern of observed present–day rates of ice–thickness change (e.g., Smith et al., 2020) is overall captured by both ice–sheet models in response to the historical NorESM1-M climate trajectory (Fig. 2a – c), with a thinning in the Amundsen and Bellingshausen Sea region and the Antarctic Peninsula and a thickening in the ice–sheet interior. The magnitude of ice–sheet thinning in the Amundsen Sea Embayment is, however, underestimated compared to present–day observations in the historical simulations with PISM presented here (Fig. 2a and c). In addition, we find ice loss for*

*Ross, Ronne–Filchner and Amery ice shelves in PISM in contrast to observations (Fig. 2a and c).*

*The evolution of the continent–wide integrated surface mass balance is relatively similar for both ice–sheet models, but occurs on a higher, though still within RCM uncertainties, level in PISM than in Kori-ULB (Fig. 2d). While sub–shelf melt increases in PISM from about 300 Gt yr−1 in 1950 towards 1100 Gt yr−1 in 2015 at the lower end of present–day observations (Fig. 2e, solid lines), the basal mass balance is on the order of the observational record in Kori-ULB over the entire historical period, slightly exceeding its upper end in 2015 with about 1800 Gt yr−1 (Fig. 2e, dashed lines). The continent–wide aggregated sub–shelf melt rates observed in present–day are thus reproduced with both sets of PICO parameters (see Sect. 2.2.3), but they result in different sensitivities of sub–shelf melt rates to ocean temperature changes over the historical period (Fig. 2e; Reese et al., 2023).*

*Mass loss in the Amundsen Sea sector dominates the overall observed ice sheet mass changes in Antarctica to date (Otosaka et al., 2023). Given the lower magnitude of ice–sheet thinning of Pine Island and Thwaites glaciers in PISM, and stronger sub–shelf melt in Kori-ULB, we find diverging ice–sheet trajectories with both ice–sheet models in terms of the Antarctic sea–level contribution over the historical period from 1950 to 2015: Kori-ULB shows an integrated mass loss with a sea–level contribution of about +4 mm in 2015 (Fig. 2f, dashed lines), while the ice sheet overall gains mass equivalent to a sea–level change ranging between -4 mm and -6 mm in PISM (Fig. 2f, solid lines; within spread of recent ensemble of historical ice–sheet trajectories, Reese et al., 2023).*

*In the future evolution of the Antarctic Ice Sheet determined by PISM (Sect. 3.2 – 3.4), changes in the regions of Ross and Ronne–Filchner ice shelves could thus be overestimated, while the lower thinning rates over the historical period in the Amundsen Sea Embayment could suggest a reduced sensitivity of Thwaites and Pine Island glaciers to changes in climate conditions in these experiments.*

[Figure]

(3) **Differences in ice-sheet model behaviour on shorter (multi-centennial) and longer (multi-millennial) timescales, in relation to the initial ice-sheet states**

In our experiments, we find differences in the ice-sheet model behaviour, in terms of the Antarctic sea-level contribution, on shorter (multi-centennial) and longer (multi-millennial) timescales that are most pronounced for low to intermediate levels of warming (corresponding to warming projected under SSP1-2.6, as outlined in the comment by Reviewer 1). For higher warming levels, the results from different ice-sheet models are in overall good agreement.

The higher short-term sensitivity under SSP1-2.6 in Kori-ULB compared to PISM is related to a stronger dynamical response, in particular in the Amundsen Sea sector, continuing the trends in this region over the historical period (see (2)). On multi-millennial timescales, both ice-sheet models eventually show a committed substantial grounding-line retreat in the Amundsen Sea Embayment under the lower-emission pathway SSP1-2.6.

The difference in the committed sea-level contribution (by year 7000) under SSP1-2.6 is explained by ice loss from the catchment draining Ross Ice Shelf in PISM experiments presented here with the loss of this ice shelf (resulting in a partial collapse of the West Antarctic Ice Sheet), opposed to a grounding-line advance and upstream thickening in the Siple Coast region in experiments with Kori-ULB.

We agree that the difference in the initial ice-sheet geometry could be a factor that contributes to the model difference in the Antarctic sea-level commitment under SSP1-2.6. As argued by the reviewer, the Antarctic sea-level contribution on multi-millennial timescales with substantial ice loss in some Antarctic regions could be higher in PISM than in Kori-ULB due to the over-estimation of the ice thickness in West Antarctica compared to observations (assuming a same pattern of mass loss; compare (1)). In our experiments the difference can be mainly explained by differences in the ice-sheet response in the Siple Coast described above leading to a higher Antarctic sea-level commitment in PISM than in Kori-ULB in the first place.

These model differences in the Siple Coast response can likely be linked to the initialization approaches and the simulated ice-sheet behaviour over the historical period, with (a) a drift in Kori-ULB, given lower sub-shelf melt rates obtained with PICO in this area compared to those that are obtained from the initialization approach to keep the ice sheet steady, and (b) the upstream location of the simulated grounding line compared to present-day observations (see (1), as previously seen in a model initialisation in a spin-up approach, e.g., Reese et al. (2023) and Sutter et al. (2023)) and the simulated thinning in Ross Ice Shelf over the historical period (compare (2)) in PISM. In addition, once grounding-line retreat is triggered, a collapse of the West Antarctic Ice Sheet may be more likely in PISM than in Kori-ULB, where low slipperiness towards the interior of West Antarctica (given low basal sliding coefficients retrieved in the inverse simulations, see (1)) slows down ice-sheet retreat. Stronger forcing (that is, warming levels reached by the end of this century under SSP5-8.5) is required in the Kori-ULB experiments presented here to overcome this low slipperiness towards the interior of West Antarctica and to induce a complete collapse of the West Antarctic Ice Sheet.

Beyond the low to intermediate warming levels covered by the lower emission pathway SSP1-2.6, the pattern of mass loss and the resulting sea-level contribution from Antarctica are robust across both ice-sheet models, irrespective of their initialization approach and structural differences. In particular, we find a stepwise long-term decline of the Antarctic Ice Sheet across two ice-sheet models: With increasing warming, our experiments suggest a committed partial collapse of the West Antarctic Ice Sheet, associated with substantial retreat in the Amundsen Sea Sector, up to its complete collapse, followed by enhanced mass loss from the East Antarctic marine Wilkes, Recovery and Aurora subglacial basins and an eventual decline of terrestrial parts of the ice sheet.

In the revised manuscript, we have reformulated the description of the projected multi-centennial Antarctic ice-sheet trajectories (Sect. 3.2; lines 339-393) as well as the multi-millennial committed Antarctic sea-level contribution (Sect. 3.3; lines 394-522), that now include explanations linking the ice-sheet response during the historical period, on shorter (multi-centennial) and longer (multi-millennial) timescales. In addition, Sect. 4 and, in particular, the discussion of model uncertainties (Sect. 4.2; lines 693-765) is reorganized and also elaborates on differences in ice-sheet model behaviour on shorter (multi-centennial) and longer (multi-millennial) timescales, in relation to ice-sheet modelling and initialization choices. Please see the attached manuscript for changes in the text.

Overall, we hope that by

- adding a detailed description of the outcomes of the initialization, compare (1),
- adding a paragraph outlining the simulated ice-sheet trajectories over the historical period, compare (2)

- and reformulating Section 3 (Results) and Section 4 (Discussion), relating ice-sheet modelling and initialization choices to the simulated ice-sheet behaviour during the historical period, on multi-centennial and multi-millennial timescales (see above),

the factors that may contribute to the differences in ice-sheet model behaviour on shorter (multi-centennial) and longer (multi-millennial) timescales become clearer in the revised manuscript.

**Specific comments**

**Could you provide a table of experiments run, both in terms of forcings and ice sheet model parameter values?**

We thank the reviewer for this suggestion, and agree that an overview table for the presented commitment experiments can support clarity on the experiments presented in our study. In addition to Table S1, showing the CMIP6 GCMs for deriving the changes in Antarctic climate in combination with the different branchoff points in time, we have added Table S2 in the revised Supplementary Material. Table S2 displays the ice-sheet model configurations used for assessing Antarctic sea-level commitment with Kori-ULB and PISM, see below.

Table S2. Ice-sheet model configurations. Ice-sheet model configurations used for assessing Antarctic sea-level commitment. While ice—sheet initial conditions with Kori-ULB are obtained in an inverse simulation for each of the atmospheric climatologies, it is derived from a spin–up ensemble for each atmospheric climatologies in PISM. The combinations of PISM parameters for the initial states selected from these ensembles are given as well.

|  | Atmospheric climatology | Sliding exponent | SIA enhancement factor | Tillwater decay rate | Till effective overburden fraction |
|---|---|---|---|---|---|
| **Kori-ULB** | MARv3.11 | - | - | - | - |
|  | RACMO2.3p2 | - | - | - | - |
| **PISM** | MARv3.11 | 0.75 | 2.0 | 7 mm yr$^{-1}$ | 0.015 |
|  | RACMO2.3p2 | 0.5 | 1.5 | 7 mm yr$^{-1}$ | 0.015 |

**Line 180: capitalise "Initialisation"**

Thanks. We have corrected this typo in the revised manuscript.

**Line 184-185: NorESM1-M is CMIP5? "ocean and atmosphere anomalies" refers to anomalies of this particular GCM?**

Yes. We create the historical climatologies, roughly representing the year 1950, based on a historical simulation by NorESM1-M in CMIP5 (Bentsen et al., 2013). This follows recommendations in ISMIP6, where CMIP5 NorESM1-M was found to represent the Antarctic climate over the historical period reasonably well, compared to other CMIP5 GCMs (Barthel et al., 2020; Nowicki et al., 2020).

In the revised manuscript (lines 189-197), we have adjusted the corresponding section to avoid confusion about the origin of the ocean and atmosphere anomalies. It reads as follows:

*The historical climatic boundary conditions for the year 1950 are constructed using the historical changes in ocean and atmosphere with respect to the reference period from 1995 to 2014 from the Norwegian Earth System Model (NorESM1-M; Bentsen et al.,*

*2013) in CMIP5. The oceanic and atmospheric anomalies from NorESM1-M are averaged over the period 1945–1955 and subsequently added to present–day atmospheric temperatures and precipitation derived from Regional Climate Models (RCMs) as well as observed present–day ocean temperatures and salinities. Present–day atmospheric climatologies are derived from the RCMs Modèle Atmosphérique Régional (MARv3.11; Kittel et al., 2021) and the Regional Atmospheric Climate MOdel (RACMO2.3p2; Van Wessem et al., 2018) to take into account uncertainties in the representation of present–day Antarctic surface climate (compare Mottram et al., 2021).*

**Fig 1a: Should the top part of the curve have a steeper slope? (i.e. accelerated mass loss)**

Figure 1a illustrates various factors that may contribute to the substantial difference between the transient realized and the long-term committed sea-level change, such as the potential of crossing critical temperature thresholds with progressing warming. While such critical temperature thresholds may already be crossed during the next decades or centuries, the corresponding ice loss could then unfold on multi-centennial to multi-millennial timescales.

The potential threshold behaviour of the ice sheet depending on the ice-sheet boundary conditions can be seen in its equilibrium response, indicated in black. It could be obtained by very slowly changing the environmental conditions (e.g., global mean temperature) at a rate which is much slower than the typical rates of changes in an ice sheet. Once a critical temperature is crossed, there is a stepwise change to a qualitatively different ice-sheet configuration, which is associated with a relatively higher sea-level contribution in Figure 1a. It corresponds to the loss of the lower stable equilibrium branch (solid black line) and the transition to the upper stable equilibrium branch (solid black line). In the transient ice-sheet response, for example following a warming trajectory under the higher-emission pathway, this stepwise change translates into an accelerated mass loss over time. That is, an accelerated mass loss, that is associated with the crossing of critical temperature thresholds, corresponds to the 'jump' from the lower to the higher sea-level contribution in Figure 1a.

In the revised manuscript, we have added the following sentences to the caption of Figure 1 for clarification:

> *For example, the crossing of critical thresholds with ongoing warming may result in accelerated mass loss. This is associated with the stepwise change (jump) towards a higher sea-level contribution indicated as (2).*

**Line 192: By atmospheric climatologies, do you mean from the RCMs or the CMIP forcing?**

> To build initial ice–sheet representations with PISM, a spin–up approach is applied for each of the atmospheric climatologies individually.

Atmospheric climatologies here refer to the climatologies representing the historical climate (around 1950). These are a combination of present-day atmospheric climatologies from the RCMs MAR and RACMO and historical changes in Antarctic climate derived from NorESM1-M. We thank the reviewer for pointing out this lack of clarity in the original manuscript, and have reformulated this paragraph as follows in the revised manuscript (lines 203-204):

*To build initial ice–sheet states with PISM, a spin–up approach is applied for each of the historical atmospheric climatologies (around 1950, see above) individually:*

In addition, lines 189-197 have been reformulated in the revised manuscript to clarify the construction of the historical climatologies. We refer to the response to the related previous reviewer comment.

**Line 204: So scoring for Bedmap2 for PISM, but Bedmachine for Kori-ULB?**

Yes, this is correct. We use the observed Bedmap2 ice thickness for scoring of the PISM initial ice-sheet states, while the Bedmachine present-day ice thickness is used in the initialization of Kori-ULB. Using the Bedmachine ice thickness instead of the Bedmap2 ice thickness likely does not change the scoring substantially, given the overall magnitudes of differences between the simulated ice thickness compared to observations relative to the differences between both datasets.

Note that there is no scoring after the initialization with Kori-ULB, but a nudging is performed for a given (fixed) set of ice-sheet model parameters to match the observed ice-sheet thickness. That is, the difference to the present-day ice thickness is minimized by iteratively adjusting the basal sliding coefficients under grounded ice and sub-shelf melt rates under floating ice in an inverse simulation (following Pollard and DeConto, 2012) under historical (1950) atmospheric conditions. Here, the Bedmachine present-day ice thickness from Morlighem et al. (2020) was chosen as a target.

For PISM, different possible initial ice-sheet states are obtained in a full-physics spin-up ensemble. Here, starting from the observed Bedmap2 ice thickness (Fretwell et al., 2013), the ice sheet evolves freely under historical (1950) climate. The ice-sheet state that is used for assessing the Antarctic sea-level commitment is chosen by scoring the ensemble members based on the mean-square-error mismatch of grounded and floating ice area, ice thickness, grounding-line location and surface velocity compared to present-day observations (Fretwell et al., 2013; Rignot et al., 2011).

Please also see our response to the general reviewer comment for a more detailed discussion of the application of different initialization approaches for assessing the Antarctic sea-level commitment in our study and related adjustments in the revised manuscript.

**Line 214: "balanced"**

Thanks. We have corrected this typo in the revised manuscript.

**Line 230: "GCMs"**

Thanks. We have corrected this typo in the revised manuscript.

**Line 240-243: Do you average at or over a particular depth?**

> Missing values for the oceanic forcing on the continental shelf (arising due to the coarse resolution of CMIP6 GCMs) and in currently ice-covered regions are filled following Kreuzer et al. (2021), i.e., by averaging over all existing values in neighbouring cells.

We thank the reviewer for pointing us to missing information on the processing of ocean forcing in this part of the original manuscript. Filling of missing values is done for every available

ocean layer of the different GCMs. In a next step, and to have forcing fields that are applicable to PICO, ocean properties derived from CMIP6 GCMs are linearly interpolated to the continental shelf depth (compare line 266 in the original manuscript).

In the revised manuscript, we have moved the remark on the linear interpolation to lines 276-277.

**Line 270: Is the reason for the difference in parameter values that they produce better fit to observations for those particular ice sheet models?**

The values of the PICO parameters are an individual choice for each ice-sheet model. They have been chosen such that for the respective ice-sheet model observed sub-shelf melt sensitivities and / or melt rates are matched, and are based on parameter optimizations for PICO (Reese et al., 2018; Reese et al., 2023).

In the revised manuscript (lines 301-302), a short explanation has been added as follows:

> *The values of the PICO overturning strength parameter C and the turbulent heat exchange coefficient $\gamma_T^*$ are an individual choice for each ice–sheet model to match sub–shelf melt sensitivities and / or observed melt rates.*

**Line 278: Is the negative SLE change from a model that has positive SLE change over the historical period? It is worth specifying if the models that show modern ice mass loss have a negative or positive SL contribution.**

> Following the lower–emission pathway SSP1-2.6 results in a sea–level change ranging from -5.0 cm to +8.0 cm by the end of this century and from -0.2 m to +0.5 m in 2300 (Fig. 2a; Tab. 1). Therein, Kori-ULB projects a positive sea–level contribution for this lower–emission scenario (dashed lines), while PISM projects a sea–level drop (solid lines).

We thank the reviewer for this important question. We agree that stressing the relation between the simulated historical ice-sheet trajectories in Kori-ULB and PISM and the projected sea-level contribution on multi-centennial to multi-millennial timescales in the revised manuscript would be helpful.

Over the historical period in response to NorESM1-M climate trajectories, Kori-ULB ice-sheet trajectories show an integrated mass loss, while the Antarctic Ice Sheet slightly gains mass in the PISM experiments presented here. The simulated trends of ice-sheet changes over the historical period are continued in future projections under SSP1-2.6 with both ice-sheet models. That is, PISM projects a sea-level drop by 2300 compared to present-day for the majority of lower-emission climate trajectories. Kori-ULB projects a positive sea-level contribution for this lower-emission scenario, related to a stronger dynamical response in the Amundsen sea sector. Ice-sheet trajectories under SSP1-2.6 are thus influenced by the simulated historical trends and differences in ice-sheet modelling choices. Under the higher-emission pathway SSP5-8.5, we find that climate drivers dominate the projected multi-centennial ice-sheet changes.

Please see our response to the general reviewer comment for a more detailed discussion of the simulated historical ice-sheet trajectories and the ice-sheet response on multi-centennial timescales as well as related adjustments in the revised manuscript.

**Line 280: I would think the difference in initial state is a larger contributor to the differences between the two models than the dynamic response to the forcings.**

> The overall sign of ice–sheet mass changes contributing to a change in sea–level depends on the balance between the dynamic response to sub–shelf melting and ice–shelf thinning and the surface mass balance. We find that the integrated surface mass balance remains positive for both ice–sheet models until 2300 under SSP1-2.6 (Fig. S3a). However, the response in dynamic discharge contributing to a sea–level increase on centennial timescales is higher in Kori-ULB (with ice–sheet thinning in the Amundsen Sea Embayment extending inland, Fig. S4–S5) than in PISM (see Fig. S6–S7 for comparison), explaining the diverging sea–level contribution under SSP1-2.6 until 2300.

We agree that the difference in the initial ice-sheet states may be one factor that contributes to the model difference in the projected Antarctic sea-level contribution. Please compare our response to the general reviewer comment for a more detailed discussion of the initial ice-sheet states that result from the different initialization approaches, their impact on the projected transient sea-level contribution from the Antarctic Ice Sheet to 2300 and related adjustments in the revised manuscript.

**Line 284-286: PISM is too thick in the ASE to start with.**

Given the spin-up approach applied in PISM, the initial ice-sheet states are characterized by larger ice-thickness differences compared to present-day observations. Please compare our response to the general reviewer comment for a more detailed discussion of the initial ice-sheet states that result from the different initialization approaches, their impact on the ice-sheet model behaviour on shorter (multi-centennial) and longer (multi-millennial) timescales and related adjustments in the revised manuscript.

**Line 297-300: The models don't include hydrofracture parameterization, correct? Does surface temperature of ice shelves reach threshold for melt pond formation? e.g. van Wessem et al. (2023)**

This is correct. In the experiments presented here, a hydrofracture parameterization is not applied.

The formation of melt ponds and subsequent hydrofracturing with future warming and increasing surface melt are discussed as precursors for ice-shelf loss or collapse (e.g., Lai et al., 2020; Pollard et al., 2015; Trusel et al., 2015). Van Wessem et al. (2023) identify temperature thresholds for melt pond formation in Antarctica. For mild and wet ice shelves such as in West Antarctica and the western Antarctic Peninsula, melt pond formation is suggested to occur for temperatures higher than -9°C. For cold and dry ice shelves, this threshold is estimated around -12 °C or even less than -15°C for Ronne-Filchner, Ross and Amery ice shelves (Van Wessem et al., 2023).

When comparing to future warming projected within CMIP6 by 2100, these thresholds for melt pond formation may be crossed for many ice shelves, in particular under SSP5-8.5 (as shown by Van Wessem et al., 2023). In our experiments, we follow the warming trajectories projected by four CMIP6 GCMs even beyond the end of the century until 2300. Consistent with Van Wessem et al. (2023), these GMCs project atmospheric temperature changes exceeding the melt pond formation thresholds under SSP5-8.5. Temperature changes due to an evolving

ice-sheet geometry by means of the atmospheric lapse rate might add to the warming projected by the GCMs.

So far, hydrofracturing is poorly represented in ice-sheet models and the availability of parameterizations is limited. One parameterization for including this process in sea-level projections has been proposed by Pollard et al. (2015) and DeConto and Pollard (2016). In addition, ice-shelf collapse, that may be caused by, among others, hydrofracturing, is prescribed by a yearly mask defining regions and timing of collapse based on the presence of mean annual surface melting above 725mm over a decade in ISMIP6 (Trusel et al., 2015; Seroussi et al., 2020).

Coulon et al. (2024) tested the sensitivity of the Antarctic sea-level contribution by the end of the millennium to hydrofracturing based on the parameterization of Pollard et al. (2015) and DeConto and Pollard (2016): In their sea-level projections, the rate of the Antarctic contribution to global mean sea-level rise is increased compared to simulations that do not account for ice-shelf collapse through hydrofracturing. This is due to an acceleration of grounding-line retreat in West Antarctica as well as in the marine basins of the East Antarctic Ice Sheet as a consequence of accelerated ice-shelf breakup.

In the revised manuscript, we have added a paragraph that discusses the potential consequences of hydrofracture for Antarctic sea-level commitment in Section 4.3 (lines 785-792). It reads as:

> *Surface melt on Antarctic ice shelves facilitates hydrofracturing and may, thereby, trigger ice–shelf collapse (Van Wessem et al., 2023; Lai et al., 2020; Pollard et al., 2015; Trusel et al., 2015) and potentially the Marine Ice Cliff Instability (MICI; Bassis and Walker, 2012; Pollard et al., 2015). While temperature thresholds for melt pond formation as a precursor for such ice–shelf loss may be exceeded by the end of the century (Van Wessem et al., 2023), the availability of parameterizations to include these processes in ice–sheet models is still limited (Pollard et al., 2015; DeConto and Pollard, 2016; Seroussi et al., 2020). Considering hydrofracturing (following Pollard et al., 2015; DeConto and Pollard, 2016; Seroussi et al., 2020) may speed–up grounding-line retreat in marine Antarctic basins due to an earlier ice-shelf breakup (Coulon et al., 2024; Seroussi et al., 2020).*

**Line 303: "(ISMIP6)"**

In the revised manuscript, the abbreviation has been introduced as:

> *Ice Sheet Model Intercomparison Project for CMIP6 (ISMIP6)*

**Line 315: But you have some simulations that show the opposite**

> Our simulations confirm that sea level may keep rising for centuries to millennia to come even if warming is kept at a constant level (Fig. 2c and d, consistent with, e.g., Winkelmann et al., 2015; Van Breedam et al., 2020).

This is correct, thanks for pointing this out. While the bulk of our simulations show that sea level may keep rising for centuries to millennia to come even if warming is kept at a constant level, there are some ice-sheet trajectories showing a decline in the Antarctic sea-level contribution towards the year 7000 after an initial positive sea-level change. This is most pronounced in simulations when following MRI-ESM2-0 climate under SSP1-2.6, and may be attributed to a thickening trend upstream of Ross Ice Shelf (in Kori-ULB only) and in the icesheet interior towards the year 7000, outweighing the initial Antarctic mass loss. Note that despite the decline of sea-level rise for some ice-sheet trajectories, these are still character-ised by an initial sharp increase in the Antarctic sea-level contribution.

In the revised manuscript, we have reformulated the description of the committed Antarctic sea-level contribution under SSP1-2.6 (Sect. 3.3.1; lines 408-476). Please see the attached manuscript for changes in the text. In particular, the related paragraph (lines 401-402 and lines 410-413) now reads as:

> *The bulk of our simulations shows that sea level may keep rising for centuries to mil-lennia to come even if warming is kept at a constant level (Fig. 5a – d; consistent with, e.g., Winkelmann et al., 2015; Van Breedam et al., 2020).*
> *[...]*
> *Some of the ice–sheet trajectories eventually show a decline in the Antarctic sea–level contribution on multi–millennial timescales (Fig. 5a and c; e.g., for MRI-ESM2-0 climate indicated in orange), with a thickening trend upstream of Ross Ice Shelf (in Kori-ULB only, see below) and in the ice–sheet interior towards the year 7000, out-weighting the initial mass loss.*

In addition, we have added the following explanation for the decline in the Antarctic sea-level contribution in lines 442-449 of the revised manuscript:

> *In Kori-ULB, both large ice shelves are preserved to year 7000, and we find a ground-ing–line advance and upstream thickening in the Siple Coast region (Fig. 6a). This long–term ice–sheet response in the Siple Coast may, in parts, result from a drift of the initialisation procedure, given lower sub–shelf melt rates obtained with PICO in this area compared to those that are obtained from the initialization approach to keep the ice sheet steady (Sect. 2.2.2). A thickening signal upstream of Ross Ice Shelf has also been observed over the past decades (with the stagnation of Kamb Ice Stream; Smith et al., 2020). The simulated thickening upstream of Ross Ice Shelf contributes to the decay in the long–term Antarctic sea–level contribution over time after the year 3000 in some Kori-ULB experiments, which is most pronounced for sustained MRI-ESM2-0 climate (Fig. 5a, orange).*

The related Figure 2 in the original manuscript has also been changed following an editor comment. In particular, a figure focusing on the committed ice-sheet response has been intro-duced as Figure 5 in the revised manuscript. We here show the multi-millennial ice-sheet re-sponse in terms of the sea-level contribution under SSP1-2.6 and SSP5-8.5, separately for Kori-ULB and PISM, together with the sea-level commitment in the year 7000, depending on the branchoff point in time. We hope that, with this separation of the long-term ice-sheet tra-jectories by the ice-sheet models, simulations can be better distinguished.

[Figure]

**Line 339-342: This sentence is confusing.**

> In PISM, a substantial portion of the marine ice–sheet in West Antarctica is lost by year 7000 under most considered climate trajectories (compare Fig. S14–S15), determining (in combination with a potential grounding–line retreat in the Wilkes basin) the upper range of Antarctic sea–level commitment under SSP1-2.6 in our ensemble of simulations (Fig. 2e).

Lines 339-342 in the original manuscript refer to the committed ice-sheet response simulated by PISM under SSP1-2.6. Here, the combined ice loss from West Antarctica and the East Antarctic Wilkes basin (especially for UKESM1-0-LL climate) gives rise to the upper end of the long-term Antarctic sea-level commitment of up to 6.5 m under this lower-emission pathway in our ensemble of simulations.

In the revised manuscript, we have reformulated the description of the committed ice-sheet changes under SSP1-2.6 (Sect. 3.3.1, lines 408-476) and we hope that it is clearer in its revised form. Please see the attached manuscript for changes in the text. In particular, a related sentence (lines 463-466) now reads as:

> *The combined ice loss from West Antarctica and the East Antarctic Wilkes subglacial basin in PISM gives rise to the upper end of the long–term Antarctic sea–level commitment of up to +6.5 m found under the lower–emission pathway in our experiments (Fig. 5c and e, grey open markers).*

**Figure 2: I suggest changing the colors because SSP1-2.6 is generally dark blue, and SSP5-8.5 is generally dark red, but here the colors refer to different GCMs.**

Thanks for this suggestion. We have changed the colours indicating the GCMs in Figure 2 and elsewhere in the figures of the revised manuscript.

**Fig 3: Clarify the time of commitment. Are these means of both models?**

In the revised manuscript, we have made sure that the time of commitment is given in all figure captions where needed.

**Line 391: "changes"**

Thanks. We have corrected this typo in the revised manuscript.

**Line 399: By regional warming, do you mean of air temperature?**

Yes. Warming levels are given as regional Antarctic-averaged atmospheric temperature changes, compared to 1995-2014. Thanks for pointing out that this information is missing in line 399 of the original manuscript. In the revised manuscript, we have made sure that this information is clearly stated where needed.

**Line 406: Is this realistic that the Ross Ice Shelf doesn't collapse under sustained SSP5 forcing? It could be an artifact of the basal melt parameterization, or the fact that there is no hydrofracture parameterization implemented.**

> While Ross Ice Shelf is maintained in simulations with Kori-ULB, it is lost in most simulations with PISM (compare Fig. 4c and d, I).

In experiments with Kori-ULB, Ross Ice Shelf does not collapse when sustaining the warming that is reached by the year 2050 under the SSP5-8.5 emission pathway and for warming levels under the lower-emission pathway SSP1-2.6. Please note, however, that it may be lost on multi-millennial timescales for stronger warming levels that are reached later in time under SSP5-8.5 (compare Figures S16 and S17 of the original Supplementary Material).

These differences in the timing of ice-shelf collapse may be related to the calving schemes employed in the ice-sheet models or different sub-shelf melt sensitivities to changes in ocean temperature in PICO. The values of the PICO parameters are an individual choice for each ice-sheet model. They have been chosen such that for the respective ice-sheet model observed sub-shelf melt sensitivities and / or melt rates are matched, and are based on parameter optimizations for PICO (Reese et al., 2018; Reese et al., 2023).

In the revised manuscript (lines 301-302), we have added an explanation on the choice of PICO parameters, following a previous reviewer comment:

> *The values of the PICO overturning strength parameter C and the turbulent heat exchange coefficient $\gamma_T$\* are an individual choice for each ice–sheet model to match sub–shelf melt sensitivities and / or observed melt rates.*

Note that overall the availability of projections of the evolution of the Antarctic Ice Sheet after 2100 is limited. In particular, substantial parametric uncertainty exists, some of which (e.g., basal melt parameterizations and related parametric uncertainty) is explored in more detail in Coulon et al. (2024) in terms of the Antarctic sea-level contribution by the end of this millennium. It thus remains an important next step for future research to assess the effects of this

parametric uncertainty on the Antarctic Ice Sheet response (including the potential loss of ice shelves) also on multi-millennial timescales as discussed e.g. in lines 594-605 of the original manuscript / lines 744-758 in the revised manuscript.

While not including hydrofracturing as a process in our assessment of the Antarctic sea-level commitment, potential consequences, e.g. via an earlier ice-shelf collapse, are discussed in the revised manuscript, in an additional paragraph in Section 4.3 (lines 785-792). Please also see our response to the previous related reviewer comment for a more detailed discussion of hydrofracturing and related adjustments in the revised manuscript.

**Line 409: For how much warming? And is this due to the initial condition (i.e. PISM has more of WAIS to lose…)?**

> While 2% of the initial ice mass in Antarctica contributing to global mean sea–level rise is lost in Kori-ULB (raising global mean sea–level by up to approximately +2.0 m), a slightly higher fraction of 6% (equivalent to a global mean sea–level change between +3.0 m and +4.0 m) is found in simulations with PISM (Fig. 4b), due to the discharge of larger parts of the West Antarctic Ice Sheet as opposed to an advance of the grounding line in the Siple Coast area in Kori-ULB (compare Fig. 4c and d, I).

The given amount of ice is lost for a regional Antarctic-averaged atmospheric warming below 4°C in our experiments. Note that, in the revised manuscript, we have reformulated the description of the committed Antarctic sea-level contribution under SSP1-2.6 (Sect. 3.3.1; lines 408-476) to improve clarity. Please see the attached manuscript for changes in the text.

We agree that the difference in the initial ice-sheet state may be one factor that contributes to the model difference in Antarctic sea-level commitment. As argued by the reviewer, the sea-level contribution could be higher in PISM than in Kori-ULB due to the overestimation of the ice thickness in West Antarctica compared to observations (assuming a same pattern of mass loss).

We here, however, also find some difference in the pattern of mass loss, with ice loss from the catchment draining Ross Ice Shelf in PISM, opposed to grounding-line advance and upstream thickening in this region in experiments with Kori-ULB. In addition, the subglacial basin draining Ronne Ice Shelf shows more mass loss in the PISM experiments presented here than in Kori-ULB experiments. This may explain the differences in the long-term sea-level contribution under SSP1-2.6 as determined by PISM and Kori-ULB in the first place.

These differences can likely be linked to the different initialization approaches applied in the ice-sheet models. Please compare our response to the general reviewer comment for a more detailed discussion of the initial ice-sheet states that result from the different initialization approaches, their impact on the ice-sheet model behaviour on shorter (multi-centennial) and longer (multi-millennial) timescales and related adjustments in the revised manuscript.

**Line 413: What do you mean by "uncertainty"?**

> The uncertainty in the initial ice–sheet configurations of each model results in differences on the order of decimeters in sea–level contribution in this warming range and is thus less significant than the inter–model spread.

We here aimed to quantify the difference in the Antarctic sea-level contribution that is caused by using distinct initial ice-sheet states for each ice-sheet model. This is sometimes referred

to as 'initial-state uncertainty'. In our experiments and related to the initialization approaches, differences in initial ice-sheet states are, however, not only related to different initial ice-sheet geometries, but at the same time basal sliding coefficients (in Kori-ULB, as obtained in the inverse simulation under historical climate) or ice-sheet model parameters (in PISM, as a result of the spin-up ensemble under historical climate) may be different across initial ice-sheet states and could influence the ice-sheet response to changes in climate.

In the revised manuscript, we have reformulated the description of the committed Antarctic sea-level contribution under SSP1-2.6 (Sect. 3.3.1; lines 408-476) to improve clarity. Please see the attached manuscript for changes in the text.

**Line 477: The projection experiments are consistent, but the initialisations are not.**

> Experiments were carried out systematically for stabilized climate at different points in time over the course of the next centuries and in a consistent way with the stand–alone ice–sheet models PISM and Kori-ULB accounting for some inter– and intra–model uncertainty.

Two different initialization approaches are included in our work, based on the state-of-the-art ice-sheet models Kori-ULB and PISM. We do not consider the application of these different initialization approaches as inconsistency in our experimental setup, but rather as an advantage. Given that we include two common ways of initializing ice-sheet models (compare e.g., Seroussi et al., 2019, 2020), we sample uncertainties associated with the choice of the initialization approach. Please compare our response to the general reviewer comment for a more detailed discussion of the application of different initialization approaches for assessing the Antarctic sea-level commitment and related adjustments in the revised manuscript.

**Line 567-593: Initial state discussion focuses on the choice of present atmospheric forcing, but what about the initialisation procedure, which seems to result in quite different ice thicknesses?**

We agree that the difference in the initial ice-sheet geometry may be one factor that contributes to the model difference in Antarctic sea-level commitment. As argued by the reviewer, the sea-level contribution could be higher in PISM than in Kori-ULB despite the same pattern of mass loss due to the overestimation of the ice thickness in West Antarctica compared to observations. We here, however, also find some difference in the pattern of mass loss, with ice loss from the catchment draining Ross Ice Shelf in PISM, opposed to the grounding-line advance and upstream thickening in this region in experiments with Kori-ULB.

Please compare our response to the general reviewer comment for a more detailed discussion of the initial ice-sheet states that result from the different initialization approaches, their impact on the ice-sheet model behaviour on shorter (multi-centennial) and longer (multi-millennial) timescales and related adjustments in the revised manuscript.

**I suggest that you reduce the number of supplemental figures by consolidating Fig S4 to Fig S19.**

We thank the reviewer for this suggestion. In the revised version of the Supplementary Material, we have consolidated Figure S4 - Figure S19 of the original Supplementary Material as Figure S2 and Figure S3.

[Figure]

*Editor comment*

**The paper by Klose et al. address the multi-millennial sea level commitment of the Antarctic ice sheet using two different ice sheet models forced by a set of four coupled climate simulations from the CMIP6 initiative. An anomaly method is used to create the climate forcing based on regional atmospheric models MAR and RACMO on which GCM future climate is added. To estimate the long-term sea level commitment corresponding to different level of global mean atmospheric warming, the simulations are branched off at different moments with the next two centuries and climatic conditions are maintained constant at their branching-off level.**

**The scientific content of the manuscript is good and very interesting. This is something that is needed for different reasons: testing the physics of ice sheet models and parameterizations, pushing to obtain ling-term multi-centennial climate forcing etc… I really liked the manuscript. However I can feel that this is perhaps the first or on of the first article written by the first author here: the writing of the manuscript needs some substantial work to be clear an readable. Information are some times provided in a very messy way, spread out in different sub sections etc…In addition I feel that the description of the results is sometimes approximative and also messy amongst the two ice sheet models. The authors should consider describing everything in depth with one model and then describing the discrepancies with the other model. The discussion (from Uncertainties to boundary conditions) is also chaotic and does not allow the reader to really appreciate the real advance of the work. I below provide some generic comments, but most of the specific comments can be found in the attached commented pdf version of the main manuscript.**

We are grateful for the overall positive evaluation of this work. We thank the editor for carefully reading our manuscript and providing us with helpful comments, in particular, with regard to the description of our results and figures.

**General comments**

**The description of the result is too "descriptive" and many times, no real explanation is provided for some observed behavior, or really little. Some of them are explained further int eh discussion, some others not at all. The results are sometimes described in a very approximative way. Sentences are sometime useless because not bringing any substantial info. In general try to group the info related to one topic or one model together. Right now, the reader needs to jump from one paragraph to another to synthesise all the info about one process or one model.**

We would like to thank the editor for the effort of creating detailed comments on our manuscript. In the revised manuscript, we have reformulated the description of the results in Section 3, also following the specific and very helpful suggestions and/or questions of the editor in the commented version of the main manuscript (see below) and the reviewer comments (see above).

The changes in the revised manuscript include

- the addition of a section on the simulated historical ice-sheet evolution (Sect. 3.1, lines 313-338)
- a reformulation of the description of the transient ice-sheet response under SSP1-2.6 and SSP5-8.5 (Sect. 3.2, lines 339-393)
- a reformulation of the description of the committed ice-sheet changes under SSP1-2.6 (Sect. 3.3.1, lines 408-476) and SSP8-5.8 (Sect. 3.3.2, lines 477-522)

Overall, results are grouped as follows in the revised manuscript: In general, committed ice-sheet changes are assessed depending on the branchoff point in time for each emission pathway (Sect. 3.3) and summarized depending on the climatic boundary conditions, to overcome the dependency on the diverging climate trajectories (Sect. 3.4). Within each section, changes that are consistent across ice-sheet models are introduced. Uncertainties in the Antarctic Ice Sheet response related to ice-sheet modelling choices are outlined afterwards. Throughout the manuscript, we aimed for a good balance between giving possible explanations for the simulated ice-sheet response and relating to previous research directly with the description of the results or later in the discussion (to avoid repetition).

We would also like to refer to our responses to the specific comments by the editor below and to the attached manuscript for changes in the text.

**I generally find the climate analysis a bit weak, given that the paper looks at sea level commitment. I would have expected a bit more climate analysis to really show the relationship between the different steps of the retreats and the competition between atmospheric warming induced melting and basal melting from oceanic warming. For example Figure S3 to my opinion should be inserted within the main manuscript and with two additional panels showing atmospheric warming and oceanic warming evolution through time. Although it is a bit complicated for the oceanic warming since it depends very much on the sector of Antarctica.**

**We never see one figure of climate forcing and this is instead very important since the climate forcing here plays a critical role in all the results. Thus I expect to see a bit more of climate in terms of figures and forcing description. That will allow the authors and the reader to better described and understand the results.**

We thank the editor for this suggestion. In the revised manuscript, we have included an additional figure as Figure 4, based on Figure S3 in the original manuscript, showing (1) projected atmospheric and ocean warming as well as (2) projected Antarctic mass balance components, depending on the CMIP6 GMCs and emission pathways. Figure 4 and the related analysis of the projected Antarctic climate is integrated in the revised description of the results (see previous editor comment), in particular in the presentation of the projected transient ice-sheet response until 2300 (Sect. 3.2, lines 339-393). For example, we find that climate drivers dominate the projected multi-centennial ice-sheet changes under the higher-emission pathway SSP5-8.5. In line with Coulon et al. (2024), our projections indicate that the atmosphere becomes an amplifying driver of Antarctic mass loss beyond the end of this century, irrespective of the ice-sheet model. Please see the attached manuscript for changes in the text.

[Figure]

**There is no real description of the outcomes of the initialization. There is only one sentence stating that ice sheet models reproduce correctly the AIS geometry- Which is not true, since PISM is far from having the GL in the right place, especially for the big ice shelves that are extensively then discussed in the rest of the manuscript. As stated in the discussion, PISM large sensitivity and large retreat of the WAIS is likely due to the already retreated grounding line. PISM performance is not very good in general because the final elevation differs quite a lot from the observed one and the discrepancies fall in the range of what is observed int terms of elevation changes by satellites. In addition there is no description of the historical run at all. This is also an important part to be added.**

We thank the editor for this important comment and agree that the manuscript would benefit from a full description of the outcomes of the initialization and the simulated ice-sheet trajectories over the historical period. Please also see our response to the general reviewer comment for a more detailed discussion of the different initialization approaches, a comparison of

the ice-sheet initial states in our study to observations, and the simulated ice-sheet trajectories over the historical period.

In line with the general reviewer comment, we have included a discussion of these initialization approaches and their advantages as well as a comparison of the initial ice-sheet states in our study to observations in Section 2.2.2 of the revised manuscript (lines 237-255):

> *Given that we include two common ways of initializing ice-sheet models (compare e.g., Seroussi et al., 2019, 2020), we sample uncertainties associated with the choice of the initialization approach. While an inverse simulation allows to reproduce the present–day ice sheet geometry well, the resulting parameter fields (such as basal sliding co-efficients in Kori-ULB) may compensate for errors or uncertainties in other ice–sheet processes (Berends et al., 2023b; Aschwanden et al., 2013). In addition, it is assumed that the field obtained in the inverse simulation to match present–day observations does not change in the future. In contrast, in the simulated ice–sheet state resulting from a spin-up the ice–sheet variables may be modelled in a consistent way, but its geometry might differ from the observed ice sheet. It is the result of the covered ice–sheet physics in the ice–sheet model for a set of uncertain parameters, without any nudging.*

> *The simulated grounding–line position and ice thickness of the initial ice–sheet states are compared to present–day observations in Figure S1. As a result of the inverse simulation, the grounding–line position and ice thickness compare well to present–day observations in the initial ice–sheet states for Kori-ULB (Fig. S1a and c). With the spin–up approach applied in PISM, the initial ice–sheet states are characterized by larger ice thickness differences compared to present–day observations (Fig. S1b and d). Overall, ice in West Antarctica and in some coastal regions in East Antarctica (e.g,. in Dronning Maud Land, upstream of Amery Ice Shelf and in Wilkes Land) is thicker than observed at present (comparable to Reese et al., 2023), while the ice thickness in the interior of East Antarctica is underestimated. In addition, the grounding line in the Siple Coast area (and in the catchment draining Ronne–Filchner Ice Shelf for the MAR cli-matology) is located upstream of the observed grounding line in the present–day (Fig. S1 b and d), as previously seen in a model initialisation in a spin–up approach, e.g., Reese et al. (2023) and Sutter et al. (2023). These differences should be taken into account when interpreting the simulated long–term evolution of the Antarctic Ice Sheet.*

As an outcome related to these initialization approaches, basal friction deviates spatially be-tween both ice-sheet models, in particular in the interior of West Antarctic marine basins. This can also be expected to influence the ice-sheet response and its timescales. In Section 2.2.2 of the revised manuscript, a paragraph describing the optimized field of basal sliding coeffi-cients in Kori-ULB in comparison to the parameterized material properties of subglacial till in PISM has been added (lines 225-230) as follows:

> *The optimized field of basal sliding coefficients in Kori-ULB is characterized by high basal sliding coefficients at the ice–sheet margins, turning into regions of low slipper-iness (low basal sliding coefficients) towards the interior of West Antarctica. It thus differs from the basal friction experienced by the Antarctic Ice Sheet in experiments with PISM, where overall slippery bed conditions in the interior of marine subglacial*

*basins are found, given the parameterized, bed–elevation dependent material proper-*
*ties of the subglacial till (in particular, the till friction angle; Sect. 2.1). These inter–*
*model differences in basal friction linked to the applied initialization approaches are*
*expected to influence the ice–sheet response.*

We have also (1) added an assessment of the simulated ice-sheet trajectories in response to
the NorESM1-M climate trajectory over the historical period as Section 3.1, (2) included a
related figure as Figure 2, and (3) linked the simulated historical ice-sheet response to the
projected ice-sheet response to 2300 (Sect. 3.2) in the revised manuscript. The main related
paragraph in the revised manuscript (lines 314-338) reads as:

*The pattern of observed present–day rates of ice–thickness change (e.g., Smith et al.,*
*2020) is overall captured by both ice–sheet models in response to the historical*
*NorESM1-M climate trajectory (Fig. 2a – c), with a thinning in the Amundsen and Bel-*
*lingshausen Sea region and the Antarctic Peninsula and a thickening in the ice–sheet*
*interior. The magnitude of ice–sheet thinning in the Amundsen Sea Embayment is,*
*however, underestimated compared to present–day observations in the historical sim-*
*ulations with PISM presented here (Fig. 2a and c). In addition, we find ice loss for*
*Ross, Ronne–Filchner and Amery ice shelves in PISM in contrast to observations (Fig.*
*2a and c).*

*The evolution of the continent–wide integrated surface mass balance is relatively sim-*
*ilar for both ice–sheet models, but occurs on a higher, though still within RCM uncer-*
*tainties, level in PISM than in Kori-ULB (Fig. 2d). While sub–shelf melt increases in*
*PISM from about 300 Gt yr−1 in 1950 towards 1100 Gt yr−1 in 2015 at the lower end*
*of present–day observations (Fig. 2e, solid lines), the basal mass balance is on the*
*order of the observational record in Kori-ULB over the entire historical period, slightly*
*exceeding its upper end in 2015 with about 1800 Gt yr−1 (Fig. 2e, dashed lines). The*
*continent–wide aggregated sub–shelf melt rates observed in present–day are thus re-*
*produced with both sets of PICO parameters (see Sect. 2.2.3), but they result in differ-*
*ent sensitivities of sub–shelf melt rates to ocean temperature changes over the histor-*
*ical period (Fig. 2e; Reese et al., 2023).*

*Mass loss in the Amundsen Sea sector dominates the overall observed ice sheet mass*
*changes in Antarctica to date (Otosaka et al., 2023). Given the lower magnitude of ice–*
*sheet thinning of Pine Island and Thwaites glaciers in PISM, and stronger sub–shelf*
*melt in Kori-ULB, we find diverging ice–sheet trajectories with both ice–sheet models*
*in terms of the Antarctic sea–level contribution over the historical period from 1950 to*
*2015: Kori-ULB shows an integrated mass loss with a sea–level contribution of about*
*+4 mm in 2015 (Fig. 2f, dashed lines), while the ice sheet overall gains mass equivalent*
*to a sea–level change ranging between -4 mm and -6 mm in PISM (Fig. 2f, solid lines;*
*within spread of recent ensemble of historical ice–sheet trajectories, Reese et al.,*
*2023).*

*In the future evolution of the Antarctic Ice Sheet determined by PISM (Sect. 3.2 – 3.4),*
*changes in the regions of Ross and Ronne–Filchner ice shelves could thus be overes-*
*timated, while the lower thinning rates over the historical period in the Amundsen Sea*
*Embayment could suggest a reduced sensitivity of Thwaites and Pine Island glaciers*
*to changes in climate conditions in these experiments.*

[Figure]

**To really appreciate the full description of the results, you also need to introduce a map, as first Figure of the manuscript show the different drainage basins, ice shelves and related names. You only show this in Fig3a, and honestly, it is so hard to understand even when printed.**

We agree that a more detailed figure showing the different drainage basins and ice shelves relevant for our study may be helpful for the reader and thank the editor for this suggestion.

In the revised manuscript, we have added such a figure as a panel in Figure 2 (compare panel a in previous editor comment).

**The figures are too dense, e.g. Fig 2., it would benefit from separating each ice sheet model simulations in different frames to better appreciate the difference and understand them.**

We thank the editor for the comments and suggestions for improving the figures in our manuscript. In the revised manuscript, we have made the following adjustments:

- Figure 2 has been reorganized, and an additional figure has been introduced. Figure 3 in the revised manuscript shows the transient ice-sheet response to 2300, in combination with the projected ice-sheet changes (in terms of the ice thickness) in the years 2050, 2100 and 2300. The committed ice-sheet response is the focus of Figure 5 in the revised manuscript. We here show the multi-millennial ice-sheet response in terms of the Antarctic sea-level contribution under SSP1-2.6 and SSP5-8.5, separately for Kori-ULB and PISM, together with the sea-level commitment in the year 7000, depending on the (branchoff) point in time.
- Figure 5 in the original manuscript has been modified as Figure 8 in the revised manuscript in terms of the numbers of displayed basins and the colouring. An additional Figure S4 in the Supplementary Material has been added to show the dependence of the committed ice loss from selected basins on changes in the ocean.

Please also see our response to the related specific editor comments below.

**The section about "Intra and inter-models uncertainties" is not useful in its present form. It would be better to divide it in several sections. Eg: "Initialization" (actually once again, the info about the impact of initialization are spread out through the section and is thus a bit messy), "model physics", etc… it would help organising a bit this part.**

In the revised manuscript, Section 4 and, in particular, the discussion of model uncertainties (including uncertainties in ice-sheet processes, their parameterization in ice-sheet models and distinct initialization approaches; Sect. 4.2; lines 693-765) has been reorganized and also elaborates on differences in ice-sheet model behaviour on shorter (multi-centennial) and longer (multi-millennial) timescales, in relation to ice-sheet modelling and initialization choices. Please see the attached manuscript for changes in the text.

**Specific comments: see the attached pdf.**

**Line 25: "presumed inception". In the paleo-antarctic community, this is the terminology that we agreed to employ.**

> The Antarctic Ice Sheet has experienced changing environmental conditions on various timescales from decadal to orbital-scale climate variability since its inception at the Eocene–Oligocene transition about 34 Mry ago (Zachos et al., 2001; DeConto and Pollard, 2003).

We thank the editor for this remark and have adjusted the revised manuscript (lines 24-26) accordingly.

**Line 28: Not happy with this modeling ref here of Pollard and Deconto. It would be better to remove it and only let observation based ref.**

We have removed this modelling reference in the revised manuscript.

**Line 28: "terrestrial parts" is more correct in general for the meaning of the sentence.**

While large parts of the terrestrial East Antarctic Ice Sheet have persisted for millions of years (Sugden et al., 1995; Shakun et al., 2018), ice–sheet variability involved an occasional collapse of the West Antarctic Ice Sheet (Naish et al., 2009) and inward migration of ice–sheet margins in marine–based sectors of East Antarctica during Pliocene warm periods (Cook et al., 2013; Patterson et al., 2014; Aitken et al., 2016).

We have changed this formulation in the revised manuscript (lines 28-32), following the editor's suggestion.

**Line 28: remove "terrestrial" and move it just before. see previous comment.**

We have changed this formulation in the revised manuscript (lines 28-32), following the editor's suggestion.

**Line 30: I think that is would be nice to define "marine-based sectors", because it is usefull further on in the text. Something like: (i.e., where the ice sheet grounds below sea level).**

We thank the editor for suggesting this addition and have adjusted the revised manuscript (lines 28-32) accordingly.

**Line 31: and also during some interglacials of the Pleistocene (e.g. Stokes et al., 2022 for a review or some of the references you cite in the next sentence.).**

While large parts of the terrestrial East Antarctic Ice Sheet have persisted for millions of years (Sugden et al., 1995; Shakun et al., 2018), ice–sheet variability involved an occasional collapse of the West Antarctic Ice Sheet (Naish et al., 2009) and inward migration of ice–sheet margins in marine–based sectors of East Antarctica during Pliocene warm periods (Cook et al., 2013; Patterson et al., 2014; Aitken et al., 2016).

In this sentence, we aim to outline the Antarctic Ice Sheet changes during Pliocene warm periods. As stated by the editor, an inward migration of ice-sheet margins in marine-based sectors of East Antarctica has also been suggested during some Pleistocene Interglacials (e.g., Wilson et al., 2018; Blackburn et al., 2020; Turney et al., 2020). This is addressed in the following sentence, and in line with the following editor comment, this sentence has been reformulated in the revised manuscript (lines 32-35) as:

*During Pleistocene Interglacials, Antarctic ice loss from the East Antarctic Wilkes subglacial basin (Wilson et al., 2018; Blackburn et al., 2020) and across the Weddel Sea Embayment (Turney et al., 2020) contributed to sea–level high–stands of 6 to 9 m higher than present (including a contribution from thermal expansion and mass loss from the Greenland Ice Sheet; Dutton et al., 2015).*

**Line 31: This sentence is too vague: to which parts does it refers? is it just to mention sea level high stands? if yes then provide some numbers (have a look at Colleoni et al., 2022 - it is a book chapter synthesis in which you can find some usefull refs, write me an email and I send you the pdf).**

> During Pleistocene Interglacials, Antarctic ice loss contributed to sea–level high–stands (Wilson et al., 2018; Blackburn et al., 2020; Turney et al., 2020).

In the revised manuscript (lines 32-35), we have reformulated this sentence and included the parts of the Antarctic Ice Sheet showing mass loss during Pleistocene Interglacials. It now reads as:

> *During Pleistocene Interglacials, Antarctic ice loss from the East AntarcticWilkes sub-glacial basin (Wilson et al., 2018; Blackburn et al., 2020) and across the Weddel Sea Embayment (Turney et al., 2020) contributed to sea–level high–stands of 6 to 9 m higher than present (including a contribution from thermal expansion and mass loss from the Greenland Ice Sheet; Dutton et al., 2015).*

**Line 34: why using brakets here? I would suggest: "knowledge and representation of", but not in brakets.**

> The future trajectory of the Antarctic Ice Sheet under progressing warming, however, is highly uncertain. This is due to uncertainties in the (representation of) ice-sheet processes and ice-climate interactions (Fox-Kemper et al., 2021) as well as the potentially high magnitudes and rates of recent and projected warming.

Thanks. We agree and have adjusted the formulation in the revised manuscript (lines 36-38).

**Line 35: How much is this rate? Please provide some number.**

> The future trajectory of the Antarctic Ice Sheet under progressing warming, however, is highly uncertain. This is due to uncertainties in the (representation of) ice-sheet processes and ice-climate interactions (Fox-Kemper et al., 2021) as well as the potentially high magnitudes and rates of recent and projected warming.

We have added numbers on the rate of warming based on the latest assessment of the IPCC (Gulev et al., 2021) in the revised manuscript (lines 38-40). This sentence now reads as:

> *The present rate of warming is unprecedented in at least 2000 years, with an increase of 1.09 °C in the global mean surface temperature between 1850–1900 and 2011–2020 (Gulev et al., 2021).*

**Line 39: Please provide a time for the beginning of the Holocene**

> The amount of warming projected for the end of this century under the Shared Socio-economic Pathways (e.g., for the higher-emission scenario SSP5-8.5 with an increase in global annual mean surface air temperature of 3.6 °C to 6.5 °C relative to 1850-1900; Lee et al., 2021) is comparable to the transition from the Last Glacial Maximum to the beginning of the Holocene, but is expected to develop on much shorter time-scales.

We have added a time for the beginning of the Holocene in the revised manuscript (lines 40-43) as follows:

*The amount of warming projected for the end of this century under the Shared Socio-economic Pathways (e.g., for the higher–emission scenario SSP5-8.5 with an increase in global annual mean surface air temperature of 3.6 °C to 6.5 °C relative to 1850 – 1900; Lee et al., 2021) is comparable to the transition from the Last Glacial Maximum to the beginning of the Holocene approximately 11,700 years before present, but is expected to develop on much shorter timescales.*

**Line 48: You could update with a ref of the last report on tipping points that was released early December.**

We have added this recent reference in the revised manuscript (line 52).

**Line 49: than what? You need some references here.**

This long-term sea-level response, that has already been triggered or may be triggered during the next decades (but unfolds over the following centuries and millennia), might be substantially higher than and is not represented in typical sea-level projections.

We agree that this formulation is misleading. We have adjusted the wording in the revised manuscript (lines 52-55), and the sentence now reads as:

*This long–term sea–level response, that has already been triggered or may be triggered during the next decades or centuries (but unfolds thereafter over multiple centuries and millennia), might be substantially higher than the transient sea–level change, while it is not represented in typical sea–level projections (Seroussi et al., 2020; Edwards et al., 2021).*

**Line 52: "the gap", what do you mean? It is unclear, please reformulate.**

We furthermore identify the gap between the transient realized sea–level contribution from Antarctica at a particular point in time and the respective long–term committed sea–level contribution (Winkelmann et al., in review).

We here used the word 'gap' to describe the substantial difference between the transient realized sea-level contribution from Antarctica (for example, projected by the year 2100) and the corresponding long-term committed sea-level change (that may already be triggered or locked-in given the warming by e.g. 2100 but unfolds thereafter on timescales on the order of centuries to millennia). We have changed the wording in the revised manuscript (lines 57-58), and hope that it is clearer now:

*We furthermore quantify the difference or offset between the transient realized sea–level contribution from Antarctica at a particular point in time and the respective long–term committed sea–level contribution.*

**Line 53: Why citing a paper in review? Is it not the purpose of this work?. O would suggest to remove this ref, unless this paper is going to be published before this one and properly citable here.**

We have removed this reference in the revised manuscript.

**Line 60: instead of using "self-reinforcing", "positive...feedback" would be enough. This is the definition of positive feedback.**

Following the recently published Global Tipping Points Report (Lenton et al., 2023), we have changed the naming of feedbacks to 'amplifying' and 'dampening' feedback or 'positive' and 'negative' feedbacks, respectively, in the revised manuscript.

**Line 62: Perhaps it is a bit too complicated...I think that if you remove the last part of the sentence from  "owing...", it is better. Atmospheric lapse rate is a parameter in the ice sheet and atmospheric models. But it describes a elevation-T° relationship found in the troposphere. So no need here.**

> With the lowering of the ice-sheet surface due to melting, it is exposed to higher air temperatures owing to the atmospheric lapse rate.

We have adjusted this sentence in the revised manuscript (lines 66-67), following the editor's remark.

**Line 63: a critical threshold in what? T°, critical mass loss?**

> Surface melting is, in turn, enhanced, promoting persistent ice loss upon crossing a critical threshold.

In the revised manuscript (lines 67-68), we have adjusted this sentence as:

> *Surface melting is, in turn, enhanced, promoting persistent ice loss upon crossing a critical temperature threshold.*

**Line 66: Once again: "self-sustainable mechanism" or "positive feedback". Pleaase carefully check throughout the manuscript.**

Following the recently published Global Tipping Points Report (Lenton et al., 2023), we have changed the naming of feedbacks to 'amplifying' and 'dampening' feedback or 'positive' and 'negative' feedbacks, respectively, in the revised manuscript.

**Line 68: Remove "grounding lines": just for writing style to avoid repeating "grounding"**

> In a theoretical flowline setup, it was shown that, due to the ice flux being a nonlinear function of the ice thickness, grounding lines of ice sheets grounded below sea level on a retrograde, inland sloping bed are unstable (Marine Ice Sheet Instability; Weertman, 1974; Schoof, 2007).

We have removed 'grounding lines' in the revised manuscript.

**Line 74: remove "to ice mass changes". This is not necessary here.**

Ice loss may be dampened, on the other hand, by negative feedbacks such as introduced by e.g., the isostatic rebound of the solid Earth underlying the ice sheet to ice mass changes, which could potentially stabilize West Antarctic grounding lines (Coulon et al., 2021; Barletta et al., 2018).

We have removed 'to ice mass changes' in the revised manuscript.

**Line 79: which system?**

Due to the inertia in the system and the related delay in the ice–sheet response under realistic forcing, the ice sheet's trajectory likely deviates from the ice–sheet equilibrium response to warming (Garbe et al., 2020; Rosier et al., 2021).

We here refer to ice sheets, and have adjusted the wording in the revised manuscript (lines 83-85).

**Line 79: "volume trajectory" would be more correct. And I am not sure to really understand what you mean here in this sentence. Please clarify in the text.**

Due to the inertia in the system and the related delay in the ice-sheet response under realistic forcing, the ice sheet's trajectory likely deviates from the ice-sheet equilibrium response to warming (Garbe et al., 2020; Rosier et al., 2021).

Various factors may contribute to the substantial difference between the transient realized and long-term committed sea-level change, as illustrated in Figure 1b, one of them being ice-sheet inertia (see also lines 54-58 in the original manuscript). This slow ice-sheet response to perturbations in its climatic boundary conditions manifests as a delay in the transient ice-sheet response, for example following a warming trajectory under the higher-emission pathway, when compared to the ice-sheet equilibrium response for a given warming level. Here, the ice-sheet equilibrium response (shown in black in Figure 1b) could be obtained by very slowly changing the environmental conditions (e.g., global mean temperature) at a rate which is much slower than the typical rates of changes in an ice sheet (e.g., Garbe et al., 2020; Rosier et al., 2021).

In the revised manuscript (lines 83-85), we have reformulated this sentence as follows, including the previous editor comment:

> *Due to the inertia of ice sheets and the related delay in their transient response following a realistic warming trajectory under e.g. a higher–emission pathway, the ice sheet's volume trajectory likely deviates from the ice–sheet equilibrium response to warming (Garbe et al., 2020; Rosier et al., 2021).*

**Figure 1**

- **I still don't understand what you mean by a gap.**

  Please see our response to a previous related editor comment for an explanation. In the revised manuscript, we have adjusted the wording in the caption of Figure 1, and hope that it is clearer now:

*Idealized and simplified stability diagram of the Antarctic Ice Sheet as possible tipping element, which illustrates some underlying factors potentially contributing to the substantial difference or offset between the transient realized and long-term committed ice-sheet response (in terms of sea-level contribution).*

- **Please precise here that blue is SSP2.6 and red is SSP8.5...it took me a while to understand...also because it i sunclear which quantoity is represented here. Is it global mean temperature for each scenario? Please refine the caption here for panel b).**

We thank the editor for pointing out the missing axis labels and legend for the emission pathways in Figure 1b. In the revised manuscript, we have modified Figure 1b by (1) adding a vertical axis and (2) providing a legend that relates the colours to the emission pathways.

[Figure]

**Line 219: Well...This is not really the case for PISM experiments. Ok, they are in the range of ISMIP6 initialised geometry, but please specify here that Kori-ULB is way better. PISM grounding line position for big ice shelves is far from being close to the current one and thickness difference is really large, and exceeds or underestimates the range of observed elevation changes for most Antarctica.**

We agree that the PISM initial ice-sheet states have larger deviations in their geometry from observations than the Kori-ULB initial ice-sheet states. This is a result of the different initialization approaches applied in our study.

In the revised manuscript, we have addressed and clarified differences of the initial ice-sheet states to observations in more detail, and here refer to our response to the general editor and reviewer comments for a more detailed discussion of the different initialization approaches, a comparison of the initial ice-sheet states in our study to observations and related adjustments in the revised manuscript.

**Line 259: Do you also correct precipitation? for the desertification effect? If not, why not?**

This is an important question. Changes in the ice-sheet surface elevation are accompanied by a 'local' change in the air temperatures given the atmospheric lapse rate and this is accounted for in our experiments with Kori-ULB and PISM. The change in air temperatures at the surface of the ice sheet also impacts the surface melt and runoff (thus ultimately the surface mass balance) by the use of the positive-degree-day model in our experiments.

In our experiments, we do not correct precipitation for changes in the ice-sheet surface elevation. The overall increase in precipitation with warmer regional temperatures (e.g., Frieler et al., 2015) is already accounted for in the GCM forcing that we apply. Further correcting precipitation based on a lapse-rate approach would artificially create changes in the amount of snowfall, but not necessarily for the correct reason. We believe that, at this stage, changes in the atmospheric circulation and respective precipitation patterns triggered by ice-sheet geometry changes may only be properly accounted for by a coupled simulation between an ice-sheet model and a climate model. Future research should explore changes in precipitation due to a changing ice-sheet geometry, to eventually include these processes in ice-sheet models.

**Line 271: missing star here.**

Thanks. We have added the missing star in the revised manuscript.

**Line 280: Not true: red solid lines also project a sea level rise by 2300. I suggest to improve this sentence with a more rigorous description of Fig2a.**

> Following the lower–emission pathway SSP1-2.6 results in a sea–level change ranging from -5.0 cm to +8.0 cm by the end of this century and from -0.2 m to +0.5 m in 2300 (Fig. 2a; Tab. 1). Therein, Kori-ULB projects a positive sea–level contribution for this lower–emission scenario (dashed lines), while PISM projects a sea–level drop (solid lines).

We thank the editor for pointing out the potential for improvement of the description of the multi-centennial Antarctic sea-level contribution under the lower-emission pathway SSP1-2.6 in the original manuscript.

It is correct that, under SSP1-2.6, some PISM ice-sheet trajectories show the onset of mass loss after an initial mass gain (e.g. for CESM2-WACCM climate, red solid lines in Figure 2a in the original manuscript). The Antarctic sea-level contribution projected by PISM in 2300, however, remains negative when compared to present-day (with -0.002 m and -0.038 m sea-level equivalent depending on the PISM initial ice-sheet state).

In the revised manuscript, we have included a more rigorous description of Figure 2a of the original manuscript, along with a reformulation of the description of the projected ice-sheet changes to 2300 (Sect. 3.2, lines 339-393). Please see the attached manuscript for changes in the text. In particular, a related sentence (lines 350-354) now reads as:

*Following the lower–emission pathway SSP1-2.6 to 2300 results in a sea–level change ranging from -0.2 m to +0.5 m compared to present–day (Fig. 3a, Tab. 1). Therein, Kori-ULB projects a steadily increasing Antarctic contribution to sea–level rise (Fig. 3a, dashed lines). While some PISM ice–sheet trajectories show the onset of mass loss after an initial mass gain (e.g. for CESM2-WACCM climate, indicated in blue), the Antarctic sea–level contribution projected by PISM in 2300 compared to present–day remains negative (Fig. 3a, solid lines).*

**Line 285: I found a pity not to have those figures within the text. The way they are presented in the Suppl. does not allow for this. however I would suggest to integrate a separate figure of ensemble mean at 2300 with 8 pannels (4 for each scenario) 4 for Kori-ULB and 4 for PISMs, here in the many manuscript, summarizing Figure S4 to S11.**

We find that the integrated surface mass balance remains positive for both ice–sheet models until 2300 under SSP1-2.6 (Fig. S3a). However, the response in dynamic discharge contributing to a sea–level increase on centennial timescales is higher in Kori-ULB (with ice–sheet thinning in the Amundsen Sea Embayment extending inland, Fig. S4–S5) 285 than in PISM (see Fig. S6–S7 for comparison), explaining the diverging sea–level contribution under SSP1-2.6 until 2300.

We thank the editor for this suggestion. We agree that such a figure is helpful in the main text to illustrate agreement and differences in the transient ice-sheet response between the ice-sheet models.

In the revised manuscript, we have followed the editor's suggestion. In particular, we have added a figure that shows the projected ice-thickness change in the years 2050, 2100 and 2300, depending on the ice-sheet model and the emission pathway, as Figure 3 in the revised manuscript. The ice-thickness change compared to present-day is averaged across the GCMs used to derive changes in Antarctic climate and the respective ice-sheet model configurations. This is accompanied by a reorganisation of Figure 2 of the original manuscript: The projected ice-sheet response in terms of the sea-level contribution is also given in this Figure 3 in the revised manuscript, while Figure 5 in the revised manuscript focuses on the committed changes. Please also see our response to the editor comment on Figure 2 in the original manuscript.

[Figure]

**Line 289: You never mentioned it in the previous paragraph about SSP1-2.6. Please provide a more detailed description also for SSP1-2.6 simus.**

> The initial sea–level drop by 2100 is again found in simulations from PISM and can be attributed to increasing snowfall with warming, which dominates the ice–sheet mass balance until the end of this century.

We thank the editor for pointing out this misleading formulation of an 'initial' sea-level drop by 2100 projected by PISM under SSP5-8.5 in the original manuscript.

In the revised manuscript, we have reformulated the description of the projected Antarctic sea-level contribution to 2300 (Sect. 3.2, lines 339-393). Please see the attached manuscript for changes in the text.

**Line 289: I find this explanation not convincing. Why is this not happening with Kori-ULB if this is only a matter of more precip? My guess is that the refreezing scheme influences a lot the SMB here. How much is the refreezing in Kori-ULB compared to PISM for this simulation?**

Under climate trajectories following the SSP5-8.5 emission pathway, the Antarctic ice loss varies between -6.0 cm and +6.0 cm sea–level equivalent by the end of this century, increasing to +0.7 − +3.1 m by 2300 (Fig. 2b; Tab. 1).The initial sea–level drop by 2100 is again found in simulations from PISM and can be attributed to increasing snowfall with warming, which dominates the ice–sheet mass balance until the end of this century.

We here respond to both editor comments that refer to the description of the projected ice-sheet response under the higher-emission pathway SSP5-8.5 (line 289 and the following comment on line 291 in the original manuscript).

Under SSP5-8.5, PISM and Kori-ULB project a sea-level contribution ranging between +0.7 - +3.1 m due to Antarctic mass loss by 2300. Until the end of this century, the Antarctic sea-level contribution compared to present-day projected by PISM is negative, while we find Antarctic mass loss with Kori-ULB. It is thus comparable to projected changes under SSP1-2.6, in line with Coulon et al. (2024), Lowry et al. (2021) and Edwards et al. (2021) and given a very similar evolution of Antarctic climate at least during the first half of the 21st century. In both ice-sheet models, the integrated surface mass balance remains positive until the end of the century under SSP5-8.5, with strong GCM-dependent variability. The magnitude of sub-shelf melt is higher for projections by Kori-ULB compared to PISM, following the respective levels reached at the end of the historical period. The dynamic ice-sheet response in the Amundsen Sea sector in 2100 in terms of its magnitude and inland extent is stronger in Kori-ULB than in PISM, explaining the difference in the overall ice-sheet mass balance by 2100.

In the revised manuscript, we have reformulated the description of the projected ice-sheet changes under SSP5-8.5 to 2300 (Sect. 3.2, lines 339-393) and hope that the explanation is better understandable now. Please see the attached manuscript for changes in the text.

**Line 291: is this initial increase in SMB also observed in Kori-ULB? I find all this paragraph a bit confusing. It could be better described and organised. If you describe a feature (e.g. SMB), do it for both models in the same sentence or couple of sentences, do not spread the info about one single process in different sentences through out the paragraph. It is hard to follow the speach then.**

Simulations by Kori-ULB show an earlier grounding–line retreat in the Amundsen Sea Embayment, outweighing the initial increase in the integrated surface mass balance and resulting in a positive sea–level contribution already during the 21st century.

Please see our response to the previous related editor comment.

**Line 298: Not sure of what Figure 3 shows: is it an average between Kori-ULB and PISM? This is not written in the caption. Why ice shelves do not retreat in Kori-ULB?**

The major ice shelves including the Ross Ice Shelf as well as the Filchner–Ronne Ice Shelf thin, in particular near the grounding line, and are (in PISM only) even lost sequentially by 2150 and 2300, respectively (Fig. 3b, realized; Fig. S10–S11).

Figure 3 in the original manuscript shows the mean ice thickness changes determined by both ice-sheet model and for all GMC forcings at the given point in time. We agree that this has not been clear in the original manuscript and have added an explanation to the caption where needed in the revised manuscript.

In our experiments, we find model differences in the timing of ice-shelf collapse: While Ross and Ronne-Filchner ice shelves are lost sequentially by 2150 and 2300, respectively, under the higher-emission pathway SSP5-8.5 in PISM, they are sustained longer in Kori-ULB. These major Antarctic ice shelves may be lost on multi-millennial timescales in Kori-ULB when sustaining warming that is projected by 2100 and thereafter under this higher-emission pathway.

The model differences in the timing of ice-shelf collapse may be related to the calving schemes employed in the ice-sheet models or different sub-shelf melt sensitivities to changes in ocean temperature in PICO. The values of the PICO parameters are an individual choice for each ice-sheet model. They have been chosen such that for the respective ice-sheet model observed sub-shelf melt sensitivities and / or melt rates are matched, and are based on parameter optimizations for PICO (Reese et al., 2018; Reese et al., 2023).

In the revised manuscript (lines 301-302), we have added an explanation on the choice of PICO parameters, following a previous reviewer comment:

> *The values of the PICO overturning strength parameter C and the turbulent heat exchange coefficient $\gamma_T^*$ are an individual choice for each ice–sheet model to match sub–shelf melt sensitivities and / or observed melt rates.*

Note that overall the availability of projections of the evolution of the Antarctic Ice Sheet after 2100 is limited. In particular, substantial parametric uncertainty exists, some of which (e.g., basal melt parameterizations and related parametric uncertainty) is explored in more detail in Coulon et al. (2024) in terms of the Antarctic sea-level contribution by the end of this millennium. It thus remains an important next step for future research to assess the effects of this parametric uncertainty on the Antarctic Ice Sheet response (including the potential loss of ice shelves) also on multi-millennial timescales as discussed e.g. in lines 594-605 of the original manuscript / lines 744-758 in the revised manuscript.

**Line 305: "contribution"**

Thanks. We have adjusted the wording in the revised manuscript.

**Line 305: Ok, so what do those studies show?**

> The forced response of the Antarctic Ice Sheet until 2300 is in line with the results of both Golledge et al. (2015) and Chambers et al. (2022) and is consistent with the range of -0.3 m - +3.2 m sea–level equivalent given as estimate for the Antarctic sea–level contribution in the latest IPCC assessment (Fox-Kemper et al., 2021).

We have added the sea-level change determined in Golledge et al. (2015) by the year 2300 under RCP2.6 and RCP8.5 in the revised manuscript. For consistency, we have removed the reference to Chambers et al. (2022) as the 21st-century climate is kept constant in the projections after the end of this century.

In the revised manuscript (lines 370-373), the sentence has been reformulated (also following the following comment of the editor) and integrated into the description of the projected ice-sheet changes under SSP5-8.5 (Sect. 3.2) as:

*The forced contribution of the Antarctic Ice Sheet to sea–level change until 2300 is in line with the results of e.g. Golledge et al. (2015) (showing an ice loss of +1.6 m – +2.96 m under RCP8.5) and is consistent with the Antarctic contribution to sea–level rise reported in the latest IPCC assessment of -0.3 m – +3.2 m sea–level equivalent (Fox-Kemper et al., 2021).*

**Line 306: perhaps it would be better this way "is consistent with Antarctic contribution to sea level rise reported in the last IPCC report AR6 (Fox Kemper et al., 2021) and ranging from -0.3m to +3.2 m".**

The forced response of the Antarctic Ice Sheet until 2300 is in line with the results of both Golledge et al. (2015) and Chambers et al. (2022) and is consistent with the range of -0.3 m - +3.2 m sea–level equivalent given as estimate for the Antarctic sea–level contribution in the latest IPCC assessment (Fox-Kemper et al., 2021).

Thanks for this suggestion. We have reformulated this sentence along the lines of the editor's suggestion and integrated it into the description of the projected ice-sheet changes under SSP5-8.5 (Sect. 3.2). In the revised manuscript (lines 370-373), this sentence now reads as:

*The forced contribution of the Antarctic Ice Sheet to sea–level change until 2300 is in line with the results of e.g. Golledge et al. (2015) (showing an ice loss of +1.6 m – +2.96 m under RCP8.5) and is consistent with the Antarctic contribution to sea–level rise reported in the latest IPCC assessment of -0.3 m – +3.2 m sea–level equivalent (Fox-Kemper et al., 2021).*

**Line 319: Please remove the parts related to this paper in review to which the reader to not have access. I suggest to repharse the sentence and remove this unpublished ref.**

We have removed the reference in the revised manuscript. With the rephrasing of the corresponding section in the revised manuscript, the sentence has been removed as well.

**Line 325: See, my previous comment on Fig2e and 2f and relative caption.**

We refer to our response to the editor's comment on Figure 2 in the original manuscript.

**Line 326: Why is this happening? What is the reason behind? Are all trajectories of PISM delayed? It is actually very hard to see on Fig2c. I suggest to separate in different frames the simulations from PISM from those of Kori-ULB. It would be much simpler to interpret the descripancies resulting from some feedbakc within each models, or amongst the two models.**

We find a sharp increase in the Antarctic sea–level contribution over the next millen-
nium, irrespective of the emission scenario. When following the lower–emission path-
way SSP1-2.6, the ice–sheet response levels off after a peak in the rate of Antarctic
ice loss within this millennium or at latest by the beginning of the following millennium.

In some cases, abrupt changes in the Antarctic sea–level contribution occur delayed
(compared to other trajectories under the lower–emission pathway) in PISM, with a lag
of multiple millennia to the onset of the perturbation in ice–sheet boundary conditions.

These are important questions and we thank the editor for pointing out the lack of explanation
for the delay in some ice-sheet trajectories under the lower-emission pathway SSP1-2.6 in the
original manuscript.

The delay in the Antarctic sea-level contribution is related to a later onset of substantial
grounding-line retreat in the Amundsen Sea Embayment. Such a delay in the Antarctic sea-
level contribution is most pronounced for MRI-ESM2-0 climate and may be explained by the
comparably smaller projected changes in circumantarctic ocean warming compared to the
other GCMs.

In the revised manuscript, we have reformulated the description of the committed Antarctic
sea-level contribution under SSP1-2.6 (Sect. 3.3.1, lines 408-476). Please see the attached
manuscript for changes in the text. In particular, we have added the following explanation for
the delay in the Antarctic sea-level contribution in lines 413-418 of the revised manuscript:

> *Abrupt changes in the Antarctic sea–level contribution may also occur delayed for*
> *MRI-ESM2-0 climate (Fig. 5a and c, orange), with a lag of up to multiple millennia to*
> *the onset of the perturbation in climatic boundary conditions in PISM experiments.*
> *This delay is related to a later onset of substantial grounding–line retreat in the*
> *Amundsen Sea Embayment in these simulations with comparably smaller projected*
> *oceanic changes in Antarctic climate in MRI-ESM2-0 (compared to other climate tra-*
> *jectories under the lower–emission pathway; Fig. 4a and d).*

In addition, we have adjusted Figure 2 of the original manuscript. In Figure 5 of the revised
manuscript, showing the committed ice-sheet changes, separate panels for Kori-ULB and
PISM are introduced. We hope that the individual ice-sheet trajectories following the different
GCM climates and determined by Kori-ULB and PISM can now be better distinguished.

**Line 328: What to do you mean here? The fnal sea level contribution at 7000 years
ranges from below 0 to 6 meters...Do you mean that the simulations carried out with
the same cliamte forcing converge towards the same magnitude? If yes, please be more
specific here. It is tto vague.**

These ice–sheet trajectories, however, eventually converge to the same magnitude of
sea–level contribution on multi–millennial timescales.

We thank the editor for pointing out the potentially misleading formulation related to the sea-
level commitment under SSP1-2.6 in the original manuscript.

In our simulations carried out with Kori-ULB and PISM, we find that under the lower-emission pathway SSP1-2.6 the magnitude of the committed Antarctic sea-level contribution is determined by the applied GCM forcing for each ice-sheet model and does not strongly depend on the point in time where climate is kept constant. In other words, the ice-sheet trajectories resulting from the climate projected by the same GCM are characterized by a very similar sea-level contribution on multi-millennial timescales, irrespective of the branchoff point in time.

In the revised manuscript, we have reformulated the description of the committed Antarctic sea-level contribution under SSP1-2.6 (Sect. 3.3.1, lines 408-476). Please see the attached manuscript for changes in the text. In particular, the related paragraph (lines 419-430) now reads as:

> *Irrespective of the timing of abrupt ice loss, the multi–millennial ice–sheet trajectories eventually are characterized by qualitatively different stages of ice–sheet decline with the same magnitude of the Antarctic sea–level contribution determined by the applied GCM forcing for each ice–sheet model (Fig. 5a and c). That is, in our simulations under the SSP1-2.6 pathway, we do not find a strong dependency of the long–term Antarctic sea–level commitment reached in year 7000 on the point in time after which climatic boundary conditions are kept constant (Fig. 5e, Fig. 6a). When sustaining the warming level potentially reached until year 2050, global mean sea–level may increase by +0.4 m to +4.0 m on the long–term (Fig. 5e, Tab. 1). For climatic boundary conditions representative of the end of this century and thereafter, Antarctic mass changes range between -0.2 m and +6.5 m of sea–level equivalent, which unfolds over the next millennia (Fig. 5e, Tab. 1).*
>
> *This strong modulation of the magnitude of the committed Antarctic sea–level contribution by the applied GCM forcing for each ice–sheet model (Fig. 5e, Fig. S2) is linked to substantial differences in the trajectories of atmospheric to oceanic warming between the applied GCMs under this lower–emission pathway (Fig. 4a and d). Their impact on the ice–sheet response plays out and becomes evident on longer timescales (on the order of millennia).*

**Line 329: Please reformulate: "we don't find a strong dependency on ice sheet parameters on the long-term contribution reached at 7000 years" or similar. What about the dependency of the ice sheet model?**

> In our simulations under SSP1-2.6, we do not find a strong dependency of the long–term ice–sheet configuration reached in year 7000 on the point in time after which climatic boundary conditions are kept constant (Fig. 2e; Fig. 3a, committed).

We here aim at describing the dependence of the long-term Antarctic sea-level contribution under SSP1-2.6 on the point in time after which climatic boundary conditions are kept constant (branchoff point in time). In our simulations, we find that the Antarctic sea-level commitment (in the year 7000) does not strongly depend on the branchoff point in time. We agree that the term 'long-term ice-sheet configuration reached in year 7000' in the original manuscript may be misleading.

In the revised manuscript, we have reformulated the description of the committed Antarctic sea-level contribution under SSP1-2.6 (Sect. 3.3.1, lines 408-476). Please see the attached manuscript for changes in the text. In particular, the paragraph in lines 419-430 (see also our response to the previous editor comment) refers to the editor comment.

**Line 333: This could be reformulated inn a way to somehow linked with the previous sentences on non-dependency on ice sheet parameters: "We find a strong dependency on the magnitude of long-term contribution committment on the climate forcing used to force the ice sheet models".**

> The applied GCM forcing determines the magnitude of the committed Antarctic sea–level contribution under SSP1-2.6 at a given (branchoff) point in time (Fig. 2e).

We thank the editor for the suggestion of linking (1) the non-dependency of Antarctic sea-level commitment on the branchoff point in time and (2) the dependence of Antarctic sea-level commitment on the applied GCM forcing.

In the revised manuscript, we have reformulated the description of the committed Antarctic sea-level contribution under SSP1-2.6 (Sect. 3.3.1, lines 408-476). Please see the attached manuscript for changes in the text. In particular, the paragraph in lines 419-430 (see also our response to the previous editor comment) refers to the editor comment.

**Line 335: We can't appreciate this. It would be nice to have a figure showing the comparison of climate and ocean fields from each GCMs.**

> For both ice–sheet models, stronger ocean warming in the Wilkes basin projected by UKESM1-0-LL and IPSLCM6A-LR compared to the other GCMs may promote grounding–line retreat in this region (compare Fig. S12–S15), giving rise to the upper limit of long–term ice loss found under the lower–emission pathway (Fig. 2e).

We agree and thank the reviewer for the suggestion of adding a separate figure on the changes in Antarctic climate that are projected by the different GCMs. We refer to the related general editor comment on the climate analysis for a more detailed response.

**Line 336: "explaining" instead of "giving" would be better?**

Thanks. We have changed the wording as suggested by the editor in the revised manuscript.

**Line 337: Fig2e actually shows the committment. So here better ref to Fig2c. is you write about ice loss**

> For both ice–sheet models, stronger ocean warming in the Wilkes basin projected by UKESM1-0-LL and IPSLCM6A-LR compared to the other GCMs may promote grounding–line retreat in this region (compare Fig. S12–S15), giving rise to the upper limit of long–term ice loss found under the lower–emission pathway (Fig. 2e).

We agree that the wording may be misleading here. This sentence (and the long-term ice loss) is supposed to refer to the Antarctic sea-level commitment under SSP1-2.6, as shown in Figure 2e of the original manuscript.

In the revised manuscript, we have reformulated the description of the committed Antarctic sea-level contribution under SSP1-2.6 (Sect. 3.3.1, lines 408-476). Please see the attached manuscript for changes in the text.

**Line 342: It would be nice also to indicate here the symbols you are refering too, it would definitely helps.**

We thank the editor for this suggestion. In the revised manuscript, we have made better use of the different symbols and colours in the figures and added references in the main text wherever applicable.

**Line 345: Why is Kori-ULB less sensitive than PISM? No explaination is provided about it here. I guess later on in the paper, but then it is strange to not already discuss this here, at least mention the reason and saying that it is further developed in the discussion.**

> This limits the long–term sea–level change from Antarctica to approximately +3.0 m in Kori-ULB (in combination with a retreating grounding line in Wilkes basin under UKESM1-0-LL and IPSL-CM6A-LR; Fig. 2e).

This is an important question. The uncertainty in the committed sea-level contribution under SSP1-2.6 determined by Kori-ULB and PISM is related to varying ice-sheet sensitivities to changes in climate in the Siple Coast catchment that drains Ross Ice Shelf. In particular, we find a grounding-line advance and thickening upstream of the grounding line in this region in simulations with Kori-ULB on multi-millennial timescales. Ross Ice Shelf is not lost in simulations with Kori-ULB under SSP1-2.6. In simulations with PISM, Ross Ice Shelf collapses and ice from the corresponding drainage basin is lost subsequently.

Both ice-sheet responses can likely be linked to the different initialization approaches applied in the ice-sheet models. Please see our response to the related general reviewer comment for a more detailed discussion.

In the revised manuscript, we have reformulated the description of the committed Antarctic sea-level contribution under SSP1-2.6 (Sect. 3.3.1, lines 408-476). Please see the attached manuscript for changes in the text. In particular, the related paragraph (lines 441-466), now also including possible explanations for the varying long-term ice-sheet response in the Siple Coast under SSP1-2.6, reads as:

> *The magnitude of Antarctic sea–level commitment under SSP1-2.6 is further modulated by the long–term consequences of a potential collapse of Ross and Ronne–Filchner ice shelves: In Kori-ULB, both large ice shelves are preserved to year 7000, and we find a grounding–line advance and upstream thickening in the Siple Coast region (Fig. 6a). This long–term ice–sheet response in the Siple Coast may, in parts, result from a drift of the initialisation procedure, given lower sub–shelf melt rates obtained*

*with PICO in this area compared to those that are obtained from the initialization approach to keep the ice sheet steady (Sect. 2.2.2). A thickening signal upstream of Ross Ice Shelf has also been observed over the past decades (with the stagnation of Kamb Ice Stream; Smith et al., 2020). The simulated thickening upstream of Ross Ice Shelf contributes to the decay in the long–term Antarctic sea–level contribution over time after the year 3000 in some Kori-ULB experiments, which is most pronounced for sustained MRI-ESM2-0 climate (Fig. 5a, orange). The preservation of these buttressing ice shelves limits the long–term sea–level change from Antarctica under SSP1-2.6 to less than +3.5 m in the Kori-ULB experiments (with the upper bound reached under UKESM1-0-LL and IPSL-CM6A-LR climate due to a combined grounding–line retreat in Wilkes subglacial basin and the Amundsen Sea sector; Fig. 5a and e, grey and pink filled markers). In PISM, a substantial portion of the marine ice–sheet in West Antarctica is lost with the collapse of Ross Ice Shelf and the subsequent retreat of the Siple Coast grounding line by year 7000 under most considered climate trajectories (Fig. 6a, Fig. S2). The loss of Ross Ice Shelf and the stronger sensitivity of the Siple Coast grounding line under SSP1-2.6 climate in the PISM experiments may be related to the initialized upstream grounding–line location compared to observations at present–day (compare Sect. 2.2.2, Fig. S1; as previously seen in a model initialisation in a spin–up approach, e.g., Reese et al., 2023; Sutter et al., 2023), and the simulated thinning in Ross Ice Shelf over the historical period (compare Sect. 3.1, Fig. 2c; also when determining the historical ice–sheet evolution on higher horizontal resolution using PISM, e.g., Reese et al., 2020, 2023). In addition, the higher basal melt sensitivity (compare Sect. 2.2.3 and Sect. 3.1) also translates the projected ocean warming into pronounced ice–shelf thinning (Fig. 4f, Fig. 6a). Furthermore, once grounding–line retreat is triggered, a collapse of the West Antarctic Ice Sheet may be more likely in PISM than in Kori-ULB, where low slipperiness towards the interior of West Antarctica (given low basal sliding coefficients retrieved in the inverse simulations, Sect. 2.2.2) slows down ice–sheet retreat. The combined ice loss from West Antarctica and the East Antarctic Wilkes subglacial basin in PISM gives rise to the upper end of the long–term Antarctic sea–level commitment of up to +6.5 m found under the lower–emission pathway in our experiments (Fig. 5c and e, grey open markers).*

**Line 367: Please show it. plot some integrated curve over this area comparing surface melt, SMB, accumulation etc…**

We also find a substantial ice thickness decrease in inner parts of East Antarctica grounded above sea level that is triggered under sustained high levels of warming and possibly exacerbated by the melt–elevation feedback (Fig. 3b, committed).

To disentangle the role of the melt-elevation feedback in the committed ice-sheet evolution, a comparison with experiments without the lapse-rate correction of atmospheric temperatures with changes in the ice-sheet surface elevation would be needed, as in Coulon et al. (2024). In their multi-centennial ensembles following similar CMIP6 warming trajectories, Coulon et al. (2024) find the ice-sheet collapse to be accelerated by the melt-elevation feedback over the next millennium, pointing towards similar mechanisms being at play in our experiments as well.

Note that, in the revised manuscript, we have reformulated the description of the committed Antarctic sea-level contribution under SSP5-8.5 (Sect. 3.3.2, lines 477-522). Please see the attached manuscript for changes in the text.

**Line 386: also in the physics…**

> Under equivalent warming, the long–term dynamical and topographical changes of the Antarctic Ice Sheet are largely consistent (for each ice–sheet model configuration) and uncertainty in Antarctic ice loss for a given warming level is due to inter– and intra– model uncertainty (e.g., arising in the ice–sheet model initialisation, compare Sect. 2.2.2, and by differences between applied atmospheric climatologies).

In the revised manuscript, we have reformulated the description of the committed Antarctic sea-level contribution depending on the changes in Antarctic climate (Sect. 3.4, lines 523-595). We have also added the aspect of model physics in the respective paragraph of the revised manuscript (lines 583-586):

> *Under equivalent warming, the long–term dynamical and topographical changes of the Antarctic Ice Sheet are largely con- sistent for each ice–sheet model configuration (compare Table S2). We find a spread in long–term mass loss at a given warming level due to model uncertainties (e.g., arising in the ice–sheet model initialisation and physics), which is pronounced for low to intermediate warming levels in Antarctica covered by the lower–emission pathway SSP1-2.6 (Fig. 7b).*

**Figure 2**

- **It would be nice to report the scale also in the Y axis in panels e and f.**

   We have added the scale of the vertical axis in the related figure of the revised manuscript.

- **"contribution" would be better than "response", as written on the Y axis of the frames**
- **"contribution" again.**

   We have changed the wording in the caption of the related figure in the revised manuscript from 'response' to 'contribution'.

- **I think panel e and f X-axis title deserve a bit more description: You should insert something like "branching off" in the caption of the x-axis. Then in the caption, refer to Figure 1 with a sentence: "Committed Antarctic sea level contribution in the year 7000 for the simulations with constant climate conditions from 2100, 2200 and 2300, respectively."...It took me a while, once again to figure it out.**

   We agree and thank the editor for this suggestion to modify the label of the horizontal axis of this figure. In the revised manuscript, the label 'branching off in the year' has been added. In addition, the corresponding part of the caption now reads as:

> *Committed Antarctic sea-level contribution in the year 7000 when stabilizing Antarctic climate at different points in time (that is, 'branching off' in the years 2050, 2100, 2150, 2200, 2250 and 2300; compare Sect. 2.2.1 and Fig. 1)*

- **This caption is so difficult to understand...Please write in a simple way, providing the info necessary to be understandable by the reader: "triangles corresponds to simulations initialied using MAR/RACMO, while circle corresponds to simulations initialised with MAR/RACMO".**

Throughout the revised manuscript, we have removed the information on the different atmospheric climatologies involved in the ice-sheet model initialization. We hope that the focus on model agreement and differences in the Antarctic sea-level contribution between Kori-ULB and PISM improves the clarity in the revised manuscript.

- **"ice loss by 7000 under SSP1-2.6 (e) is reported by the gray shade in f)." or something like this.**
- **"the gray shade" woudl be better?**

We have adjusted this part of the caption as suggested in the revised manuscript:

> *For comparison of the committed sea-level rise under both emission pathways, the range of Antarctic ice loss by 7000 under SSP1-2.6 (e) is reported by the light grey shade in (f).*

Please note that Figure 2 of the original manuscript has been reorganized, and an additional figure has been introduced in the revised manuscript as Figure 3. Figure 3 in the revised manuscript shows the transient ice-sheet response to 2300, in combination with the projected ice-sheet changes (in terms of the ice thickness) in the years 2050, 2100 and 2300. The committed ice-sheet response is the focus of Figure 5 in the revised manuscript. We here show the multi-millennial ice-sheet response in terms of the Antarctic sea-level contribution under SSP1-2.6 and SSP5-8.5, separately for PISM and Kori-ULB, together with the sea-level commitment in the year 7000, depending on the branchoff point in time.

[Figure]

[Figure]

**Figure 3: This caption is not clear. What is represented here? an averaged of Kori-ULB and PISM?**

Yes, the mean ice thickness change determined by both ice-sheet models and for all GMC forcings at the given point in time is shown in Figure 3 of the original manuscript. We agree that this has not been clear in the original manuscript and have added an explanation to the caption where needed in the revised manuscript.

**Line 419: we don't see this anywhere. I would suggest to make a similar figure but relative to oceanic warming,**

> This region contributes up to +1.5 m to the long–term sea–level change, which may occur for a mean ocean–temperature change exceeding +0.5°C – +1°C in this basin (depending on the ice–sheet model, with earlier onset of retreat in Kori-ULB; Fig. 5i).

We agree that the related basin-averaged ocean temperature change for Wilkes subglacial basin is difficult to infer from the colouring in Figure 5i in the original manuscript.

In the revised manuscript, Figure 5 has been modified (also compare our response to the editor comment below and the related general editor comment), and the colouring by ocean temperature change was removed. Instead, an additional Figure S4 in the Supplementary Material was added to show the dependence of the committed ice loss from this basin on changes in the ocean.

**Figure 4: What do the colors of symbols corresponds to? This is not indicated in the caption?**

We thank the editor for pointing out the missing explanation / legend for the colouring of the markers in Figure 4 of the original manuscript. The colours of the markers refer to

- the emission pathways, where the blue-green colour scale indicates SSP1-2.6 while the orange-purple colour scale corresponds to SSP5-8.5
- the branchoff point in time going from light to darker colours for keeping climate constant later in time.

In addition, the GCMs used to derive changes in Antarctic climate are indicated by the marker shape. Filled and open markers refer to simulations by Kori-ULB and PISM, respectively.

In the revised manuscript, we have added a colourbar and legend to this figure, explaining the colours and shapes of the markers.

[Figure]

**Figure 5: It is impossible to distinguish the two models here. The panels are too dense. As such, most of the description of the results in the main text is hard to follow and thus the arguments are not convincing. Please find a different display solution.**

We agree with the editor and have changed Figure 5 of the original manuscript as follows:

- We have merged basins 'Ross / Siple Coast I' and 'Ross / Siple Coast II', as they show qualitatively similar behaviour.
- The colouring has been changed to indicate the ice-sheet model.
- The marker shape has been changed to refer to the GCM used to derive the changes in Antarctic climate.

In addition, the information on basin-averaged ocean temperature changes has been removed from Figure 8 in the revised manuscript. Instead, an additional Figure S4 in the Supplementary Material has been added to show the dependence of the committed ice loss from selected basins on changes in the ocean.

[Figure]

[Figure]

**Line 479: I don't understand why you call it "a gap"? A gap would be when there is missing info. Here you have a substantial offset between the short-term and long-term sea level contribution.**

Please see our response to a previous related editor comment. In the revised manuscript, we have changed the wording from 'gap' to 'difference', and hope this sentence is clearer now.

**Lines 532 – 605: I think this section would benefit from a speach spearating PISM and Kori-ULB. First a section including all analysis of uncertainties for one model, and then another section for the model. Righ now, it is a bit complicated to follow.**

In the revised manuscript, Section 4 and, in particular, the discussion of model uncertainties (including uncertainties in ice-sheet processes, their parameterization in ice-sheet models and distinct initialization approaches; Sect. 4.2; lines 693-765) have been reorganized and elaborate on differences in ice-sheet model behaviour on shorter (multi-centennial) and longer (multi-millennial) timescales, in relation to ice-sheet modelling and initialization choices. Please see the attached manuscript for changes in the text.

**Lines 551 – 559: So this paragraph in general deals with the initialisation procedure, more than the difference in the physics. Please put a different hear then for this specific pargraph.**

In the revised manuscript, Section 4 and, in particular, the discussion of model uncertainties (including uncertainties in ice-sheet processes, their parameterization in ice-sheet models and distinct initialization approaches; Sect. 4.2; lines 693-765) have been reorganized and elaborate on differences in ice-sheet model behaviour on shorter (multi-centennial) and longer (multi-millennial) timescales, in relation to ice-sheet modelling and initialization choices. Please see the attached manuscript for changes in the text.

**Line 566: And actually the initial present-day Antarctic geometry is much better than that of PISM.**

We agree that the Kori-ULB initial ice-sheet states are closer to the observed present-day geometry of the Antarctic Ice Sheet. This is related to the initialisation approach.

In the revised manuscript, we discuss these differences between the initial ice-sheet states and observed ice-sheet geometries in more detail by including an additional paragraph in Section 2.2.2. Please see our response to the related general editor and reviewer comments for a more detailed discussion of the application of different initialization approaches for assessing the Antarctic sea-level commitment in our study and related adjustments in the revised manuscript.

**Line 572: not only, also the magnitude of melting and sublimation...RCMs are far from agreeing amoungst each other and are only calibratted on a present-day state with little melting so far.**

This is correct. We here focused on precipitation and atmospheric temperatures as variables that are used as atmospheric climatologies for driving the ice-sheet models. This sentence has been reformulated in the revised manuscript and moved to lines 197-200, following the restructuring of Section 3 and Section 4:

> *While a recent intercomparison concluded that Antarctic climate is represented reasonably well compared to observations in state–of–the–art RCMs, disagreement between the RCMs with respect to surface mass balance components (such as precipitation and atmospheric temperatures as applied to the ice–sheet models here) exists for some areas (Mottram et al., 2021).*

[revised manuscript text omitted]
 = 3\mathrm{x}10^6 \; \mathrm{m}^6 \; \mathrm{s}^{-1} \; \mathrm{kg}^{-1}$ and $\gamma_T^* = 7\mathrm{x}10^{-5} \; \mathrm{m} \; \mathrm{s}^{-1}$ (with correction of ocean properties of Schmidtko et al. (2014) to match observed present–day melt rates from Adusumilli et al. (2020)) are used for PISM experiments, as they have been found to fit melt sensitivities well (Reese et al., 2023). For Kori-ULB experiments, the overturning strength and the turbulent heat exchange coefficient are cho­sen as $C = 1\mathrm{x}10^6 \; \mathrm{m}^6 \; \mathrm{s}^{-1} \; \mathrm{kg}^{-1}$ and $\gamma_T^* = 4\mathrm{x}10^{-5} \; \mathrm{m} \; \mathrm{
[revised manuscript text omitted]

| | | MRI-ESM2-0 | CESM2-WACCM | IPSL-CM6A-LR | UKESM1-0-LL |
|---|---|---|---|---|---|
| **2050** | SSP1-2.6 | 0.90 | 1.26 | 1.54 | 1.34 |
| | SSP5-8.5 | 1.38 | 1.90 | 1.85 | 1.86 |
| **2100** | SSP1-2.6 | 1.38 | 2.43 | 1.56 | 1.26 |
| | SSP5-8.5 | 5.02 | 6.35 | 5.30 | 6.20 |
| **2150** | SSP1-2.6 | 1.77 | 2.50 | 1.10 | 1.35 |
| | SSP5-8.5 | 8.50 | 10.47 | 8.99 | 9.66 |
| **2200** | SSP1-2.6 | 1.89 | 2.86 | 1.16 | 1.39 |
| | SSP5-8.5 | 10.70 | 13.36 | 11.67 | 11.80 |
| **2250** | SSP1-2.6 | 2.26 | 2.81 | 1.44 | 1.18 |
| | SSP5-8.5 | 11.57 | 15.14 | 13.39 | 13.05 |
| **2300** | SSP1-2.6 | 2.36 | 3.58 | 0.98 | 1.22 |
| | SSP5-8.5 | 12.14 | 16.4 | 14.08 | 13.52 |

**Table S2. Ice-sheet model configurations.** Ice-sheet model configurations used for assessing Antarctic sea-level commitment. While ice—sheet initial conditions with Kori-ULB are obtained in an inverse simulation for each of the atmospheric climatologies, it is derived from a spin–up ensemble for each atmospheric climatologies in PISM. The combinations of PISM parameters for the initial states selected from these ensembles are given as well.

| | Atmospheric climatology | Sliding exponent | SIA enhancement factor | Tillwater decay rate | Till effective overburden fraction |
|---|---|---|---|---|---|
| **Kori-ULB** | MARv3.11 | - | - | - | - |
| | RACMO2.3p2 | - | - | - | - |
| **PISM** | MARv3.11 | 0.75 | 2.0 | 7 mm yr$^{-1}$ | 0.015 |
| | RACMO2.3p2 | 0.5 | 1.5 | 7 mm yr$^{-1}$ | 0.015 |

[Figure]

**Figure S1. Comparison between modelled and observed ice–sheet geometry**. Modelled present–day ice thickness in Kori-ULB (upper row) and PISM (lower row), using atmospheric climatologies based on MAR (left column) and RACMO (right column) relative to observed ice thickness (Bedmachine for Kori-ULB and Bedmap2 for PISM). Modelled and observed grounding line and calving front position are shown in black and grey, respectively. Note the different scales of the colourbar for Kori-ULB and PISM.

[Figure]

**Figure S2. Committed ice–sheet configuration under lower–emission pathways SSP1-2.6** in response to changing climatic boundary conditions projected by CMIP6 GCMs following the lower–emission pathway SSP1-2.6, as determined by Kori-ULB and PISM. Shown is the mean committed thickness change and grounding–line location (marked in black), averaged across all branchoff points in time and the respective ice–sheet model configurations. A potential loss of ice shelves is indicated by hatches. Dots mark ice–sheet advance areas.

[Figure]

**Figure S3. Committed ice–sheet configuration** in response to changing climatic boundary conditions projected by CMIP6 GCMs in year 2050, 2100 and 2300 following the higher–emission pathway SSP5-8.5, as determined by Kori-ULB and PISM. Shown is the mean committed thickness change and grounding–line location (marked in black), averaged across the respective ice–sheet model configurations. A potential loss of ice shelves is indicated by hatches. Dots mark ice–sheet advance areas.

[Figure]

**Figure S4. Ice loss from Antarctic drainage basins depending on ocean temperature change.** Long–term ice loss from different Antarctic drainage basins (as fraction of respective sea–level rise potential) for the year 7000 in response to basin–averaged ocean temperature change (compared to 1995–2014) as projected by four different GCMs (given by marker shape). Filled, green and open, blue markers correspond to the long–term ice loss determined by Kori-ULB and PISM, respectively.